# Exploring New Frontiers in Vertical Federated Learning: the Role of Saddle Point Reformulation

## Abstract

The objective of Vertical Federated Learning (VFL) is to collectively train a model using features available on different devices while sharing the same users. This paper focuses on the saddle point reformulation of the VFL problem via the classical Lagrangian function. We first demonstrate how this formulation can be solved using deterministic methods. More importantly, we explore various stochastic modifications to adapt to practical scenarios, such as employing compression techniques for efficient information transmission, enabling partial participation for asynchronous communication, and utilizing coordinate selection for faster local computation. We show that the saddle point reformulation plays a key role and opens up possibilities to use mentioned extension that seem to be impossible in the standard minimization formulation. Convergence estimates are provided for each algorithm, demonstrating their effectiveness in addressing the VFL problem. Additionally, alternative reformulations are investigated, and numerical experiments are conducted to validate performance and effectiveness of the proposed approach.

## 1 Introduction

Federated Learning is an emergent paradigm that involves training a model on private data from several devices. It can be divided into two types: horizontal (HFL)(Konečný et al., 2016; McMahan et al., 2017), where data samples are distributed across clients, and vertical (VFL) (Liu et al., 2022; Yang et al., 2023; Wei et al., 2022; Khan et al., 2023) with orthogonal data partitioning. In contrast to HFL, VFL divides the features of the same samples across clients. In this paper, we focus on the VFL problem, which appears in various fields from scoring problems (Chen et al., 2021a) to healthcare (Dankar et al., 2019) and smart manufacturing (Ge et al., 2021).

Since we deal with a distributed environment in both horizontal and vertical data partitioning, the organization of the communication process plays a crucial role in developing learning algorithms. Due to the difference in the formulations, unique characteristics and issues can arise. The HFL problem statement is very similar to the classical distributed cluster learning (Verbraeken et al., 2020), therefore, the study of various kinds of specialized HFL algorithms that take into account different aspects ranging from communication efficiency to personalization is quite extensive and comprehensive (Kairouz et al., 2021). It is a natural idea to transfer most of the techniques and useful stories from the horizontal scenario to the vertical one. And there are such results – see e.g. (Liu et al., 2022, Table 3), but not many at the moment. This can be due to the fact that the VFL problem is more ambiguous and complex from a formal optimization point of view, then it is not easy to use the theory from HFL.

Formally, VFL can be viewed as a classical minimization problem, with specifics in calculating the loss function, its gradient, or possibly higher-order derivatives. But there is another way to look at. In particular, the VFL problem can be rewritten as an augmented Lagrangian (Boyd et al., 2011, Section 8), which can be solved using the `ADMM` method (Glowinski & Marroco, 1975; Gabay & Mercier, 1976). Recent works argue that such a view of VFL is more private (Hu et al., 2019; Xie et al., 2022b). The augmented Lagrangian reformulation combined with the `ADMM` algorithm is a powerful tool for solving many practical optimization problems (not just VFL) (Bioucas-Dias & Figueiredo, 2010; Forero et al., 2010; Wahlberg et al., 2012; Wang & Banerjee, 2013; Sedghi et al.,

2014; Li et al., 2014). It provides privacy and an efficient solution for various scenarios, offering superior performance compared to other methods, making it a viable choice.

In spite of this, the already mentioned modifications from (Liu et al., 2022, Table 3) focus on the basic minimization formulation. Notably, many of these results are often empirical and lack a theoretical foundation for convergence. Moreover, current results around the Lagrangian statement have also shortcomings and are weakly studied. In particular, the widely-used approach to VFL based on ADMM is costly, as two additional minimization subproblems must be solved on each iteration. Thus, we propose to expand the theory around saddle point reformulations in this paper since the augmented Lagrangian reformulation is a good alternative to the classical minimization formulation. In particular, we address three research questions:

1. *Is there any other way to rewrite the VFL problem which can provide advantages to the standard minimization formulation?*
2. *What basic method should be used to solve the new VFL problem reformulation?*
3. *Is it possible to modify the basic method for practical use?*

### 1.1 OUR CONTRIBUTION

• **New look at VFL.** We first consider the VFL reformulation via classical Lagrangian and show that if the original VFL problem is convex, then the reformulation is convex-concave Saddle Point Problem (SPP), hence methods for SPP, such as ExtraGradient (Korpelevich, 1977; Nemirovski, 2004), can be applied to it.

• **New basic method for VFL.** The classical Lagrangian is a convex-concave SPP that can be solved using optimal methods. We introduce the basic deterministic algorithm and its efficient stochastic modifications for VFL and prove that they significantly outperform existing techniques, e.g., ADMM, in terms of iteration cost (Table 1 in Appendix A).

• **Family of practical modifications.** We present various modifications of the basic version of the algorithm to address practical needs and to make the basic algorithm more robust, including i) introducing compression operators to reduce the amount of transmitted information and solve the communication bottleneck (Alistarh et al., 2017; 2018); ii) allowing partial participation for asynchronous device communication (Ribero & Vikalo, 2020); iii) a coordinate modification to reduce the cost of local computing (Nesterov, 2012). Moreover, we show that the saddle reformulation allows to fully reveal the possibilities of these modifications.

• **More VFL reformulations.** We consider additional saddle point reformulations of the VFL problem, which have advantages, such as easier stepsize estimation, but require extra memory or existence of dual function of the loss.

• **Extension to non-convex problems.** We show how our approach can be generalized to handle non-convex learning problems. It is worth noting that all modifications are easily transferable.

• **Numerical experiments.** We show empirically that our approach can outperform existing VFL solutions in the standard minimization formulation and the saddle problem reformulation.

### 1.2 TECHNICAL PRELIMINARIES

We use $\langle a, b \rangle = \sum_{i=1}^{d} [a]_i [b]_i$ to denote the standard inner product of $a, b \in \mathbb{R}^d$ where $[a]_i$ corresponds to the $i$-th component of $a$ in the standard basis in $\mathbb{R}^d$. It induces $\ell_2$-norm in $\mathbb{R}^d$ in the following way $\|x\|_2 = \sqrt{\langle x, x \rangle}$. To denote maximal eigenvalue of positive semidefinite matrix $M \in \mathbb{R}^{d \times d}$ we use $\lambda_{\max}(M)$. Operator $\mathbb{E}\cdot$ denotes mathematical expectation, and operator $\mathbb{E}_\xi[\cdot]$ express conditional mathematical expectation w.r.t. all randomness coming from random variable $\xi$.

We also need two classical definitions for the function $f$.

**Definition 1.1.** The function $f : \mathbb{R}^d \to \mathbb{R}$, is *L-smooth*, if there exists a constant $L > 0$ such that $\forall x, y \in \mathbb{R}^d \; \|\nabla f(x) - \nabla f(y)\| \le L\|x - y\|$.

**Definition 1.2.** The function $f : \mathbb{R}^d \to \mathbb{R}$, is *convex*, if $f(x) \ge f(y) + \langle \nabla f(y), x - y \rangle$ for all $x, y \in \mathbb{R}^d$.

## 2 SADDLE POINT REFORMULATION AND EXTRAGRADIENT

**Reformulation.** The most common problem in machine learning, known as empirical risk minimization (Shalev-Shwartz & Ben-David, 2014), can be formulated as follows:

$$\min_{x \in \mathbb{R}^d} \quad [f(x) := \ell(Ax, b) + r(x)], \tag{1}$$

where $x$ is a vector of model parameters, $A \in \mathbb{R}^{s \times d}$ is a data matrix, $b \in \mathbb{R}^s$ is a vector of labels, $\ell : \mathbb{R}^s \times \mathbb{R}^s \to \mathbb{R}$ is a loss function, $r : \mathbb{R}^d \to \mathbb{R}$ is a separable regularizer, $s$ is a number of data samples and $d$ is a size of the model. This paper considers a VFL setting where data is stored across $n$ different devices. Here, the matrix $A$ is divided by columns, and each device gets different features of each of the $s$ data points (for simplicity, we assume that the matrix $A$ contains no missing data). Thus, we can rewrite (1) in the form of the VFL problem (Liu et al., 2024):

$$\min_{x \in \mathbb{R}^d} \left[ \ell \left( \sum_{i=1}^n A_i x_i, b \right) + \sum_{i=1}^n r_i(x_i) \right], \tag{2}$$

where $A_i \in \mathbb{R}^{s \times d_i}$ is a local data matrix on the $i$-th device, $x_i$ is a part of the parameters corresponding to the features of $i$-th device. It is natural to assume that $x_i$ lies on $i$th device. We additionally assume that labels $b$ contain private information and are stored in the first device. Problem (2) can be rewritten as a constrained problem with additional variable $z \in \mathbb{R}^s$:

$$\min_{x \in \mathbb{R}^d} \min_{z \in \mathbb{R}^s} \left[ \ell(z, b) + \sum_{i=1}^n r_i(x_i) \right], \text{ s.t. } \sum_{i=1}^n A_i x_i = z. \tag{3}$$

In turn, the problem with constraints can be rewritten as a saddle point problem, where the target function is the Lagrangian function

$$\min_{(x,z) \in \mathbb{R}^{d+s}} \max_{y \in \mathbb{R}^s} \left[ L(x, z, y) := \ell(z, b) + \sum_{i=1}^n r_i(x_i) + y^T \left( \sum_{i=1}^n A_i x_i - z \right) \right]. \tag{4}$$

Formulation (4) is the focus of our paper. Meanwhile, as we mentioned earlier, approaches to VFL based on ADMM also consider the Lagrangian functions with a regularizer $(\rho/2) \| \sum_{i=1}^n A_i x_i - z \|^2$ (we consider this case in Appendix C.1). For both reformulations, we propose a method that guarantees its convergence.

**Why saddle point?** Let us try to motivate the use of the saddle point reformulation (4) instead of the classical minimization problem (1) with the following example.

If we consider the classical formulation (1), which is valid for both vertical and horizontal cases, the main difference between these two types of data partitioning is the nature of the gradient computation, in particular concerning the communication process. In the horizontal case of (1), all workers have the same parameter vectors but different training samples: $\ell(Ax, b) = \sum_{i=1}^n \ell(\hat{A}_i x, b_i)$, where $\hat{A}_i \in \mathbb{R}^{s_i \times d}$, $b_i \in \mathbb{R}^{s_i}$. To compute the gradient, we simply accumulate $\nabla_x \ell(\hat{A}_j x^k, \hat{b}_j)$ from all the workers: $\nabla f(x) = \sum_{j=1}^n \nabla_x \ell(\hat{A}_j x^k, \hat{b}_j)$. In the vertical case, to calculate the gradient for the parameters $x_i$ stored on the $i$th device, it is necessary to obtain $A_j x_j$ from all the devices: $\nabla_{x_i} f(x) = A_i^T \nabla_z \ell(z, b)$ with $z = \sum_{j=1}^n A_j x_j$.

In modern applications, various kinds of stochasticity arise in communication: compression to speed up information transfer or random noise for privacy (Abadi et al., 2016). Let us consider the simplest model in which the stochasticity of communication is additive to the package on which it acts: package + noise $\xi$. As we discussed, we send different things in the horizontal and vertical cases. More specifically, the randomness we introduce has the following effect on the true gradients:

$$\nabla f(x) \to \sum_{j=1}^n [\nabla_x \ell(\hat{A}_j x^k, \hat{b}_j) + \hat{\xi}_j] = \sum_{j=1}^n \nabla_x \ell(\hat{A}_j x^k, \hat{b}_j) + \sum_{j=1}^n \hat{\xi}_j \text{ in the horizontal case,}$$

$$\nabla_{x_i} f(x) \to A_i^T \nabla_z \ell(z, b), \quad \text{where} \quad z = \sum_{j=1}^n A_j x_j + \xi_j, \text{ in the vertical one.}$$

A key detail can be seen here: the simplest additive stochasticity in the horizontal case remains additive, but in the vertical case, the influence of randomness dips much more firmly into the gradient structure. Let us look at how this kind of stochasticity affects the saddle point reformulation (4). One can note that it is also necessary to collect $A_j x_j$ during gradient computing. In more details, $\nabla_y L(x, z, y) = \sum_{j=1}^n A_j x_j - z$. With communication stochasticity this transfers to $\sum_{j=1}^n [A_j x_j + \xi_j] - z = \sum_{j=1}^n A_j x_j + \sum_{j=1}^n \xi_j - z$. The impact of randomness is additive. Because the saddle point reformulation "separates" the loss function $\ell$ and the data matrix $A$, the influence of stochasticity becomes more straightforward compared to (1).

Before moving to stochastic methods, we must learn a deterministic algorithm as a base for constructing.

**Basic method.** The most straightforward idea is to solve the saddle problem using the gradient descent-ascent method: $x^{k+1} = x^k - \gamma \nabla_x L(x^k, y^k)$, $y^{k+1} = y^k + \gamma \nabla_y L(x^k, y^k)$. Gradient descent-ascent is not the best solution (4). Indeed, it gives relatively poor convergence guarantees for strongly convex – strongly concave problems (Browder, 1966; Rockafellar, 1969; Sibony, 1970), and may diverge for convex-concave problems altogether (Goodfellow, 2016, Sections 7.2 and 8.2). Therefore, it is suggested to take the `ExtraGradient`/`Mirror Prox` method (Korpelevich, 1977; Nemirovski, 2004). The essence of this method is the use of an additional extrapolation step: $x^{k+1/2} = x^k - \gamma \nabla_x L(x^k, y^k)$, $x^{k+1} = x^k - \gamma \nabla_x L(x^{k+1/2}, y^{k+1/2})$ (the same for $y$). It can be explained by the simplest example of a two-dimensional saddle point problem $\min_{x \in \mathbb{R}} \max_{y \in \mathbb{R}} g(x, y) = xy$.

For the first-order optimality condition, it has the unique saddle point with $(x^*, y^*) = (0, 0)$. In any point $(x^k, y^k)$, the step direction of gradient descent-ascent $(-\nabla_x L(x^k, y^k), \nabla_y L(x^k, y^k))$ is orthogonal to $(x^k - x^*, y^k - y^*)$; thus the iteration of gradient descent-ascent enlarges the distance to the saddle point. However, if we make the step of `ExtraGradient`, the direction $(-\nabla_x L(x^{k+1/2}, y^{k+1/2}), \nabla_y L(x^{k+1/2}, y^{k+1/2}))$ attracts to the saddle point. Furthermore, `ExtraGradient` is optimal for convex-concave saddle point problems (Zhang et al., 2021). Iteration of the `ExtraGradient` method

---

**Algorithm 1** `EGVFL` for (4)

1: **Input:** starting point $(x^0, z^0, y^0) \in \mathbb{R}^{d+2s}$, stepsize $\gamma > 0$, number of steps $K$
2: **for** $k = 0$ **to** $K - 1$ **do**
3:   1st device sends $y^k$ to other devices
4:   All send $A_i x_i^k$ to 1st device
5:   All update: $x_i^{k+1/2} = x_i^k - \gamma \left( A_i^T y^k + \nabla r_i(x_i^k) \right)$
6:   1st updates: $z^{k+1/2} = z^k - \gamma(\nabla \ell(z^k, b) - y^k)$
7:   1st updates: $y^{k+1/2} = y^k + \gamma(\sum_{i=1}^n A_i x_i^k - z^k)$
8:   1st device sends $y^{k+1/2}$ to other devices
9:   All devices send $A_i x_i^{k+1/2}$ to 1st device
10:  All update: $x_i^{k+1} = x_i^k - \gamma(A_i^T y^{k+1/2} + \nabla r_i(x_i^{k+1/2}))$
11:  1st updates: $z^{k+1} = z^k - \gamma(\nabla \ell(z^{k+1/2}, b) - y^{k+1/2})$
12:  1st updates: $y^{k+1} = y^k + \gamma(\sum_{i=1}^n A_i x_i^{k+1/2} - z^{k+1/2})$
13: **end for**

---

for our problem (4) is given in Algorithm 1, and convergence is proved in Theorem 2.2. The proof is postponed to Appendix D.1.

**Assumption 2.1.** The function $\ell : \mathbb{R}^s \to \mathbb{R}$, is $L_\ell$-smooth and convex. Each function $r_i : \mathbb{R}^{d_i} \to \mathbb{R}$, is $L_r$-smooth and convex.

**Theorem 2.2.** *Let Assumption 2.1 hold. Let problem (4) be solved by Algorithm 1. Then for $\gamma = \frac{1}{2} \cdot \min\{1; \frac{1}{\sqrt{\lambda_{\max}(A^T A)}}; \frac{1}{L_r}; \frac{1}{L_\ell}\}$, it holds that $gap(\bar{x}^K, \bar{z}^K, \bar{y}^K) = \mathcal{O}(\frac{(1+\sqrt{\lambda_{\max}(A^T A)}+L_\ell+L_r)D^2}{K})$, where $\bar{x}^K := \frac{1}{K} \sum_{k=0}^{K-1} x^{k+1/2}$, $\bar{z}^K := \frac{1}{K} \sum_{k=0}^{K-1} z^{k+1/2}$, $\bar{y}^K := \frac{1}{K} \sum_{k=0}^{K-1} y^{k+1/2}$ and $D^2 := \max_{x,z,y \in \mathcal{X}, \mathcal{Z}, \mathcal{Y}} \left[ \|x^0 - x\|^2 + \|z^0 - z\|^2 + \|y^0 - y\|^2 \right]$.*

In Theorem 2.2, we use the convergence criterion for convex-concave saddle point problems

$$\text{gap}(x, z, y) := \max_{\tilde{y} \in \mathcal{Y}} L(x, z, \tilde{y}) - \min_{\tilde{x}, \tilde{z} \in \mathcal{X}, \mathcal{Z}} L(\tilde{x}, \tilde{z}, y). \tag{5}$$

It is important that in the formulation of Theorem 2.2 and in the definition (5), we use some bounded sets $\mathcal{X}$, $\mathcal{Z}$, $\mathcal{Y}$ although the original problem (1) is unbounded. Such an assumption is standard for the analysis of methods for convex-concave problems. Criterion (5) can also be used for unconstrained/unbounded problems. To do this, one can use the trick from (Nesterov, 2007) and introduce bounded sets $\mathcal{X}$, $\mathcal{Z}$, $\mathcal{Y}$ artificially as compact subsets of $\mathbb{R}^d, \mathbb{R}^s, \mathbb{R}^s$. This trick is valid if some solution $x^*, y^*, z^*$ lies in $\mathcal{X}$, $\mathcal{Z}$, $\mathcal{Y}$. Moreover, following (Beck, 2017, Theorems 3.59, 3.60), one can show that in Theorem 2.2 we can use the criterion: $\ell(\bar{z}^K, b) - \ell(z^*, b) + \|A\bar{x}^K - \bar{z}^K\|$, instead of (5). It is more natural and means that $A\bar{x}^K \to \bar{z}^K$ and $\ell(\bar{z}^K, b) \to \ell(Ax^*, b)$, which is what is required in the original problem (1) (see Section D.11 for more details). One can find a simplified version of gap (Xu, 2017). If we assume the existence of some solution $(x^*, z^*, y^*)$, it can be used as follows: $\text{gap}^*(x, z, y) = L(x, z, y^*) - L(x^*, z^*, y)$. Theorem 2.2 can be rewritten with $\text{gap}^*(x, z, y)$: there is no maximum in the right-hand side of the estimate, simply $\|x^0 - x^*\|^2 + \|z^0 - z^*\|^2 + \|y^0 - y^*\|^2$. But still, the use of (5) is preferable, e.g., for the already

mentioned problem $\min_{x \in \mathbb{R}} \max_{y \in \mathbb{R}} g(x, y) = xy$ with the solution $(0, 0)$, but then $\text{gap}^*(x, z, y)$ is always exactly 0. The following corollary can be easily derived from Theorem 2.2.

**Corollary 2.3.** *Under the conditions of Theorem 2.2, to achieve $\varepsilon$-solution we need $\mathcal{O}(\frac{(1+\sqrt{\lambda_{\max}(A^T A)}+L_\ell+L_r)D^2}{\varepsilon})$ iterations.*

An intriguing feature of the saddle point reformulation is that the expression $\ell(Ax, b)$ can be equivalently rewritten as $\tilde{\ell}(\tilde{A}x, b)$ with $\tilde{\ell}(y, b) = \ell(y/\beta, b)$ and $\tilde{A} = \beta A$. We can select $\beta$ such that in Theorem 2.2 we have $\sqrt{\lambda_{\max}(\tilde{A}^T \tilde{A})} = L_{\tilde{\ell}}$. It becomes evident that the appropriate choice for $\beta = L_\ell^{1/3}/\lambda_{\max}^{1/6}(A^T A)$. Then in Corollary 2.3 we can achieve $\mathcal{O}(\frac{(1+\sqrt[3]{\lambda_{\max}(A^T A) \cdot L_\ell}+L_r)D^2}{\varepsilon})$ iteration complexity (further details can be found in Section D.12). One can not `Gradient Descent` (`GD`) and `Accelerated Gradient Descent` (`AGD`) – classical deterministic methods for (1) have the following convergence estimates (Nesterov, 2003): $\mathcal{O}(\frac{(\lambda_{\max}(A^T A) \cdot L_\ell+L_r)D^2}{\varepsilon})$ and $\mathcal{O}(\frac{\sqrt{\lambda_{\max}(A^T A) \cdot L_\ell}+L_r D}{\sqrt{\varepsilon}})$ respectively. It can be seen that the obtained result is better than `Gradient Descent`, and better than `Accelerated Gradient Descent` in terms of $\lambda_{\max}(A^T A)$ viewpoint (but worse in $\varepsilon$). As we mentioned in Section 1.1, there are approaches for the saddle point reformulation, e.g., `ADMM` (Xie et al., 2022a). We compare the results in Table 1 (Appendix A).

This is possible because the loss function $\ell$ and the data matrix $A$ are "separated". As mentioned before, "separation" can be also good the stochastic algorithms, we explore them in the next section, but now let us note that Algorithm 1 presents several drawbacks. One notable limitation is its deterministic nature. In the subsequent section, we underscore the disadvantages of this characteristic and suggest alterations to enhance the foundational version of our general approach. Another significant drawback of Algorithm 1 is its reliance on the knowledge of $\lambda_{\max}(A^T A)$. Given that parts of matrix $A$ are dispersed across different devices, determining $\lambda_{\max}(A^T A)$ is challenging. However, an estimation can be made using $\lambda_{\max}(A_i^T A_i)$, as illustrated in Lemma D.13. Alternatively, we can contemplate a reformulation that negates the need for $\lambda_{\max}(A^T A)$ entirely, as discussed in Section 4. Furthermore, incorporating augmentation, as outlined in (Boyd et al., 2011), can be beneficial and straightforward for implementation. It is crucial to highlight that any variations derived from Section 4 and Appendix C, as well as any adaptations of Algorithm 1 from Section 3, can be seamlessly integrated. Another limitation is that Algorithm 1 assumes we can calculate the gradient of the function $\ell$ and the function $r$. But not all functions even allow this. For example, one can choose the $\ell_1$ regularizer as the function $r$. Or if we want to solve the constrained version of (1), we can take $r$ as an indicator function of some set $\mathcal{X}$. We consider the case where $\ell$ and $r$ are generally non-smooth but simple in Appendix B.1.

## 3 FAMILY OF MODIFICATIONS

This section presents the different modifications of Algorithm 1. These stochastic modifications are one of the main reasons for using saddle point reformulation. In any distributed optimization, including federated learning, both in its vertical and horizontal setting, the issue of communication organization is crucial. In particular, a lot of research is related to the efficiency to spend less time on communications (Konečný et al., 2016; Smith et al., 2018; Ghosh et al., 2020; Gorbunov et al., 2021), since they are from some point of view a waste of time (Kairouz et al., 2021).

### 3.1 MODIFICATION WITH QUANTIZATION FOR EFFECTIVE COMMUNICATIONS

Let us take a look at one of one of the main techniques in the fight for communication efficiency – compression (Seide et al., 2014; Alistarh et al., 2017). The following definition can formally describe the compression of communicated vectors.

**Definition 3.1.** Operator $Q : \mathbb{R}^d \to \mathbb{R}^d$ is called *unbiased compressor/quantization* if there exists a constant $\omega \geq 1$ such that for all $x \in \mathbb{R}^d$ it holds $\mathbb{E}[Q(x)] = x$, $\mathbb{E}[\|Q(x)\|^2] \leq \omega \|x\|^2$.

Operator $Q$ can be e.g., random coordinate choice or randomized rounding (Beznosikov et al., 2020).

Methods with compression in horizontal distributed learning are studied for quite a long time (Seide et al., 2014; Alistarh et al., 2017). Variance reduction methods provide a breakthrough here, initially

proposed to solve non-distributed stochastic finite-sum problems (Schmidt et al., 2017; Defazio et al., 2014; Johnson & Zhang, 2013; Nguyen et al., 2017). Several papers have shown that the variance reduction technique can be transferred to the distributed case, where stochasticity appears not from the random choice of the batch number but from the compression (Mishchenko et al., 2019; Gorbunov et al., 2021; Qian et al., 2020). For our algorithm with unbiased compression (Algorithm 2), we take the variance reduction method for saddle point problems from (Alacaoglu & Malitsky, 2021). We introduce an additional sequence of points $w_i^k$ (reference points for $x_i^k$) and $u^k$ (reference points for $y^k$). In contrast to the classical variance reduction technique, we do not introduce reference points for the $z$ variables since we do not communicate them. To update all $w_i^k$ and $u^k$ synchronously, we need to generate $b_k \in \{0, 1\}$, one can set the same random seed for generating $b^k$ on all devices to avoid additional communication. Next, we have to send full vectors $Aw_i^k$ and $u^k$ to the first device and all the others, respectively. The key is that the reference points are updated rarely, namely with low probability $p$, only when $b_k = 1$ (see lines $11 - 19$). When $w_i^k$ and $u^k$ are not updated, we only send compressed vectors $Q(y^{k+1/2} - u^k)$ and $Q(A_i x_i^{k+1/2} - A_i w_i^k)$ (lines 6, 7). Sending compressed information, rarely forwarding full packages, is the main point of Algorithm 2. Theorem 3.2 gives the convergence; its proof can be found in Appendix D.3.

**Theorem 3.2.** *Let Assumption 2.1 hold. Let problem (4) be solved by Algorithm 2 with operator $Q$ that satisfies Definition 3.1. Then for $\gamma = \frac{1}{4} \min \left\{ 1; \frac{1}{L_r}; \frac{1}{L_\ell}; \sqrt{\frac{1-\tau}{\omega \lambda_{\max}(AA^T)}}; \sqrt{\frac{1-\tau}{\omega \lambda_{\max}(AA^T)}} \right\}$, $\tau = 1 - p$ it holds that*

$$\mathbb{E} gap(\bar{x}^K, \bar{z}^K, \bar{y}^K) = \mathcal{O}([1 + \sqrt{\frac{\omega}{p}}(\sqrt{\lambda_{\max}(AA^T)}) + L_\ell + L_r]\frac{D^2}{K}),$$

*where $\bar{x}^K$, $\bar{z}^K$, $\bar{y}^K$, $D^2$ are defined in Theorem 2.2.*

In Algorithm 2, one mandatory communication round with compression occurs and possibly one more (without compression) with probability $p$. If $Q$ compress a package by a factor of $\beta$, then each iteration requires $\mathcal{O}\left(\beta^{-1} + p\right)$ data transfers on average. If $p$ is close to 1, Theorem 3.2 gives faster convergence, but more data transfer is needed. If $p$ tends to 0, the transmitted information complexity per iteration decreases but the iterative convergence rate drops. The optimal choice of $p$ is $\beta^{-1}$. For Theorem 3.2, one can obtain an analogue of Corollary 2.3, which states that if $p = \beta^{-1}$, then the iterative complexity of Algorithm 2 is $\sqrt{w\beta}$ times higher than for Algorithm 1. But the estimated amount of information transferred for Algorithm 2 is $\beta$ times less than the iterative complexity. For Algorithm 1, the complexity of the transmitted information matches the iterative one. Moreover, for most practical operators $\beta \geq w$. Hence, in the view of full information transferred, Algorithm 2 may be better than Algorithm 1.

---

**Algorithm 2** `EGVFL` with unbiased compression for (4)

1: **Input:** initial point $(x^0, z^0, y^0) \in \mathbb{R}^{d+2s}$, $(w^0, u^0) \in \mathbb{R}^{d+s}$, stepsize $\gamma > 0$, number of steps $K$
2: **for** $k = 0$ **to** $K - 1$ **do**
3:   All update: $x_i^{k+1/2} = \tau x_i^k + (1-\tau)w_i^k$
$\qquad\qquad\qquad - \gamma \left( A_i^T u^k + \nabla r_i(x_i^k) \right)$
4:   1st updates: $z^{k+1/2} = z^k - \gamma(\nabla \ell(z^k, b) - y^k)$
5:   1st updates: $y^{k+1/2} = \tau y^k + (1-\tau)u^k$
$\qquad\qquad\qquad + \gamma(\sum_{i=1}^n A_i w_i^k - z^k)$
6:   1st sends $Q(y^{k+1/2} - u^k)$ to other devices
7:   All send $Q(A_i x_i^{k+1/2} - A_i w_i^k)$ to 1st
8:   All update: $x_i^{k+1} = \tau x_i^k + (1-\tau)w_i^k$
$\qquad - \gamma(A_i^T[Q(y^{k+1/2} - u^k) + u^k] + \nabla r_i(x_i^{k+1/2}))$
9:   1st update: $z^{k+1} = z^k - \gamma(\nabla \ell(z^{k+1/2}, b) - y^{k+1/2})$
10:   1st update: $y^{k+1} = \tau y^k + (1-\tau)u^k$
$\quad + \gamma(\sum_{i=1}^n [Q(A_i x_i^{k+1/2} - A_i w_i^k) + A_i w_i^k] - z^{k+1/2})$
11:   Flip a coin $b_k \in \{0, 1\}$ where $\mathbb{P}\{b_k = 1\} = p$
12:   **if** $b_k = 1$ **then**
13:     All update: $w_i^{k+1} = x_i^k$
14:     1st updates: $u^{k+1} = y^k$
15:     All send uncompressed $A_i w_i^{k+1}$ to 1st
16:     1st sends uncompressed $u^{k+1}$ to other devices
17:   **else**
18:     All update: $w_i^{k+1} = w_i^k$
19:     1st updates: $u^{k+1} = u^k$
20:   **end if**
21: **end for**

---

The use of compression was investigated for the VFL problem, but not in the saddle point formulation. The papers (Chen et al., 2021b; Xu et al., 2021; Cai et al., 2022; Sun et al., 2023) do not provide theoretical guarantees at all. The work (Castiglia et al., 2023) investigates only special

cases of compression operators. Only the authors of paper Stanko et al. (2024) give guarantees only for the quadratic loss $\ell$: $\mathcal{O}(\omega^2 \frac{\lambda_{\max}^3(AA^T)}{\lambda_{\min}^2(AA^T)} \frac{D^2}{K^2})$. This is much worse than our guarantee estimates.

## 3.2 MODIFICATION WITH BIASED COMPRESSION FOR MORE EFFECTIVE COMMUNICATIONS

Using unbiased compression operators is more straightforward in theory, but the most popular compression operators in practice are biased (deterministic rounding (Horvath et al., 2019), greedy coordinate selection (Alistarh et al., 2018), vector decomposition (Vogels et al., 2019)) and can be described as follows.

**Definition 3.3.** Operator $C : \mathbb{R}^d \to \mathbb{R}^d$ (possibly randomized) is called a biased compressor if there exists a constant $\delta \geq 1$ such that for all $x \in \mathbb{R}^d$ it holds $\mathbb{E}\|C(x) - x\|^2 \leq \left(1 - \frac{1}{\delta}\right)\|x\|^2$.

Using biased compressors is a complex issue. It can cause divergence even for quadratic problems (Beznosikov et al., 2020). To fix this, an error compensation technique (Stich et al., 2018; Karimireddy et al., 2019; Stich & Karimireddy, 2019) can be applied. This approach accumulates non-transmitted information ($\{e_k\}, \{e_i^k\}$) and adds it to a new package at the next iteration of Algorithm 2.

---

6: 1st sends $C(y^{k+1/2} - u^k + e^k)$ to other devices
  1st updates: $e^{k+1} = y^{k+1/2} - u^k + e^k - C(y^{k+1/2} - u^k + e^k)$
7: All send $C(A_i x_i^{k+1/2} - A_i w_i^k + e_i^k)$ to 1st device
  All update: $e_i^{k+1} = A_i x_i^{k+1/2} - A_i w_i^k + e_i^k - C(A_i x_i^{k+1/2} - A_i w_i^k + e_i^k)$
8: All update: $x_i^{k+1} = \tau x_i^k + (1 - \tau)w_i^k - \gamma(A_i^T[C(y^{k+1/2} - u^k + e^k) + u^k] + \nabla r_i(x_i^{k+1/2}))$
10: 1st update: $y^{k+1} = \tau y^k + (1 - \tau)u^k + \gamma(\sum_{i=1}^n [C(A_i x_i^{k+1/2} - A_i w_i^k + e_i^k) + A_i w_i^k] - z^{k+1/2})$

---

The full version of the algorithm is given in Appendix A, Theorem 3.4 gives the convergence, and proof can be found in Appendix D.4. Note that the proof techniques of Theorems 3.2 and Theorem 3.4 differ considerably, just as the proofs of convergence of distributed GD with unbiased and biased compression (Mishchenko et al., 2019; Stich & Karimireddy, 2019).

**Theorem 3.4.** *Let Assumption 2.1 holds. Let problem (4) be solved by Algorithm 3 (Appendix A) with operators and $C$, which satisfy Definition 3.3. Then for $\tau = 1 - p$ and $\gamma = \frac{1}{4} \min\{1; \frac{1}{L_r}; \frac{1}{L_\ell}; \sqrt{\frac{1-\tau}{\delta^2[\lambda_{\max}(AA^T) + n \cdot \max_i \{\lambda_{\max}(A_i A_i^T)\}]}}; \sqrt{\frac{1-\tau}{\omega\lambda_{\max}(AA^T)}};\}$, it holds that*

$$\mathbb{E}gap(\bar{x}^K, \bar{z}^K, \bar{y}^K) = \mathcal{O}([\frac{\delta}{\sqrt{p}}(\sqrt{\lambda_{\max}(AA^T)} + n \cdot \max_{i=1,\ldots,n}\{\sqrt{\lambda_{\max}(A_i A_i^T)}\}) + L_\ell + L_r]\frac{D^2}{K}),$$

*where $\bar{x}^K, \bar{z}^K, \bar{y}^K, D^2$ are defined in Theorem 2.2.*

The choice of optimal $p$ is the same as Algorithm 2. It is enough to take $p = \beta^{-1}$, where $\beta$ is the compression power of $C$. The estimate from Theorem 3.4 shows the central theoretical problem with biased compressors. If $\delta \sim w$, the results in Theorem 3.4 are worse than in Theorem 3.2. Unfortunately, this kind of problem is inherent in all work around biased compressions – one cannot fully theoretically justify that biased compressors perform better (Gorbunov et al., 2021; Stich & Karimireddy, 2019; Richtárik et al., 2021). The only thing we can fight for is more or less acceptable convergence. Meanwhile, intuition and practical results show that biased operators are superior to unbiased ones (Beznosikov et al., 2020; Richtárik et al., 2021).

## 3.3 PARTIAL PARTICIPATION FOR ASYNCHRONOUS CLIENT CONNECTION

Algorithm 1 requires that at each iteration all devices communicate (send and receive messages). It is possible that some devices may drop out of the learning process. In this subsection, we consider a modification of Algorithm 1, where only 1 randomly selected device communicates at each iteration (Ribero & Vikalo, 2020; Chen et al., 2020; Cho et al., 2020; Lai et al., 2021). We take Algorithm 2 as a

---

6: 1st sends $y^{k+1/2}$ to other devices
7: Random device $i_k$ sends $A_{i_k} x_{i_k}^{k+1/2} - A_{i_k} w_{i_k}^k$ to 1st
8: All update: $x_i^{k+1} = \tau x_i^k + (1 - \tau)w_i^k$
  $-\gamma(A_i^T y^{k+1/2} + \nabla r_i(x_i^{k+1/2}))$
10: 1st update: $y^{k+1} = \tau y^k + (1 - \tau)u^k$
  $+\gamma(n \cdot [A_{i_k} x_{i_k}^{k+1/2} - A_{i_k} w_{i_k}^k] + \sum_{i=1}^n A_i w_i^k - z^{k+1/2})$

---

base, but instead of compression, we use random client selection and only send information from this client to the first device (see Algorithm 4 for full description).

Even though the first device sends $y^{k+1/2}$ to all devices, this does not mean that all devices need to receive the message at the exact same moment. They can get a set of several messages with $y$ at once when they contact the first device. In that case they just do several sequential updates of $x_i$.

**Theorem 3.5.** *Let Assumption 2.1 holds. Let problem (4) be solved by Algorithm 4 (Appendix A). Then for $\tau = 1 - p$ and $\gamma = \frac{1}{4}\min\{1; \frac{1}{L_r}; \frac{1}{L_\ell}; \sqrt{\frac{1-\tau}{\lambda_{\max}(AA^T)+n\cdot\max_i\{\lambda_{\max}(A_iA_i^T)\}}}\}$, it holds that*

$$\mathbb{E}gap(\bar{x}^K, \bar{z}^K, \bar{y}^K) = \mathcal{O}\left(\left[\frac{1}{\sqrt{p}}(\sqrt{\lambda_{\max}(AA^T)} + n \cdot \max_{i=1,\ldots,n}\{\sqrt{\lambda_{\max}(A_iA_i^T)}\}) + L_\ell + L_r\right]\frac{D^2}{K}\right),$$

*where $\bar{x}^K$, $\bar{z}^K$, $\bar{y}^K$, $D^2$ are defined in Theorem 2.2.*

Using the same reasonings as after Theorem 3.2, one can find the optimal choice of $p$. In Algorithm 4, one mandatory communication round with only 1 client occurs and possibly one more (with all clients) with probability $p$. Then each iteration requires $\mathcal{O}\left(n^{-1} + p\right)$ data transfers on average. The optimal choice of $p$ is $\beta^{-1}$.

### 3.4 COORDINATE MODIFICATION FOR LOW-COST LOCAL COMPUTING

The last modification is related to cheapening the cost of local computation in Algorithm 1. The most expensive local operations are matrix vector multiplications: $A_ix_i^k$ and $A_i^Ty^k$. To make them cheaper, we can apply to the idea of coordinate descent (Nesterov, 2012; Nesterov & Stich, 2017; Richtárik & Takáč, 2013; Qu & Richtárik, 2016) and compute not all coordinates for the resulting vectors $A_ix_i^k$ and $A_i^Ty^k$ but only 1, then instead of multiplying matrix by vector, we just compute the scalar product of two vectors. This is implemented in the following modification of Algorithm 2. The full version of algorithm (Algorithm 5) is given in Appendix A, Theorem 3.6 gives the convergence, proof can be found in Appendix D.6.

---

6: 1st sends $y^{k+1/2}$ to other devices

7: All choice coordinate(s) $c_i^k$, computes $\langle A_i(x_i^{k+1/2} - w_i^k), e_{c_i^k}\rangle e_{c_i^k}$ and send to 1st

8: All choice coordinate(s) $j_i^k$ and update:
$$x_i^{k+1} = \tau x_i^k + (1-\tau)w_i^k - \gamma(d_i \cdot \langle A_i^T(y^{k+1/2} - u^k), e_{j_i^k}\rangle e_{j_i^k} + A_i^Tu^k + \nabla r_i(x_i^{k+1/2}))$$

10: 1st update:
$$y^{k+1} = \tau y^k + (1-\tau)u^k + \gamma(\sum_{i=1}^n[s \cdot \langle A_i(x_i^{k+1/2} - w_i^k), e_{c_i^k}\rangle e_{c_i^k} + A_iw_i^k] - z^{k+1/2})$$

---

**Theorem 3.6.** *Let Assumption 2.1 holds. Let problem (4) be solved by Algorithm 5 (Appendix A). Then for $\gamma = \frac{1}{4}\min\{1; \frac{1}{L_r}; \frac{1}{L_\ell}; \sqrt{\frac{1-\tau}{s\lambda_{\max}(A^TA)}}; \sqrt{\frac{1-\tau}{d\max_i\{\lambda_{\max}(A_i^TA_i)\}}}\}$ and $\tau = 1-p$, it holds that*

$$\mathbb{E}gap(\bar{x}^K, \bar{z}^K, \bar{y}^K) = \mathcal{O}\left(\left[\frac{s}{\sqrt{p}}\sqrt{\lambda_{\max}(A^TA)} + \frac{d}{\sqrt{p}} \cdot \max_{i=1,\ldots,n}\{\sqrt{\lambda_{\max}(A_iA_i^T)}\} + L_\ell + L_r\right]\frac{D^2}{K}\right),$$

*where $\bar{x}^K$, $\bar{z}^K$, $\bar{y}^K$, $D^2$ are defined in Theorem 2.2.*

Using the same reasonings as after Theorem 3.2, one can find the optimal choice of $p$. In Algorithm 5, two mandatory computing of scalar products (instead of matrix vector multiplication) take places and possibly two matrix vector multiplications with probability $p$. Then each iteration requires $\mathcal{O}\left(n + s + p \cdot ns\right)$ local computations on average. The optimal choice of $p$ is $(n+s)/(ns)$.

## 4 FAMILY OF REFORMULATIONS

Let us discuss other reformulations beyond (4), e.g., a reformulation with additional variables. In formulation (3), instead of $Ax = z$, we can introduce constraints in a different way with variables $z_i \in \mathbb{R}^s$, for $i \in \{1, 2, \ldots, n\}$ as follows

$$\min_{x\in\mathbb{R}^d}\min_{z\in\mathbb{R}^s} [\ell(\sum_{i=1}^n z_i, b) + \sum_{i=1}^n r_i(x_i)], \text{ s.t. } A_ix_i = z_i \text{ for } i = 1, \ldots, n.$$

The expression in the form of a Lagrangian function is

$$\min_{(x,z)\in\mathbb{R}^{d+sn}}\max_{y\in\mathbb{R}^{sn}}[\tilde{L}(x,z,y) := \ell(\sum_{i=1}^n z_i, b) + \sum_{i=1}^n r_i(x_i) + \sum_{i=1}^n y_i^T(A_ix_i - z_i)]. \quad (6)$$

This saddle can be also solved using `ExtraGradient` (see Algorithm 6 in Appendix A).

**Theorem 4.1.** *Let Assumption 2.1 holds. Let problem (6) be solved by Algorithm 6 (Appendix A). Then for $\gamma = \frac{1}{2} \cdot \min\left\{1; \frac{1}{\sqrt{\max_i\{\lambda_{\max}(A_i^T A_i)\}}}; \frac{1}{L_r}; \frac{1}{nL_\ell}\right\}$, it holds that*

$$gap_1(\bar{x}^K, \bar{z}^K, \bar{y}^K) = \mathcal{O}\left(\frac{(1+\sqrt{\max_{i=1,\dots,n}\{\lambda_{\max}(A_i^T A_i)\}}+nL_\ell+L_r)D^2}{K}\right),$$

*where $gap_1(x,y) := \max_{\tilde{y}_i \in \tilde{\mathcal{Y}}} \tilde{L}(x,z,\tilde{y}) - \min_{\tilde{x}, z \in \mathcal{X}, \tilde{\mathcal{Z}}} \tilde{L}(\tilde{x}, \tilde{z}, y)$ and $\bar{x}^K, \bar{z}^K, \bar{y}^K, D^2$ are defined in Theorem 2.2.*

An important detail to note it is that the step $\gamma$ in Theorem 4.1 depends on $\lambda_{\max}(A_i A_i^T)$. Previous algorithms assumed knowledge of the estimate for $\lambda_{\max}(AA^T)$ which can be disadvantageous because we cannot collect $A$ on a single device, and estimating $\lambda_{\max}(AA^T)$ through $\lambda_{\max}(A_i A_i^T)$ can give deplorable results.

Other reformulations are presented in Appendix C. Although this paper focuses primarily on the classical Lagrangian, we also consider the *augmented* version, present an algorithm for it, and prove convergence estimates. The convergence estimates of the method for the augmented Lagrangian are no better (or even worse if the augmentation parameter is high) than those of the method for the classical Lagrangian. That is why we focus on the non-augmented formulation and put the augmented one in Appendix C.1. It is important to emphasize that for all reformulations, all modifications from Section 3 can be made.

## 5    EXTENSION TO NON-CONVEX MODELS

Let us consider a more general formulation where we can use arbitrary functions/models $g_i(A_i, w_i) : \mathbb{R}^{d_{w_i}} \to \mathbb{R}^{s \times d_i}$ with weights/tuning variables $w_i \in \mathbb{R}^{d_{w_i}}$ instead of fixed data matrices $A_i$ (2): $\min_{(x,w) \in \mathbb{R}^{d+d_w}} \left[\ell\left(\sum_{i=1}^n g_i(A_i, w_i)x_i, b\right) + \sum_{i=1}^n r_i(x_i)\right]$, Here, the analogue of the Lagrangian function (4) can be written as follows:

$$\min_{(x,w,z) \in \mathbb{R}^{d+d_w+s}} \max_{y \in \mathbb{R}^s} \left[\ell(z,b) + \sum_{i=1}^n r_i(x_i) + y^T\left(\sum_{i=1}^n g_i(A_i, w_i)x_i - z\right)\right]. \quad (7)$$

This SPP is generally not convex-concave, but can be solved by the modified version of Algorithm 1. The complete listing of the algorithm can be found in Algorithm 7 (Appendix A).

4: All send $g_i(A_i, w_i^k)x_i^k$ to 1st device
5: All update: $x_i^{k+1/2} = x_i^k - \gamma(g_i^T(A_i, w_i^k)y^k + \nabla r_i(x_i^k))$
    All update: $w_i^{k+1/2} = w_i^k - \gamma((y^k)^T \nabla g_i(A_i, w_i^k)x_i^k)$
7: 1st updates: $y^{k+1/2} = y^k + \gamma(\sum_{i=1}^n g_i(A_i, w_i^k)x_i^k - z^k)$
9: All send $g_i(A_i, w_i^{k+1/2})x_i^{k+1/2}$ to 1st device
10: All update: $x_i^{k+1} = x_i^k - \gamma(g_i^T(A_i, w_i^{k+1/2})y^{k+1/2} + \nabla r_i(x_i^{k+1/2}))$
    All update: $w_i^{k+1} = w_i^k - \gamma((y^{k+1/2})^T \nabla g_i(A_i, w_i^{k+1/2})x_i^{k+1/2})$
12: 1st updates: $y^{k+1} = y^k + \gamma(\sum_{i=1}^n g_i(A_i, w_i^{k+1/2})x_i^{k+1/2} - z^{k+1/2})$

## 6    EXPERIMENTS

**Regression.** We conduct experiments on the linear regression problem: $\min_{x \in \mathbb{R}^d} f(x) = \frac{1}{2}\|Ax - b\|^2 + \lambda\|x\|_2^2$. Here, the smoothness constant of gradients is $L = \lambda_{\max}(AA^T) + \lambda$ with $\lambda = \lambda_{\max}(AA^T)/10^3$. Other smoothness constants, which we use in theory for our method, are $L_\ell = 1$, $L_r = \lambda$. We take `mushrooms`, `a9a`, `w8a` and `MNIST` datasets from LibSVM library (Chang & Lin, 2011). We vertically (by features) uniformly divide the dataset between 5 devices.

This experiment uses different formulations to compare deterministic methods for solving the VFL problem. Here, we're not focusing on the distributed nature of the problems; instead, we aim to show that the saddle point reformulation using the classical Lagrangian function has merit (we investigate modifications in Appendix E) and methods for solving it can compete effectively with other approaches.

Since there are two formulations of VFL, classical minimization and saddle point, we choose several methods for each formulation. For the minimization formulation, we take `GD` as the most popular method, and `AGD` (Nesterov, 2003) as the theoretically unimprovable first-order method for smooth

convex problems. For the saddle point formulation, we consider `ADMM` and Algorithm 1. The methods are tuned according to the corresponding theory. For `GD` we choose step as $\frac{1}{L}$ (Polyak, 2020), for `Nesterov` – step as $\frac{1}{L}$ and momentum as $\frac{\sqrt{L}-\sqrt{\lambda}}{\sqrt{L}+\sqrt{\lambda}}$ (Nesterov, 2003), for `ADMM` we take regularizer parameter equal to $\frac{1}{\sqrt{\lambda_{\max}(AA^T)}}$ (Lu & Yang, 2023). Algorithm 1 is tuned according to Theorem 2.2 with and without $\beta$-trick (see disscusion after Corollary 2.3). The application of the $\beta$-trick can also be considered for other methods. However, in the case of `GD` and `Nesterov`, it does not alter the method since the data matrix $A$ and the loss function $\ell$ are not split. All methods start from zero.

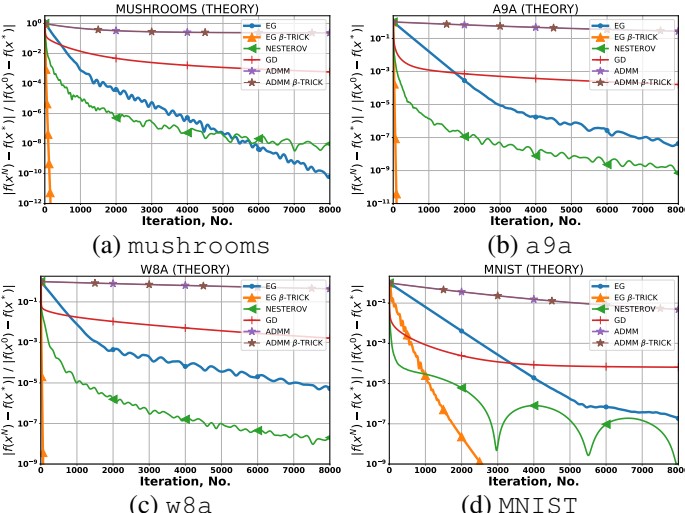

(a) `mushrooms`  (b) `a9a`

(c) `w8a`  (d) `MNIST`

Figure 1: Comparison of methods for solving the VFL problem in different formulations: minimization (`GD`, `Nesterov`) and saddle point (`ADMM`, `ExtraGradient`/Algorithm 1). The comparison is made on LibSVM datasets `mushrooms`, `a9a`, `w8a` and `MNIST`.

The results, illustrated in Figure 1, show that Algorithm 1 with the choice $\beta$ convergence dramatically faster, the basic version Algorithm 1 initially lags behind `GD` and `Nesterov`, but in terms of steady-state convergence Algorithm 1 converges faster and eventually surpasses both `GD` and sometimes `Nesterov`. Furthermore, as previously discussed in Section 1, the saddle reformulation offers advantages in terms of privacy. Significantly, we surpass our competitor in solving SPP – `ADMM`, with `ADMM` also exhibiting notably costlier iterations. The same experiments but with grid-search tuning of parameters for all methods is presented in Appendix E. In this setting, Algorithm 1 is even more faster than competitors.

**Fine-tuning of neural network.** We consider the pre-trained ResNet18 model on the ImageNet dataset. Our goal is to fine tune it on the CIFAR-10 dataset. As in the previous experiments, we take 5 clients, each client gets all the images, but only parts of them (about 1/5 of the whole image for each client). Then, each client passes its image portions through the pre-trained ResNet without the last linear layer, adjusting for the square size. As a result, each client receives embeddings corresponding to its sliced images. A new linear layer with the cross-entropy loss is trained on the embeddings of all clients, which means that the partitioning of the data is also vertical in this case. As in the previous paragraph, we use `GD`, `AGD`, `ADMM` and Algorithm 1 as methods for comparison. The methods are tuned as in the corresponding theory, since for this problem we can also estimate $L$. In the case of the 1 and `ADMM` algorithms, we also use the $\beta$-trick.

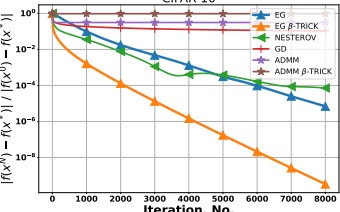

Figure 2: Comparison of methods for solving the VFL problem in different formulations: minimization (`GD`, `Nesterov`) and saddle point (`ADMM`, `ExtraGradient`/Algorithm 1). The comparison is made on `CIFAR-10` dataset.

The results reflected in Figure 2 show the superiority of Algorithm 1 over competitors. When the $\beta$-trick is used, `ExtraGradient` significantly outperforms other methods, but even without the $\beta$-trick Algorithm 1 converges slightly worse than `AGD`, but later overtakes it as well.

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

# Appendix

## A    MISSING ALGORITHMS AND TABLE

---

**Algorithm 3** `EGVFL` with biased compression for (4)

---

1: **Input:** starting point $(x^0, z^0, y^0) \in \mathbb{R}^{d+2s}$, $(w^0, u^0) \in \mathbb{R}^{d+s}$, stepsize $\gamma > 0$, number of steps $K$

2: **for** $k = 0$ **to** $K - 1$ **do**

3:     All devices in parallel update: $x_i^{k+1/2} = \tau x_i^k + (1 - \tau)w_i^k - \gamma \left(A_i^T u^k + \nabla r_i(x_i^k)\right)$

4:     First device updates: $z^{k+1/2} = z^k - \gamma(\nabla \ell(z^k, b) - y^k)$

5:     First device updates: $y^{k+1/2} = \tau y^k + (1 - \tau)u^k + \gamma(\sum_{i=1}^n A_i w_i^k - z^k)$

6:     First device compresses $C(y^{k+1/2} - u^k + e^k)$ and sends to other devices

7:     First device updates: $e^{k+1} = y^{k+1/2} - u^k + e^k - C(y^{k+1/2} - u^k + e^k)$ and sends to other devices

8:     All devices in parallel compress $C(A_i x_i^{k+1/2} - A_i w_i^k + e_i^k)$ and send to first device

9:     All devices in parallel update: $e_i^{k+1} = A_i x_i^{k+1/2} - A_i w_i^k + e_i^k - C(A_i x_i^{k+1/2} - A_i w_i^k + e_i^k)$

10:     All devices update: $x_i^{k+1} = \tau x_i^k + (1 - \tau)w_i^k - \gamma(A_i^T[C(y^{k+1/2} - u^k + e^k) + u^k] + \nabla r_i(x_i^{k+1/2}))$

11:     First device update: $z^{k+1} = z^k - \gamma(\nabla \ell(z^{k+1/2}, b) - y^{k+1/2})$

12:     First device update: $y^{k+1} = \tau y^k + (1 - \tau)u^k + \gamma(\sum_{i=1}^n[C(A_i x_i^{k+1/2} - A_i w_i^k + e_i^k) + A_i w_i^k] - z^{k+1/2})$

13:     Flip a coin $b_k \in \{0, 1\}$ where $\mathbb{P}\{b_k = 1\} = p$

14:     **if** $b_k = 1$ **then**

15:         All devices in parallel update: $w_i^{k+1} = x_i^k$

16:         First device updates: $u^{k+1} = y^k$

17:         All devices send uncompressed $A_i w_i^{k+1}$ to first device

18:         First device sends uncompressed $u^{k+1}$ to other devices

19:     **else**

20:         All devices in parallel update: $w_i^{k+1} = w_i^k$

21:         First device updates: $u^{k+1} = u^k$

22:     **end if**

23: **end for**

---

---

**Algorithm 4** `EGVFL` with partial participation for (4)

1: **Input:** starting point $(x^0, z^0, y^0) \in \mathbb{R}^{d+2s}$, $(w^0, u^0) \in \mathbb{R}^{d+s}$, stepsize $\gamma > 0$, number of steps $K$
2: **for** $k = 0$ **to** $K - 1$ **do**
3:     All devices in parallel update: $x_i^{k+1/2} = \tau x_i^k + (1 - \tau) w_i^k - \gamma \left( A_i^T u^k + \nabla r_i(x_i^k) \right)$
4:     First device updates: $z^{k+1/2} = z^k - \gamma(\nabla \ell(z^k, b) - y^k)$
5:     First device updates: $y^{k+1/2} = \tau y^k + (1 - \tau) u^k + \gamma(\sum_{i=1}^n A_i w_i^k - z^k)$
6:     First device sends $y^{k+1/2}$ to other devices
7:     Random device $i_k$ sends $(A_{i_k} x_{i_k}^{k+1/2} - A_{i_k} w_{i_k}^k)$ to first device
8:     All devices update: $x_i^{k+1} = \tau x_i^k + (1 - \tau) w_i^k - \gamma(A_i^T y^{k+1/2} + \nabla r_i(x_i^{k+1/2}))$
9:     First device update: $z^{k+1} = z^k - \gamma(\nabla \ell(z^{k+1/2}, b) - y^{k+1/2})$
10:    First device update: $y^{k+1} = \tau y^k + (1-\tau) u^k + \gamma(n \cdot [A_{i_k} x_{i_k}^{k+1/2} - A_{i_k} w_{i_k}^k] + \sum_{i=1}^n A_i w_i^k - z^{k+1/2})$
11:    Flip a coin $b_k \in \{0, 1\}$ where $\mathbb{P}\{b_k = 1\} = p$
12:    **if** $b_k = 1$ **then**
13:       All devices in parallel update: $w_i^{k+1} = x_i^k$
14:       First device updates: $u^{k+1} = y^k$
15:       All devices send uncompressed $A_i w_i^{k+1}$ to first device
16:       First device sends uncompressed $u^{k+1}$ to other devices
17:    **else**
18:       All devices in parallel update: $w_i^{k+1} = w_i^k$
19:       First device updates: $u^{k+1} = u^k$
20:    **end if**
21: **end for**

---

**Algorithm 5** `EGVFL` with coordinate choice for (4)

1: **Input:** starting point $(x^0, z^0, y^0) \in \mathbb{R}^{d+2s}$, $(w^0, u^0) \in \mathbb{R}^{d+s}$, stepsize $\gamma > 0$, number of steps $K$
2: **for** $k = 0$ **to** $K - 1$ **do**
3:     All devices in parallel update: $x_i^{k+1/2} = \tau x_i^k + (1 - \tau) w_i^k - \gamma \left( A_i^T u^k + \nabla r_i(x_i^k) \right)$
4:     First device updates: $z^{k+1/2} = z^k - \gamma(\nabla \ell(z^k, b) - y^k)$
5:     First device updates: $y^{k+1/2} = \tau y^k + (1 - \tau) u^k + \gamma(\sum_{i=1}^n A_i w_i^k - z^k)$
6:     First device sends $y^{k+1/2}$ sends to other devices
7:     All devices in parallel choice coordinate(s) $c_i^k$, computes $\langle A_i(x_i^{k+1/2} - w_i^k), e_{c_i^k} \rangle e_{c_i^k}$ and send to first device
8:     All devices choice coordinate(s) $j_i^k$ and update: $x_i^{k+1} = \tau x_i^k + (1 - \tau) w_i^k - \gamma(d_i \cdot \langle A_i^T(y^{k+1/2} - u^k), e_{j_i^k} \rangle e_{j_i^k} + A_i^T u^k + \nabla r_i(x_i^{k+1/2}))$
9:     First device update: $z^{k+1} = z^k - \gamma(\nabla \ell(z^{k+1/2}, b) - y^{k+1/2})$
10:    First device update: $y^{k+1} = \tau y^k + (1 - \tau) u^k + \gamma(\sum_{i=1}^n [s \cdot \langle A_i(x_i^{k+1/2} - w_i^k), e_{c_i^k} \rangle e_{c_i^k} + A_i w_i^k] - z^{k+1/2})$
11:    Flip a coin $b_k \in \{0, 1\}$ where $\mathbb{P}\{b_k = 1\} = p$
12:    **if** $b_k = 1$ **then**
13:       All devices in parallel update: $w_i^{k+1} = x_i^k$
14:       First device updates: $u^{k+1} = y^k$
15:       All devices send uncompressed $A_i w_i^{k+1}$ to first device
16:       First device sends uncompressed $u^{k+1}$ to other devices
17:    **else**
18:       All devices in parallel update: $w_i^{k+1} = w_i^k$
19:       First device updates: $u^{k+1} = u^k$
20:    **end if**
21: **end for**

---

---

**Algorithm 6** `EGVFL` for (6)

---

1: **Input:** starting point $(x^0, z^0, y^0) \in \mathbb{R}^{d+2s}$, stepsize $\gamma > 0$, number of steps $K$
2: **for** $k = 0$ **to** $K - 1$ **do**
3:     First device sends $y_i^k$ to other devices
4:     All devices in parallel send $A_i x_i^k$ to first device
5:     All devices in parallel update: $x_i^{k+1/2} = x_i^k - \gamma(A_i^T y_i^k + \nabla r_i(x_i^k))$
6:     First device updates: $z_i^{k+1/2} = z_i^k - \gamma(\nabla \ell(\sum_{i=1}^n z_i^k, b) - y_i^k)$
7:     First device updates: $y_i^{k+1/2} = y_i^k + \gamma(A_i x_i^k - z_i^k)$
8:     First device sends $y_i^{k+1/2}$ to other devices
9:     All devices in parallel send $A_i x_i^{k+1/2}$ to first device
10:     All devices in parallel update: $x_i^{k+1} = x_i^k - \gamma(A_i^T y_i^{k+1/2} + \nabla r_i(x_i^{k+1/2}))$
11:     First device updates: $z_i^{k+1} = z_i^k - \gamma(\nabla \ell(\sum_{i=1}^n z_i^{k+1/2}, b) - y_i^{k+1/2})$
12:     First device updates: $y_i^{k+1} = y_i^k + \gamma(A_i x_i^{k+1/2} - z_i^{k+1/2})$
13: **end for**

---

**Algorithm 7** `EGVFL` for (7)

---

1: **Input:** starting point $(x^0, w^0, z^0, y^0) \in \mathbb{R}^{d+d_w+2s}$, stepsize $\gamma > 0$, number of steps $K$
2: **for** $k = 0$ **to** $K - 1$ **do**
3:     First device sends $y^k$ to other devices
4:     All devices in parallel send $g_i(A_i, w_i^k) x_i^k$ to first device
5:     All devices in parallel update: $x_i^{k+1/2} = x_i^k - \gamma \left( g_i^T(A_i, w_i^k) y^k + \nabla r_i(x_i^k) \right)$
6:     All devices in parallel update: $w_i^{k+1/2} = w_i^k - \gamma \left( (y^k)^T \nabla g_i(A_i, w_i^k) x_i^k \right)$
7:     First device updates: $z^{k+1/2} = z^k - \gamma(\nabla \ell(z^k, b) - y^k)$
8:     First device updates: $y^{k+1/2} = y^k + \gamma \left( \sum_{i=1}^n g_i(A_i, w_i^k) x_i^k - z^k \right)$
9:     First device sends $y^{k+1/2}$ to other devices
10:     All devices in parallel send $g_i(A_i, w_i^{k+1/2}) x_i^{k+1/2}$ to first device
11:     All devices in parallel update: $x_i^{k+1} = x_i^k - \gamma \left( g_i^T(A_i, w_i^{k+1/2}) y^{k+1/2} + \nabla r_i(x_i^{k+1/2}) \right)$
12:     All devices in parallel update: $w_i^{k+1} = w_i^k - \gamma \left( (y^{k+1/2})^T \nabla g_i(A_i, w_i^{k+1/2}) x_i^{k+1/2} \right)$
13:     First device updates: $z^{k+1} = z^k - \gamma(\nabla \ell(z^{k+1/2}, b) - y^{k+1/2})$
14:     First device updates: $y^{k+1} = y^k + \gamma \left( \sum_{i=1}^n g_i(A_i, w_i^{k+1/2}) x_i^{k+1/2} - z^{k+1/2} \right)$
15: **end for**

---

Table 1: Comparison of different saddle point reformulations of the VFL problem (2) and deterministic methods for solving these reformulations.

| Formulation | Method | Iteration complexity | Local device cost of iteration | Local server cost of iteration | Communication cost of iteration |
|---|---|---|---|---|---|
| (4) | Algorithm 1 | $\mathcal{O}\left(\frac{1+\sqrt{\lambda_{\max}(A^T A)}+L_\ell+L_r}{\varepsilon}\right)$ | $\mathcal{O}\left(ds \cdot \text{Cost(a.o.)} + \text{Cost}(\nabla r)\right)$ | $\mathcal{O}\left(s \cdot \text{Cost(a.o.)} + \text{Cost}(\nabla \ell)\right)$ | $\mathcal{O}\left(s \cdot \text{Send}(1\text{ coord.})\right)$ |
| (6) | Algorithm 6 | $\mathcal{O}\left(\frac{1+\sqrt{\max_i\{\lambda_{\max}(A_i^T A_i)\}}+nL_\ell+L_r}{\varepsilon}\right)$ | $\mathcal{O}\left(ds \cdot \text{Cost(a.o.)} + \text{Cost}(\nabla r)\right)$ | $\mathcal{O}\left(s \cdot \text{Cost(a.o.)} + \text{Cost}(\nabla \ell)\right)$ | $\mathcal{O}\left(s \cdot \text{Send}(1\text{ coord.})\right)$ |
| (8) | Algorithm 9 | $\mathcal{O}\left(\frac{1+\sqrt{\lambda_{\max}(A^T A)}+L_\ell+L_r}{\varepsilon}\right)$ [1] | $\mathcal{O}\left(ds \cdot \text{Cost(a.o.)} + \text{Cost}(\nabla r)\right)$ | $\mathcal{O}\left(s \cdot \text{Cost(a.o.)} + \text{Cost}(\nabla \ell)\right)$ | $\mathcal{O}\left(s \cdot \text{Send}(1\text{ coord.})\right)$ |
| | ADMM (Boyd et al., 2011; Lu & Yang, 2023) | $\mathcal{O}\left(\frac{1+\sqrt{\lambda_{\max}(A^T A)}}{\varepsilon}\right)$ [2] | $\mathcal{O}\left(\sqrt{\frac{1+\sqrt{\lambda_{\max}(A^T A)}+L_r}{\varepsilon}}\right) \cdot \mathcal{O}\left(d^2 \cdot \text{Cost(a.o.)} + \text{Cost}(\nabla r)\right) + \mathcal{O}\left(ds \cdot \text{Cost(a.o.)}\right)$ [3] | $\mathcal{O}\left(\sqrt{\frac{nL_\ell}{\varepsilon}}\right) \cdot \mathcal{O}\left(s \cdot \text{Cost(a.o.)} + \text{Cost}(\nabla \ell)\right)$ [3] | $\mathcal{O}\left(s \cdot \text{Send}(1\text{ coord.})\right)$ |
| (9) | Algorithm 10 | $\mathcal{O}\left(\frac{1+\sqrt{\lambda_{\max}(A^T A)}+L_{\ell^*}+L_r}{\varepsilon}\right)$ | $\mathcal{O}\left(ds \cdot \text{Cost(a.o.)} + \text{Cost}(\nabla r)\right)$ | $\mathcal{O}\left(s \cdot \text{Cost(a.o.)} + \text{Cost}(\nabla \ell^*)\right)$ | $\mathcal{O}\left(s \cdot \text{Send}(1\text{ coord.})\right)$ |

[1] detailed estimate has a dependence on the parameter $\rho$, here we substitute $\rho = 1/\sqrt{\lambda_{\max}(A^T A)}$; [2] detailed estimate has a dependence on the parameter $\rho$ (see Corollary 3 from (Lu & Yang, 2023)), in particular $\rho\lambda_{\max}(A^T A) + 1/\rho$, then we substitute $\rho = 1/\sqrt{\lambda_{\max}(A^T A)}$; [3] at each iteration of ADMM (see Section 8.3 from (Boyd et al., 2011)) we need to solve subproblems on the first device/server and on all other devices, we assume that these subproblems are solved to precision $\varepsilon$ using an optimal 1st order method - Nesterov's accelerated method (Nesterov, 2003); here we also substitute $\rho = 1/\sqrt{\lambda_{\max}(A^T A)}$

*Columns:* Iteration complexity = number of iterations to achieve $\varepsilon$-solution, Local device cost of iteration = computational cost of operations on devices per one iteration, Local server cost of iteration = computational cost of operations on server per one iteration, Communication cost of iteration = communication spending during one iteration

*Notation:* $L_\ell$ = smoothness constant of $\ell$, $L_r$ = smoothness constant of $r$, $\varepsilon$ = precision of the solution, Cost(a.o.) = cost of calculations of atomic operations: addition, multiplication of two numbers, Cost($\nabla r$), Cost($\nabla \ell$) = cost of computations of gradients for $r$ and $\ell$, Send(1 coord.) = cost of sending one coordinate/one number.

## B  Missing Modifications

### B.1  Proximal modification for computational friendly losses/regularizers and constrained setting

Here we consider the case of non-smooth, but computing-friendly $\ell$ and $r$. One can modify lines 5, 6, 10, 11 in Algorithm 1 as follows.

---

**Algorithm 8** `EGVFL` for (4) with proximal friendly functions

---

1: **Input:** starting point $(x^0, z^0, y^0) \in \mathbb{R}^{d+2s}$, stepsize $\gamma > 0$, number of steps $K$
2: **for** $k = 0$ to $K - 1$ **do**
3:     First device sends $y^k$ to other devices
4:     All devices in parallel send $A_i x_i^k$ to first device
5:     All devices in parallel update: $x_i^{k+1/2} = \text{prox}_{\gamma r_i}(x_i^k - \gamma A_i^T y^k)$
6:     First device updates: $z^{k+1/2} = \text{prox}_{\gamma \ell}(z^k + \gamma y^k)$
7:     First device updates: $y^{k+1/2} = y^k + (\sum_{i=1}^n \gamma A_i x_i^k - \gamma z^k)$
8:     First device sends $y^{k+1/2}$ to other devices
9:     All devices send $\gamma A_i x_i^{k+1/2}$ to first device
10:     All devices in parallel update: $x_i^{k+1} = \text{prox}_{\gamma r_i}(x_i^k - \gamma A_i^T y^{k+1/2})$
11:     First device: $z^{k+1} = \text{prox}_{\gamma \ell}(z^k + \gamma y^{k+1/2})$
12:     First device: $y^{k+1} = y^k + (\sum_{i=1}^n \gamma A_i x_i^{k+1/2} - \gamma z^{k+1/2})$
13: **end for**

---

Here $\text{prox}_{\gamma f}$ is a proximal operator (Parikh et al., 2014): $\text{prox}_{\gamma f}(x) = \arg\min_{y \in \mathbb{R}^d}(\gamma f(y) + \frac{1}{2}\|x - y\|^2)$. In the general case, solving an additional minimization problem to calculate such an operator is necessary. But, in the case of simple, proximal-friendly functions $\ell$ and $r$, the proximal operator has a closed-form solution and can be computed exactly and sometimes for free. Theorem B.1 gives the convergence, and proof can be found in Appendix D.2.

**Theorem B.1.** *Let $\ell$ and $r$ be proximal-friendly and convex functions. Let problem (4) be solved by Algorithm 8 (Appendix A). Then for $\gamma = \frac{1}{\sqrt{2}} \cdot \min\{1; \frac{1}{\sqrt{\lambda_{\max}(A^T A)}}\}$, it holds that*

$$gap(\bar{x}^K, \bar{z}^K, \bar{y}^K) = \mathcal{O}(\frac{(1 + \sqrt{\lambda_{\max}(A^T A)})D^2}{K}), \text{ where } \bar{x}^K, \bar{z}^K, \bar{y}^K, D^2 \text{ are defined in Theorem 2.2.}$$

## C  Family of Reformulations

### C.1  Reformulation with augmentation

Let us consider the augmented version of (4):

$$\min_{(x,z)\in\mathbb{R}^{d+s}} \max_{y\in\mathbb{R}^s} \left[ L_{\text{aug}}(x,z,y) := \ell(z,b) + \sum_{i=1}^n r_i(x_i) + y^T \left( \sum_{i=1}^n A_i x_i - z \right) + \frac{\rho}{2} \| \sum_{i=1}^n A_i x_i - z \|^2 \right],$$
(8)

where $\rho \geq 0$. The statement (8) is classical and is considered in (Boyd et al., 2011). The saddle point problem (8) can also be solved using the `ExtraGradient` technique.

**Theorem C.1.** *Let Assumption 2.1 holds. Let problem (8) be solved by Algorithm 9. Then for*

$$\gamma = \frac{1}{4} \cdot \min\left\{ 1; \frac{1}{\rho}; \frac{1}{\sqrt{\lambda_{\max}(A^T A)}}; \frac{1}{\sqrt{\rho\lambda_{\max}(A^T A)}}; \frac{1}{\rho\lambda_{\max}(A^T A)}; \frac{1}{L_r}; \frac{1}{L_\ell} \right\},$$

*it holds that*

$$gap_{aug}(\bar{x}^K, \bar{z}^K, \bar{y}^K) = \mathcal{O}\left( \frac{\left(1 + \rho + \sqrt{(1+\rho)\lambda_{\max}(A^T A)} + \rho\lambda_{\max}(A^T A) + L_\ell + L_r\right)D^2}{K} \right),$$

---

**Algorithm 9** EGVFL for (8)

---

1: **Input:** starting point $(x^0, z^0, y^0) \in \mathbb{R}^{d+2s}$, stepsize $\gamma > 0$, regularizer $\rho$, number of steps $K$
2: **for** $k = 0$ **to** $K - 1$ **do**
3:     All devices in parallel send $A_i x_i^k$ to first device
4:     First device sends $y^k$ and $\sum_{i=1}^n A_i x_i^k - z^k$ to other devices
5:     All devices in parallel update:
        $x_i^{k+1/2} = x_i^k - \gamma \left( A_i^T y^k + \nabla r_i(x_i^k) + \rho A_i^T (\sum_{i=1}^n A_i x_i^k - z^k) \right)$
6:     First device updates: $z^{k+1/2} = z^k - \gamma \left( \nabla \ell(z^k, b) - y^k + \rho(z^k - \sum_{i=1}^n A_i x_i^k) \right)$
7:     First device updates: $y^{k+1/2} = y^k + \gamma(\sum_{i=1}^n A_i x_i^k - z^k)$
8:     All devices send $A_i x_i^{k+1/2}$ to first device
9:     First device sends $y^{k+1/2}$ and $\sum_{i=1}^n A_i x_i^{k+1/2} - z^{k+1/2}$ to other devices
10:    All devices in parallel update:
        $x_i^{k+1} = x_i^k - \gamma(A_i^T y^{k+1/2} + \nabla r_i(x_i^{k+1/2}) + \rho A_i^T (\sum_{i=1}^n A_i x_i^k - z^k))$
11:    First device: $z^{k+1} = z^k - \gamma(\nabla \ell(z^{k+1/2}, b) - y^{k+1/2} + \rho(z^k - \sum_{i=1}^n A_i x_i^k))$
12:    First device: $y^{k+1} = y^k + \gamma(\sum_{i=1}^n A_i x_i^{k+1/2} - z^{k+1/2})$
13: **end for**

---

*where $gap_{aug}(x, z, y)$ $:=$ $\max_{\tilde{y} \in \mathcal{Y}} L_{aug}(x, z, \tilde{y}) - \min_{\tilde{x}, \tilde{z} \in \mathcal{X}, \mathcal{Z}} L_{aug}(\tilde{x}, \tilde{z}, y)$ and $\bar{x}^K$ $:=$ $\frac{1}{K} \sum_{k=0}^{K-1} x^{k+1/2}$, $\bar{z}^K$ $:=$ $\frac{1}{K} \sum_{k=0}^{K-1} z^{k+1/2}$, $\bar{y}^K$ $:=$ $\frac{1}{K} \sum_{k=0}^{K-1} y^{k+1/2}$ and $D^2$ $:=$ $\max_{x,z,y \in \mathcal{X}, \mathcal{Z}, \mathcal{Y}} \left[ \|x^0 - x\|^2 + \|z^0 - z\|^2 + \|y^0 - y\|^2 \right]$.*

The proof is postponed to Appendix D.8. The results of Theorem C.1 are no better than Theorem 2.2, and in the case of large $\rho$ are even worse. Based on these guarantees (and they seem reasonable to us) the use of augmentation with `ExtraGradeint` in the theory does not give bonuses.

## C.2 REFORMULATION WITH DUAL LOSS

The definition of the dual function gives $\ell^*(y, b) = \max_{x \in \mathbb{R}^s} \{ \langle z, y \rangle - \ell(z, b) \}$. With small reformulation and $z = Ax$, we get that $\ell(Ax, b) = \max_{y \in \mathbb{R}^s} \{ \langle y, Ax \rangle - \ell^*(y, b) \}$. Then, one can rewrite (1) as follows,

$$\min_{x \in \mathbb{R}^d} \max_{y \in \mathbb{R}^s} \hat{L}(x, y) := \left[ -l^*(y, b) + \sum_{i=1}^n r_i(x_i) + y^T \left( \sum_{i=1}^n A_i x_i \right) \right]. \tag{9}$$

The statement (9) is simpler than (4), since it does not contain additional variables $z$, but it requires the existence of a dual function for $\ell$. The saddle point problem (9) can also be solved using the `ExtraGradient` technique.

---

**Algorithm 10** EGVFL for (9)

---

1: **Input:** starting point $(x^0, y^0) \in \mathbb{R}^{d+s}$, stepsize $\gamma > 0$, number of steps $K$
2: **for** $k = 0$ **to** $K - 1$ **do**
3:     First device sends $y^k$ to other devices
4:     All devices in parallel send $A_i x_i^k$ to first device
5:     All devices in parallel update: $x_i^{k+1/2} = x_i^k - \gamma(A_i^T y^k + \nabla r_i(x_i^k))$
6:     First device updates: $y^{k+1/2} = y^k - \gamma(\nabla \ell^*(y^k, b) - \sum_{i=1}^n A_i x_i^k)$
7:     First device sends $y^{k+1/2}$ to other devices
8:     All devices in parallel send $A_i x_i^{k+1/2}$ to first device
9:     All devices in parallel update: $x_i^{k+1} = x_i^k - \gamma(A_i^T y^{k+1/2} + \nabla r_i(x_i^{k+1/2}))$
10:    First device updates: $y^{k+1} = y^k - \gamma(\nabla \ell^*(y^{k+1/2}, b) - \sum_{i=1}^n A_i x_i^{k+1/2})$
11: **end for**

---

**Theorem C.2.** *Let $l^*$ be $L_{\ell^*}$-smooth and convex, $r$ be $L_r$-smooth and convex. Let problem (9) be solved by Algorithm 10. Then for*

$$\gamma = \tfrac{1}{2} \cdot \min\left\{ 1; \frac{1}{\sqrt{\lambda_{\max}(A^T A)}}; \frac{1}{L_r}; \frac{1}{L_{\ell^*}} \right\},$$

*it holds that*

$$gap_2(\bar{x}^K, \bar{y}^K) = \mathcal{O}\left( \frac{\left(1 + \sqrt{\lambda_{\max}(A^T A)} + L_{\ell^*} + L_r\right) \hat{D}^2}{K} \right),$$

*where $gap_2(x, y) := \max_{\tilde{y} \in \mathcal{Y}} \hat{L}(x, \tilde{y}) - \min_{\tilde{x} \in \mathcal{X}} \hat{L}(\tilde{x}, y)$ and $\bar{x}^K := \frac{1}{K} \sum_{k=0}^{K-1} x^{k+1/2}$, $\bar{y}^K := \frac{1}{K} \sum_{k=0}^{K-1} y^{k+1/2}$ and $\hat{D}^2 := \max_{x,y \in \mathcal{X}, \mathcal{Y}} \left[ \|x^0 - x\|^2 + \|y^0 - y\|^2 \right]$.*

The proof is postponed to Appendix D.9.

## C.3 Reformulation with dual loss and regularizer

If we introduce dual functions for both $\ell$ and $r$, then equation 4 can be rewritten as follows

$$\max_{y \in \mathbb{R}^s} \left[ -\sum_{i=1}^{n} r_i^*(-A_i^T y) - \ell^*(y, b) \right]. \tag{10}$$

To prove it, we start from (4)

$$\min_{(x,z) \in \mathbb{R}^{d+s}} \max_{y \in \mathbb{R}^s} \quad \ell(z, b) + r(x) + y^T (Ax - z)$$

$$= \max_{y \in \mathbb{R}^s} \left[ \min_{(x,z) \in \mathbb{R}^{d+s}} \left[ (-\langle z, y \rangle + \ell(z, b)) + (\langle Ax, y \rangle + r(x)) \right] \right]$$

$$= \max_{y \in \mathbb{R}^s} \left[ -\max_{z \in \mathbb{R}^s} (\langle z, y \rangle - \ell(z, b)) - \max_{x \in \mathbb{R}^d} (\langle -A^T y, x \rangle - r(x)) \right].$$

Definitions of dual functions: $\ell^*(y, b) = \max_{z \in \mathbb{R}^s} \{\langle z, y \rangle - \ell(z, b)\}$ and $r^*(-A^T y) = \max_{x \in \mathbb{R}^s} \{\langle -A^T y, x \rangle - r(x)\}$, give

$$\max_{y \in \mathbb{R}^s} \left[ -\ell^*(y, b) - r^*(-A^T y) \right].$$

Due to the separability of $r$, its conjugate is also separable. Hence, we have (10).

In fact (10) is the maximization of a concave function, which is very close to the original formulation (1). This problem can be solved by distributed variants of `GD` and not only.

# D MISSING PROOFS

## D.1 PROOF OF THEOREM 2.2

**Theorem D.1** (Theorem 2.2). *Let Assumption 2.1 holds. Let problem equation 4 be solved by Algorithm 1. Then for*

$$\gamma = \tfrac{1}{2} \cdot \min\left\{ 1; \frac{1}{\sqrt{\lambda_{\max}(A^T A)}}; \frac{1}{L_r}; \frac{1}{L_\ell} \right\},$$

*it holds that*

$$gap(\bar{x}^K, \bar{z}^K, \bar{y}^K) = \mathcal{O}\left( \frac{(1 + \sqrt{\lambda_{\max}(A^T A)} + L_\ell + L_r) D^2}{K} \right),$$

*where $\bar{x}^K := \frac{1}{K} \sum_{k=0}^{K-1} x^{k+1/2}$, $\bar{z}^K := \frac{1}{K} \sum_{k=0}^{K-1} z^{k+1/2}$, $\bar{y}^K := \frac{1}{K} \sum_{k=0}^{K-1} y^{k+1/2}$ and $D^2 := \max_{x,z,y \in \mathcal{X}, \mathcal{Z}, \mathcal{Y}} \left[ \|x^0 - x\|^2 + \|z^0 - z\|^2 + \|y^0 - y\|^2 \right]$.*

To prove the convergence it is sufficient to show that the problem is convex–concave (Lemma D.11), to estimate the Lipschitz constant of gradients and use the general results from (Nemirovski, 2004). But since proofs of the other algorithms is somewhat similar to proof of the basic algorithm, we provide the proof of Theorem 2.2 to complete the picture and to move from basic proofs to more complex ones.

*Proof.* We start the proof with the following equations on the variables $x_i^{k+1}$, $x_i^{k+1/2}$, $x_i^k$ and any $x_i \in \mathbb{R}^{d_i}$:

$$\|x_i^{k+1} - x_i\|^2 = \|x_i^k - x_i\|^2 + 2\langle x_i^{k+1} - x_i^k, x_i^{k+1} - x_i \rangle - \|x_i^{k+1} - x_i^k\|^2,$$

$$\|x_i^{k+1/2} - x_i^{k+1}\|^2 = \|x_i^k - x_i^{k+1}\|^2 + 2\langle x_i^{k+1/2} - x_i^k, x_i^{k+1/2} - x_i^{k+1} \rangle - \|x_i^{k+1/2} - x_i^k\|^2.$$

Summing up two previous inequalities and making small rearrangements, we get

$$\begin{aligned}
\|x_i^{k+1} - x_i\|^2 =& \|x_i^k - x_i\|^2 - \|x_i^{k+1/2} - x_i^k\|^2 - \|x_i^{k+1/2} - x_i^{k+1}\|^2 \\
&+ 2\langle x_i^{k+1} - x_i^k, x_i^{k+1} - x_i \rangle + 2\langle x_i^{k+1/2} - x_i^k, x_i^{k+1/2} - x_i^{k+1} \rangle.
\end{aligned}$$

Using that $x_i^{k+1} - x_i^k = -\gamma(A_i^T y^{k+1/2} + \nabla r_i(x_i^{k+1/2}))$ and $x_i^{k+1/2} - x_i^k = -\gamma(A_i^T y^k + \nabla r_i(x_i^k))$ (see lines 5 and 10 of Algorithm 1), we obtain

$$\begin{aligned}
\|x_i^{k+1} - x_i\|^2 =& \|x_i^k - x_i\|^2 - \|x_i^{k+1/2} - x_i^k\|^2 - \|x_i^{k+1/2} - x_i^{k+1}\|^2 \\
&- 2\gamma\langle A_i^T y^{k+1/2} + \nabla r_i(x_i^{k+1/2}), x_i^{k+1} - x_i \rangle \\
&- 2\gamma\langle A_i^T y^k + \nabla r_i(x_i^k), x_i^{k+1/2} - x_i^{k+1} \rangle \\
=& \|x_i^k - x_i\|^2 - \|x_i^{k+1/2} - x_i^k\|^2 - \|x_i^{k+1/2} - x_i^{k+1}\|^2 \\
&- 2\gamma\langle A_i^T y^{k+1/2} + \nabla r_i(x_i^{k+1/2}), x_i^{k+1/2} - x_i \rangle \\
&- 2\gamma\langle A_i^T(y^{k+1/2} - y^k) + \nabla r_i(x_i^{k+1/2}) - \nabla r_i(x_i^k), x_i^{k+1} - x_i^{k+1/2} \rangle \\
=& \|x_i^k - x_i\|^2 - \|x_i^{k+1/2} - x_i^k\|^2 - \|x_i^{k+1/2} - x_i^{k+1}\|^2 \\
&- 2\gamma\langle A_i(x_i^{k+1/2} - x_i), y^{k+1/2} \rangle - 2\gamma\langle \nabla r_i(x_i^{k+1/2}), x_i^{k+1/2} - x_i \rangle \\
&- 2\gamma\langle A_i(x_i^{k+1} - x_i^{k+1/2}), y^{k+1/2} - y^k \rangle \\
&- 2\gamma\langle \nabla r_i(x_i^{k+1/2}) - \nabla r_i(x_i^k), x_i^{k+1} - x_i^{k+1/2} \rangle.
\end{aligned} \tag{11}$$

Summing over all $i$ from 1 to $n$, we deduce

$$\begin{aligned}
\sum_{i=1}^n \|x_i^{k+1} - x_i\|^2 =& \sum_{i=1}^n \|x_i^k - x_i\|^2 - \sum_{i=1}^n \|x_i^{k+1/2} - x_i^k\|^2 - \sum_{i=1}^n \|x_i^{k+1/2} - x_i^{k+1}\|^2 \\
&- 2\gamma\langle \sum_{i=1}^n A_i(x_i^{k+1/2} - x_i), y^{k+1/2} \rangle - 2\gamma \sum_{i=1}^n \langle \nabla r_i(x_i^{k+1/2}), x_i^{k+1/2} - x_i \rangle
\end{aligned}$$

$$- 2\gamma\langle\sum_{i=1}^{n} A_i(x_i^{k+1} - x_i^{k+1/2}), y^{k+1/2} - y^k\rangle$$

$$- 2\gamma\sum_{i=1}^{n}\langle\nabla r_i(x_i^{k+1/2}) - \nabla r_i(x_i^k), x_i^{k+1} - x_i^{k+1/2}\rangle.$$

With notation of $A = [A_1, \ldots, A_i, \ldots, A_n]$ and notation of $x = [x_1^T, \ldots, x_i^T, \ldots, x_n^T]^T$ from equation 1 and equation 2, one can obtain that $\sum_{i=1}^{n} A_i x_i = Ax$:

$$\|x^{k+1} - x\|^2 = \|x^k - x\|^2 - \|x^{k+1/2} - x^k\|^2 - \|x^{k+1/2} - x^{k+1}\|^2$$

$$- 2\gamma\langle A(x^{k+1/2} - x), y^{k+1/2}\rangle - 2\gamma\sum_{i=1}^{n}\langle\nabla r_i(x_i^{k+1/2}), x_i^{k+1/2} - x_i\rangle$$

$$- 2\gamma\langle A(x^{k+1} - x^{k+1/2}), y^{k+1/2} - y^k\rangle$$

$$- 2\gamma\sum_{i=1}^{n}\langle\nabla r_i(x_i^{k+1/2}) - \nabla r_i(x_i^k), x_i^{k+1} - x_i^{k+1/2}\rangle$$

$$= \|x^k - x\|^2 - \|x^{k+1/2} - x^k\|^2 - \|x^{k+1/2} - x^{k+1}\|^2$$

$$- 2\gamma\langle A(x^{k+1/2} - x), y^{k+1/2}\rangle - 2\gamma\sum_{i=1}^{n}\langle\nabla r_i(x_i^{k+1/2}), x_i^{k+1/2} - x_i\rangle$$

$$- 2\gamma\langle A^T(y^{k+1/2} - y^k), x^{k+1} - x^{k+1/2}\rangle$$

$$- 2\gamma\sum_{i=1}^{n}\langle\nabla r_i(x_i^{k+1/2}) - \nabla r_i(x_i^k), x_i^{k+1} - x_i^{k+1/2}\rangle.$$

By Cauchy Schwartz inequality: $2\langle a, b\rangle \le \eta\|a\|^2 + \frac{1}{\eta}\|b\|^2$ with $a = A^T(y^{k+1/2} - y^k)$, $b = x^{k+1/2} - x^{k+1}$, $\eta = 2\gamma$ and $a = \nabla r_i(x_i^{k+1/2}) - \nabla r_i(x_i^k)$, $b = x_i^{k+1/2} - x_i^{k+1}$, $\eta = 2\gamma$, we get

$$\|x^{k+1} - x\|^2 \le \|x^k - x\|^2 - \|x^{k+1/2} - x^k\|^2 - \|x^{k+1/2} - x^{k+1}\|^2$$

$$- 2\gamma\langle A(x^{k+1/2} - x), y^{k+1/2}\rangle - 2\gamma\sum_{i=1}^{n}\langle\nabla r_i(x_i^{k+1/2}), x_i^{k+1/2} - x_i\rangle$$

$$+ 2\gamma^2\|A^T(y^{k+1/2} - y^k)\|^2 + \frac{1}{2}\|x^{k+1} - x^{k+1/2}\|^2$$

$$+ 2\gamma^2\sum_{i=1}^{n}\|\nabla r_i(x_i^{k+1/2}) - \nabla r_i(x_i^k)\|^2 + \frac{1}{2}\sum_{i=1}^{n}\|x_i^{k+1} - x_i^{k+1/2}\|^2$$

$$= \|x^k - x\|^2 - \|x^{k+1/2} - x^k\|^2$$

$$- 2\gamma\langle A(x^{k+1/2} - x), y^{k+1/2}\rangle - 2\gamma\sum_{i=1}^{n}\langle\nabla r_i(x_i^{k+1/2}), x_i^{k+1/2} - x_i\rangle$$

$$+ 2\gamma^2\|A^T(y^{k+1/2} - y^k)\|^2 + 2\gamma^2\sum_{i=1}^{n}\|\nabla r_i(x_i^{k+1/2}) - \nabla r_i(x_i^k)\|^2. \quad (12)$$

Using the same steps, one can obtain for $z \in \mathbb{R}^s$,

$$\|z^{k+1} - z\|^2 \le \|z^k - z\|^2 - \|z^{k+1/2} - z^k\|^2$$

$$+ 2\gamma\langle y^{k+1/2}, z^{k+1/2} - z\rangle - 2\gamma\langle\nabla\ell(z^{k+1/2}, b), z^{k+1/2} - z\rangle$$

$$+ 2\gamma^2\|y^{k+1/2} - y^k\|^2 + 2\gamma^2\|\nabla\ell(z^{k+1/2}, b) - \nabla\ell(z^k, b)\|^2. \quad (13)$$

and for all $y \in \mathbb{R}^s$,

$$\|y^{k+1} - y\|^2 \le \|y^k - y\|^2 - \|y^{k+1/2} - y^k\|^2$$

$$- 2\gamma\langle z^{k+1/2}, y^{k+1/2} - y\rangle + 2\gamma\langle\sum_{i=1}^{n} A_i x_i^{k+1/2}, y^{k+1/2} - y\rangle$$

$$+ 2\gamma^2 \|z^{k+1/2} - z^k\|^2 + 2\gamma^2 \left\| \sum_{i=1}^n A_i(x_i^{k+1/2} - x_i^k) \right\|^2$$

$$= \|y^k - y\|^2 - \|y^{k+1/2} - y^k\|^2$$
$$- 2\gamma \langle z^{k+1/2}, y^{k+1/2} - y \rangle + 2\gamma \langle Ax^{k+1/2}, y^{k+1/2} - y \rangle$$
$$+ 2\gamma^2 \|z^{k+1/2} - z^k\|^2 + 2\gamma^2 \|A(x^{k+1/2} - x^k)\|^2. \tag{14}$$

Here we also use notation of $A$ and $x$. Summing up (12), (13) and (14), we obtain

$$\|x^{k+1} - x\|^2 + \|z^{k+1} - z\|^2 + \|y^{k+1} - y\|^2$$
$$\leq \|x^k - x\|^2 + \|z^k - z\|^2 + \|y^k - y\|^2$$
$$- \|x^{k+1/2} - x^k\|^2 - \|z^{k+1/2} - z^k\|^2 - \|y^{k+1/2} - y^k\|^2$$
$$- 2\gamma \langle A(x^{k+1/2} - x), y^{k+1/2} \rangle + 2\gamma \langle y^{k+1/2}, z^{k+1/2} - z \rangle$$
$$- 2\gamma \langle z^{k+1/2}, y^{k+1/2} - y \rangle + 2\gamma \langle Ax^{k+1/2}, y^{k+1/2} - y \rangle$$
$$- 2\gamma \sum_{i=1}^n \langle \nabla r_i(x_i^{k+1/2}), x_i^{k+1/2} - x_i \rangle - 2\gamma \langle \nabla \ell(z^{k+1/2}, b), z^{k+1/2} - z \rangle$$
$$+ 2\gamma^2 \|A^T(y^{k+1/2} - y^k)\|^2 + 2\gamma^2 \sum_{i=1}^n \|\nabla r_i(x_i^{k+1/2}) - \nabla r_i(x_i^k)\|^2$$
$$+ 2\gamma^2 \|y^{k+1/2} - y^k\|^2 + 2\gamma^2 \|\nabla \ell(z^{k+1/2}, b) - \nabla \ell(z^k, b)\|^2$$
$$+ 2\gamma^2 \|z^{k+1/2} - z^k\|^2 + 2\gamma^2 \|A(x^{k+1/2} - x^k)\|^2.$$

Using convexity and $L_r$-smoothness of the function $r_i$ with convexity and $L_\ell$-smoothness of the function $\ell$ (Assumption 2.1), we have

$$\|x^{k+1} - x\|^2 + \|z^{k+1} - z\|^2 + \|y^{k+1} - y\|^2$$
$$\leq \|x^k - x\|^2 + \|z^k - z\|^2 + \|y^k - y\|^2$$
$$- \|x^{k+1/2} - x^k\|^2 - \|z^{k+1/2} - z^k\|^2 - \|y^{k+1/2} - y^k\|^2$$
$$- 2\gamma \langle A(x^{k+1/2} - x), y^{k+1/2} \rangle + 2\gamma \langle y^{k+1/2}, z^{k+1/2} - z \rangle$$
$$- 2\gamma \langle z^{k+1/2}, y^{k+1/2} - y \rangle + 2\gamma \langle Ax^{k+1/2}, y^{k+1/2} - y \rangle$$
$$- 2\gamma \sum_{i=1}^n \left( r_i(x_i^{k+1/2}) - r_i(x_i) \right) - 2\gamma \left( \ell(z^{k+1/2}, b) - \ell(z, b) \right)$$
$$+ 2\gamma^2 \|A^T(y^{k+1/2} - y^k)\| + 2\gamma^2 L_r^2 \sum_{i=1}^n \|x_i^{k+1/2} - x_i^k\|^2$$
$$+ 2\gamma^2 \|y^{k+1/2} - y^k\|^2 + 2\gamma^2 L_\ell^2 \|z^{k+1/2} - z^k\|^2$$
$$+ 2\gamma^2 \|z^{k+1/2} - z^k\|^2 + 2\gamma^2 \|A(x^{k+1/2} - x^k)\|^2.$$

Using the definition of $\lambda_{\max}(\cdot)$ as a maximum eigenvalue, we get

$$\|x^{k+1} - x\|^2 + \|z^{k+1} - z\|^2 + \|y^{k+1} - y\|^2$$
$$\leq \|x^k - x\|^2 + \|z^k - z\|^2 + \|y^k - y\|^2$$
$$- \|x^{k+1/2} - x^k\|^2 - \|z^{k+1/2} - z^k\|^2 - \|y^{k+1/2} - y^k\|^2$$
$$- 2\gamma \langle A(x^{k+1/2} - x), y^{k+1/2} \rangle + 2\gamma \langle y^{k+1/2}, z^{k+1/2} - z \rangle$$
$$- 2\gamma \langle z^{k+1/2}, y^{k+1/2} - y \rangle + 2\gamma \langle Ax^{k+1/2}, y^{k+1/2} - y \rangle$$
$$- 2\gamma \sum_{i=1}^n \left( r_i(x_i^{k+1/2}) - r_i(x_i) \right) - 2\gamma \left( \ell(z^{k+1/2}, b) - \ell(z, b) \right)$$
$$+ 2\gamma^2 \lambda_{\max}(AA^T) \|y^{k+1/2} - y^k\| + 2\gamma^2 L_r^2 \|x^{k+1/2} - x^k\|^2$$

$$+ 2\gamma^2 \|y^{k+1/2} - y^k\|^2 + 2\gamma^2 L_\ell^2 \|z^{k+1/2} - z^k\|^2$$
$$+ 2\gamma^2 \|z^{k+1/2} - z^k\|^2 + 2\gamma^2 \lambda_{\max}(A^T A)\|x^{k+1/2} - x^k\|^2.$$

With the choice of $\gamma \leq \frac{1}{2} \cdot \min\left\{1; \frac{1}{\sqrt{\lambda_{\max}(A^T A)}}; \frac{1}{L_r}; \frac{1}{L_\ell}\right\}$, we get

$$\|x^{k+1} - x\|^2 + \|z^{k+1} - z\|^2 + \|y^{k+1} - y\|^2$$
$$\leq \|x^k - x\|^2 + \|z^k - z\|^2 + \|y^k - y\|^2$$
$$- 2\gamma\langle A(x^{k+1/2} - x), y^{k+1/2}\rangle + 2\gamma\langle y^{k+1/2}, z^{k+1/2} - z\rangle$$
$$- 2\gamma\langle z^{k+1/2}, y^{k+1/2} - y\rangle + 2\gamma\langle Ax^{k+1/2}, y^{k+1/2} - y\rangle$$
$$- 2\gamma\sum_{i=1}^{n}\left(r_i(x_i^{k+1/2}) - r_i(x_i)\right) - 2\gamma\left(\ell(z^{k+1/2}, b) - \ell(z, b)\right)$$
$$= \|x^k - x\|^2 + \|z^k - z\|^2 + \|y^k - y\|^2$$
$$+ 2\gamma\langle Ax - z, y^{k+1/2}\rangle - 2\gamma\langle Ax^{k+1/2} - z^{k+1/2}, y\rangle$$
$$- 2\gamma\sum_{i=1}^{n}\left(r_i(x_i^{k+1/2}) - r_i(x_i)\right) - 2\gamma\left(\ell(z^{k+1/2}, b) - \ell(z, b)\right).$$

After small rearrangements, we obtain

$$\left(\ell(z^{k+1/2}, b) - \ell(z, b)\right) + \sum_{i=1}^{n}\left(r_i(x_i^{k+1/2}) - r_i(x_i)\right)$$
$$+ \langle Ax^{k+1/2} - z^{k+1/2}, y\rangle - \langle Ax - z, y^{k+1/2}\rangle$$
$$\leq \frac{1}{2\gamma}\left(\|x^k - x\|^2 + \|z^k - z\|^2 + \|y^k - y\|^2\right.$$
$$\left. - \|x^{k+1} - x\|^2 - \|z^{k+1} - z\|^2 - \|y^{k+1} - y\|^2\right).$$

Then we sum all over $k$ from $0$ to $K - 1$, divide by $K$, and have

$$\frac{1}{K}\sum_{k=0}^{K-1}\left(\ell(z^{k+1/2}, b) - \ell(z, b)\right) + \sum_{i=1}^{n}\frac{1}{K}\sum_{k=0}^{K-1}\left(r_i(x_i^{k+1/2}) - r_i(x_i)\right)$$
$$+ \langle A \cdot \frac{1}{K}\sum_{k=0}^{K-1}x^{k+1/2} - \frac{1}{K}\sum_{k=0}^{K-1}z^{k+1/2}, y\rangle - \langle Ax - z, \frac{1}{K}\sum_{k=0}^{K-1}y^{k+1/2}\rangle$$
$$\leq \frac{1}{2\gamma K}\left(\|x^0 - x\|^2 + \|z^0 - z\|^2 + \|y^0 - y\|^2\right.$$
$$\left. - \|x^K - x\|^2 - \|z^K - z\|^2 - \|y^K - y\|^2\right)$$
$$\leq \frac{1}{2\gamma K}(\|x^0 - x\|^2 + \|z^0 - z\|^2 + \|y^0 - y\|^2).$$

With Jensen inequality for convex functions $\ell$ and $r_i$, one can note that

$$\ell\left(\frac{1}{K}\sum_{k=0}^{K-1}z^{k+1/2}, b\right) \leq \frac{1}{K}\sum_{k=0}^{K-1}\ell(z^{k+1/2}, b),$$
$$r_i\left(\frac{1}{K}\sum_{k=0}^{K-1}x_i^{k+1/2}\right) \leq \frac{1}{K}\sum_{k=0}^{K-1}r_i(x_i^{k+1/2}).$$

Then, with notation $\bar{x}_i^K = \frac{1}{K}\sum_{k=0}^{K-1}x_i^{k+1/2}$, $\bar{z}^K = \frac{1}{K}\sum_{k=0}^{K-1}z^{k+1/2}$, $\bar{y}^K = \frac{1}{K}\sum_{k=0}^{K-1}y^{k+1/2}$, we have

$$\ell(\bar{z}^K, b) - \ell(z, b) + \sum_{i=1}^{n}\left(r_i(\bar{x}_i^K) - r_i(x_i)\right) + \langle A\bar{x}^K - \bar{z}^K, y\rangle - \langle Ax - z, \bar{y}^K\rangle$$

$$\leq \frac{1}{2\gamma K}(\|x^0 - x\|^2 + \|z^0 - z\|^2 + \|y^0 - y\|^2).$$

Following the definition equation 5, we only need to take the maximum in the variable $y \in \mathcal{Y}$ and the minimum in $x \in \mathcal{X}$ and $z \in \mathcal{Z}$.

$$\text{gap}(\bar{x}^K, \bar{z}^K, \bar{y}^K)$$

$$= \max_{y \in \mathcal{Y}} L(\bar{x}^K, \bar{z}^K, y) - \min_{x,z \in \mathcal{X}, \mathcal{Z}} L(x, z, \bar{y}^K)$$

$$= \max_{y \in \mathcal{Y}} \left[ \ell(\bar{z}^K, b) + \sum_{i=1}^n r_i(\bar{x}_i^K) + \langle A\bar{x}^K - \bar{z}^K, y \rangle \right]$$

$$- \min_{x,z \in \mathcal{X}, \mathcal{Z}} \left[ \ell(z, b) + \sum_{i=1}^n r_i(x_i) + \langle Ax - z, \bar{y}^K \rangle \right]$$

$$= \max_{y \in \mathcal{Y}} \max_{x,z \in \mathcal{X}, \mathcal{Z}} \left[ \ell(\bar{z}^K, b) - \ell(z, b) + \sum_{i=1}^n \left( r_i(\bar{x}_i^K) - r_i(x_i) \right) + \langle A\bar{x}^K - \bar{z}^K, y \rangle - \langle Ax - z, \bar{y}^K \rangle \right]$$

$$\leq \frac{1}{2\gamma K} (\max_{x \in \mathcal{X}} \|x^0 - x\|^2 + \max_{z \in \mathcal{Z}} \|z^0 - z\|^2 + \max_{y \in \mathcal{Y}} \|y^0 - y\|^2).$$

$$(15)$$

To complete the proof in the cases equation 15, it remains to put $\gamma = \frac{1}{2} \cdot \min\left\{1; \frac{1}{\sqrt{\lambda_{\max}(A^T A)}}; \frac{1}{L_r}; \frac{1}{L_\ell}\right\}$. $\qquad\square$

### D.2 PROOF OF THEOREM B.1

**Theorem D.2** (Theorem B.1). *Let $\ell$ and $r$ be proximal-friendly and convex functions. Let problem (4) be solved by Algorithm 8 (Appendix A). Then for*

$$\gamma = \frac{1}{\sqrt{2}} \cdot \min\left\{1; \frac{1}{\sqrt{\lambda_{\max}(A^T A)}}\right\},$$

*it holds that*

$$gap(\bar{x}^K, \bar{z}^K, \bar{y}^K) = \mathcal{O}\left(\frac{(1 + \sqrt{\lambda_{\max}(A^T A)})D^2}{K}\right),$$

*where $\bar{x}^K := \frac{1}{K} \sum_{k=0}^{K-1} x^{k+1/2}$, $\bar{z}^K := \frac{1}{K} \sum_{k=0}^{K-1} z^{k+1/2}$, $\bar{y}^K := \frac{1}{K} \sum_{k=0}^{K-1} y^{k+1/2}$ and $D^2 := \max_{x,z,y \in \mathcal{X}, \mathcal{Z}, \mathcal{Y}} \left[\|x^0 - x\|^2 + \|z^0 - z\|^2 + \|y^0 - y\|^2\right]$.*

Before we start proving Theorem B.1, we need a small lemma concerning the proximal operator.

**Lemma D.3.** *Let $h$ be convex and $z^+ = prox_{\gamma h}(z)$ with some $\gamma > 0$. Then for all $x \in \mathbb{R}^d$ the following inequality holds*

$$\langle z^+ - z, x - z^+ \rangle \geq \gamma \left( h(z^+) - h(x) \right).$$

*Proof of Lemma D.3.* We use convexity of the function $\gamma h$ and get for any $h'(z^+) \in \partial h(z^+)$

$$\gamma(h(x) - h(z^+)) - \langle h'(z^+), x - z^+ \rangle \geq 0.$$

With definition of the proximal operator and the optimality condition, one can note that $z - z^+ \in \gamma \partial h(z^+)$. The only thing left to do is to take $\gamma h'(z^+) = z - z^+$ and finish the proof. $\qquad\square$

*Proof of Theorem B.1.* By Lemma D.3 for convex function $h = r_i$, $z^+ = x_i^{k+1}$, $z = x_i^k - \gamma A_i^T y^{k+1/2}$ (see line 10 of Algorithm 8) and $x = x_i \in \mathbb{R}^{d_i}$, we get

$$\langle x_i^{k+1} - x_i^k + \gamma A_i^T y^{k+1/2}, x_i - x_i^{k+1} \rangle \geq \gamma \left( r_i(x_i^{k+1}) - r_i(x_i) \right),$$

and for $z^+ = x_i^{k+1/2}$, $z = x_i^k - \gamma A_i^T y^k$ (see line 10 of Algorithm 8), $x = x_i^{k+1}$,

$$\langle x_i^{k+1/2} - x_i^k + \gamma A_i^T y^k, x_i^{k+1} - x_i^{k+1/2} \rangle \geq \gamma \left( r_i(x_i^{k+1/2}) - r_i(x_i^{k+1}) \right).$$

Summing up two previous inequalities, we get

$$\langle x_i^{k+1} - x_i^k + \gamma A_i^T y^{k+1/2}, x_i - x_i^{k+1} \rangle + \langle x_i^{k+1/2} - x_i^k + \gamma A_i^T y^k, x_i^{k+1} - x_i^{k+1/2} \rangle$$
$$\geq \gamma \left( r_i(x_i^{k+1/2}) - r_i(x_i) \right).$$

After small rearrangements and multiplying by 2, we have

$$2\langle x_i^{k+1} - x_i^k, x_i - x_i^{k+1} \rangle + 2\langle x_i^{k+1/2} - x_i^k, x_i^{k+1} - x_i^{k+1/2} \rangle$$
$$+ 2\gamma\langle A_i^T y^{k+1/2}, x_i - x_i^{k+1/2} \rangle + 2\gamma\langle A_i^T (y^{k+1/2} - y^k), x_i^{k+1/2} - x_i^{k+1} \rangle$$
$$\geq 2\gamma \left( r_i(x_i^{k+1/2}) - r_i(x_i) \right).$$

For the first line we use identity $2\langle a, b \rangle = \|a + b\|^2 - \|a\|^2 - \|b\|^2$, and get

$$\left( \|x_i^k - x_i\|^2 - \|x_i^{k+1} - x_i\|^2 - \|x_i^{k+1} - x_i^k\|^2 \right)$$
$$+ \left( \|x_i^{k+1} - x_i^k\|^2 - \|x_i^{k+1/2} - x_i^k\|^2 - \|x_i^{k+1} - x_i^{k+1/2}\|^2 \right)$$
$$+ 2\gamma\langle A_i^T y^{k+1/2}, x_i - x_i^{k+1/2} \rangle + 2\gamma\langle A_i^T (y^{k+1/2} - y^k), x_i^{k+1/2} - x_i^{k+1} \rangle$$
$$\geq 2\gamma \left( r_i(x_i^{k+1/2}) - r_i(x_i) \right).$$

A small rearrangement gives

$$\|x_i^{k+1} - x_i\|^2 \leq \|x_i^k - x_i\|^2 - \|x_i^{k+1/2} - x_i^k\|^2 - \|x_i^{k+1} - x_i^{k+1/2}\|^2$$
$$+ 2\gamma\langle A_i^T y^{k+1/2}, x_i - x_i^{k+1/2} \rangle$$
$$+ 2\gamma\langle A_i^T (y^{k+1/2} - y^k), x_i^{k+1/2} - x_i^{k+1} \rangle - 2\gamma \left( r_i(x_i^{k+1/2}) - r_i(x_i) \right)$$
$$= \|x_i^k - x_i\|^2 - \|x_i^{k+1/2} - x_i^k\|^2 - \|x_i^{k+1} - x_i^{k+1/2}\|^2$$
$$+ 2\gamma\langle A_i(x_i - x_i^{k+1/2}), y^{k+1/2} \rangle$$
$$+ 2\gamma\langle A_i(x_i^{k+1/2} - x_i^{k+1}), y^{k+1/2} - y^k \rangle - 2\gamma \left( r_i(x_i^{k+1/2}) - r_i(x_i) \right).$$

Summing over all $i$ from 1 to $n$, we deduce

$$\sum_{i=1}^n \|x_i^{k+1} - x_i\|^2 \leq \sum_{i=1}^n \|x_i^k - x_i\|^2 - \sum_{i=1}^n \|x_i^{k+1/2} - x_i^k\|^2 - \sum_{i=1}^n \|x_i^{k+1} - x_i^{k+1/2}\|^2$$
$$+ 2\gamma\langle \sum_{i=1}^n A_i(x_i - x_i^{k+1/2}), y^{k+1/2} \rangle$$
$$+ 2\gamma\langle \sum_{i=1}^n A_i(x_i^{k+1/2} - x_i^{k+1}), y^{k+1/2} - y^k \rangle - 2\gamma \sum_{i=1}^n \left( r_i(x_i^{k+1/2}) - r_i(x_i) \right).$$

With notation of $A = [A_1, \ldots, A_i, \ldots, A_n]$ and notation of $x = [x_1^T, \ldots, x_i^T, \ldots, x_n^T]^T$ from (1) and (2), one can obtain that $\sum_{i=1}^n A_i x_i = Ax$:

$$\|x^{k+1} - x\|^2 \leq \|x^k - x\|^2 - \|x^{k+1/2} - x^k\|^2 - \|x^{k+1} - x^{k+1/2}\|^2$$
$$+ 2\gamma\langle A(x - x^{k+1/2}), y^{k+1/2} \rangle$$
$$+ 2\gamma\langle A(x^{k+1/2} - x^{k+1}), y^{k+1/2} - y^k \rangle - 2\gamma \sum_{i=1}^n \left( r_i(x_i^{k+1/2}) - r_i(x_i) \right)$$
$$= \|x^k - x\|^2 - \|x^{k+1/2} - x^k\|^2 - \|x^{k+1} - x^{k+1/2}\|^2$$

$$+ 2\gamma\langle A(x - x^{k+1/2}), y^{k+1/2}\rangle$$

$$+ 2\gamma\langle A^T(y^{k+1/2} - y^k), x^{k+1/2} - x^{k+1}\rangle - 2\gamma\sum_{i=1}^n \left(r_i(x_i^{k+1/2}) - r_i(x_i)\right).$$

By Cauchy Schwartz inequality: $2\langle a, b\rangle \leq \eta\|a\|^2 + \frac{1}{\eta}\|b\|^2$ with $a = A^T(y^{k+1/2} - y^k)$, $b = x^{k+1/2} - x^{k+1}$ and $\eta = \gamma$, we get

$$\begin{aligned}
\|x^{k+1} - x\|^2 \leq &\|x^k - x\|^2 - \|x^{k+1/2} - x^k\|^2 \\
&+ 2\gamma\langle A(x - x^{k+1/2}), y^{k+1/2}\rangle \\
&+ \gamma\|A^T(y^{k+1/2} - y^k)\|^2 - 2\gamma\sum_{i=1}^n \left(r_i(x_i^{k+1/2}) - r_i(x_i)\right).
\end{aligned} \tag{16}$$

Using the same steps, one can obtain for $z \in \mathbb{R}^s$,

$$\begin{aligned}
\|z^{k+1} - z\|^2 \leq &\|z^k - z\|^2 - \|z^{k+1/2} - z^k\|^2 \\
&- 2\gamma_z\langle y^{k+1/2}, z - z^{k+1/2}\rangle \\
&+ \gamma_z^2\|y^{k+1/2} - y^k\|^2 - 2\gamma\left(\ell(z^{k+1/2}, b) - \ell(z, b)\right),
\end{aligned} \tag{17}$$

and for all $y \in \mathbb{R}^s$,

$$\begin{aligned}
\|y^{k+1} - y\|^2 \leq &\|y^k - y\|^2 - \|y^{k+1/2} - y^k\|^2 \\
&- 2\gamma\langle\sum_{i=1}^n A_i x_i^{k+1/2} - z^{k+1/2}, y - y^{k+1/2}\rangle \\
&+ \gamma^2\left\|\sum_{i=1}^n A_i(x_i^{k+1/2} - x_i^k) - (z^{k+1/2} - z^k)\right\|^2 \\
= &\|y^k - y\|^2 - \|y^{k+1/2} - y^k\|^2 \\
&- 2\gamma\langle Ax^{k+1/2} - z^{k+1/2}, y - y^{k+1/2}\rangle \\
&+ \gamma^2\|A(x^{k+1/2} - x^k) - (z^{k+1/2} - z^k)\|^2.
\end{aligned} \tag{18}$$

Here we also use notation of $A$ and $x$. Summing up (16), (17) and (18), we obtain

$$\begin{aligned}
\|x^{k+1} - x\|^2 &+ \|z^{k+1} - z\|^2 + \|y^{k+1} - y\|^2 \\
\leq &\|x^k - x\|^2 + \|z^k - z\|^2 + \|y^k - y\|^2 \\
&- \|x^{k+1/2} - x^k\|^2 - \|z^{k+1/2} - z^k\|^2 - \|y^{k+1/2} - y^k\|^2 \\
&+ 2\gamma\langle A(x - x^{k+1/2}), y^{k+1/2}\rangle - 2\gamma\langle y^{k+1/2}, z - z^{k+1/2}\rangle \\
&- 2\gamma\langle Ax^{k+1/2} - z^{k+1/2}, y - y^{k+1/2}\rangle \\
&+ \gamma^2\|A^T(y^{k+1/2} - y^k)\|^2 + \gamma^2\|y^{k+1/2} - y^k\|^2 \\
&+ \gamma^2\left\|A(x^{k+1/2} - x^k) - (z^{k+1/2} - z^k)\right\|^2 \\
&- 2\gamma\left(\ell(z^{k+1/2}, b) - \ell(z, b)\right) - 2\gamma\sum_{i=1}^n \left(r_i(x_i^{k+1/2}) - r_i(x_i)\right).
\end{aligned}$$

Again by Cauchy Schwartz inequality: $\|a - b\|^2 \leq 2\|a\|^2 + 2\|b\|^2$ with $a = A(x^{k+1/2} - x^k)$, $b = \gamma(z^{k+1/2} - z^k)$, we get

$$\begin{aligned}
\|x^{k+1} - x\|^2 &+ \|z^{k+1} - z\|^2 + \|y^{k+1} - y\|^2 \\
\leq &\|x^k - x\|^2 + \|z^k - z\|^2 + \|y^k - y\|^2 \\
&- \|x^{k+1/2} - x^k\|^2 - \|z^{k+1/2} - z^k\|^2 - \|y^{k+1/2} - y^k\|^2 \\
&+ 2\gamma\langle A(x - x^{k+1/2}), y^{k+1/2}\rangle - 2\gamma\langle y^{k+1/2}, z - z^{k+1/2}\rangle
\end{aligned}$$

$$- 2\gamma \langle Ax^{k+1/2} - z^{k+1/2}, y - y^{k+1/2} \rangle$$

$$+ \gamma^2 \|A^T(y^{k+1/2} - y^k)\|^2 + \gamma^2 \|y^{k+1/2} - y^k\|^2$$

$$+ 2\gamma^2 \left\|A(x^{k+1/2} - x^k)\right\|^2 + 2\gamma^2 \left\|z^{k+1/2} - z^k\right\|^2$$

$$- 2\gamma \left(\ell(z^{k+1/2}, b) - \ell(z, b)\right) - 2\gamma \sum_{i=1}^{n}\left(r_i(x_i^{k+1/2}) - r_i(x_i)\right).$$

Using the definition of $\lambda_{\max}(\cdot)$ as a maximum eigenvalue, we get

$$\|x^{k+1} - x\|^2 + \|z^{k+1} - z\|^2 + \|y^{k+1} - y\|^2$$

$$\leq \|x^k - x\|^2 + \|z^k - z\|^2 + \|y^k - y\|^2$$

$$- \|x^{k+1/2} - x^k\|^2 - \|z^{k+1/2} - z^k\|^2 - \|y^{k+1/2} - y^k\|^2$$

$$+ 2\langle \tilde{A}(x - x^{k+1/2}), y^{k+1/2}\rangle - 2\gamma\langle y^{k+1/2}, z - z^{k+1/2}\rangle$$

$$- 2\gamma\langle Ax^{k+1/2} - z^{k+1/2}, y - y^{k+1/2}\rangle$$

$$+ \lambda_{\max}(AA^T)\|y^{k+1/2} - y^k\|^2 + \gamma^2\|y^{k+1/2} - y^k\|^2$$

$$+ 2\lambda_{\max}(A^T A)\gamma^2 \left\|x^{k+1/2} - x^k\right\|^2 + 2\gamma^2 \left\|z^{k+1/2} - z^k\right\|^2$$

$$- 2\gamma \left(\ell(z^{k+1/2}, b) - \ell(z, b)\right) - 2\gamma \sum_{i=1}^{n}\left(r_i(x_i^{k+1/2}) - r_i(x_i)\right).$$

With the choice of $\gamma \leq \frac{1}{\sqrt{2}} \cdot \min\left\{1; \frac{1}{\sqrt{\lambda_{\max}(A^T A)}}\right\}$, we get

$$\|x^{k+1} - x\|^2 + \|z^{k+1} - z\|^2 + \|y^{k+1} - y\|^2$$

$$\leq \|x^k - x\|^2 + \|z^k - z\|^2 + \|y^k - y\|^2$$

$$+ 2\gamma\langle A(x - x^{k+1/2}), y^{k+1/2}\rangle - 2\gamma\langle y^{k+1/2}, z - z^{k+1/2}\rangle$$

$$- 2\gamma\langle Ax^{k+1/2} - z^{k+1/2}, y - y^{k+1/2}\rangle$$

$$- 2\gamma \left(\ell(z^{k+1/2}, b) - \ell(z, b)\right) - 2\gamma \sum_{i=1}^{n}\left(r_i(x_i^{k+1/2}) - r_i(x_i)\right)$$

$$= \|x^k - x\|^2 + \|z^k - z\|^2 + \|y^k - y\|^2$$

$$+ 2\gamma\langle Ax - z, y^{k+1/2}\rangle - 2\gamma\langle Ax^{k+1/2} - z^{k+1/2}, y\rangle$$

$$- 2\gamma \left(\ell(z^{k+1/2}, b) - \ell(z, b)\right) - 2\gamma \sum_{i=1}^{n}\left(r_i(x_i^{k+1/2}) - r_i(x_i)\right).$$

After small rearrangements, we obtain

$$\left(\ell(z^{k+1/2}, b) - \ell(z, b)\right) + \sum_{i=1}^{n}\left(r_i(x_i^{k+1/2}) - r_i(x_i)\right)$$

$$+ \langle Ax^{k+1/2} - z^{k+1/2}, y\rangle - \langle Ax - z, y^{k+1/2}\rangle$$

$$\leq \frac{1}{2\gamma}\left(\|x^k - x\|^2 + \|z^k - z\|^2 + \|y^k - y\|^2\right.$$

$$\left. - \|x^{k+1} - x\|^2 - \|z^{k+1} - z\|^2 - \|y^{k+1} - y\|^2\right).$$

Then we sum all over $k$ from 0 to $K - 1$, divide by $K$, and have

$$\frac{1}{K}\sum_{k=0}^{K-1}\left(\ell(z^{k+1/2}, b) - \ell(z, b)\right) + \sum_{i=1}^{n}\frac{1}{K}\sum_{k=0}^{K-1}\left(r_i(x_i^{k+1/2}) - r_i(x_i)\right)$$

$$+ \left\langle A \cdot \frac{1}{K}\sum_{k=0}^{K-1}x^{k+1/2} - \frac{1}{K}\sum_{k=0}^{K-1}z^{k+1/2}, y\right\rangle - \left\langle Ax - z, \frac{1}{K}\sum_{k=0}^{K-1}y^{k+1/2}\right\rangle$$

$$\leq \frac{1}{2\gamma K}\Big(\|x^0 - x\|^2 + \|z^0 - z\|^2 + \|y^0 - y\|^2$$

$$- \|x^K - x\|^2 - \|z^K - z\|^2 - \|y^K - y\|^2\Big)$$

$$\leq \frac{1}{2\gamma K}(\|x^0 - x\|^2 + \|z^0 - z\|^2 + \|y^0 - y\|^2).$$

With Jensen inequality for convex functions $\ell$ and $r_i$, one can note that

$$\ell\left(\frac{1}{K}\sum_{k=0}^{K-1} z^{k+1/2}, b\right) \leq \frac{1}{K}\sum_{k=0}^{K-1} \ell(z^{k+1/2}, b),$$

$$r_i\left(\frac{1}{K}\sum_{k=0}^{K-1} x_i^{k+1/2}\right) \leq \frac{1}{K}\sum_{k=0}^{K-1} r_i(x_i^{k+1/2}).$$

Then, with notation $\bar{x}_i^K = \frac{1}{K}\sum_{k=0}^{K-1} x_i^{k+1/2}$, $\bar{z}^K = \frac{1}{K}\sum_{k=0}^{K-1} z^{k+1/2}$, $\bar{y}^K = \frac{1}{K}\sum_{k=0}^{K-1} y^{k+1/2}$, we have

$$\ell(\bar{z}^K, b) - \ell(z, b) + \sum_{i=1}^n \left(r_i(\bar{x}_i^K) - r_i(x_i)\right) + \langle A\bar{x}^K - \bar{z}^K, y\rangle - \langle Ax - z, \bar{y}^K\rangle$$

$$\leq \frac{1}{2\gamma K}(\|x^0 - x\|^2 + \|z^0 - z\|^2 + \|y^0 - y\|^2).$$

To complete the proof of the theorem, it is sufficient to do the same steps as when obtaining (15). Finally, we need to put $\gamma = \frac{1}{\sqrt{2}} \cdot \min\left\{1; \frac{1}{\sqrt{\lambda_{\max}(A^T A)}}\right\}$. $\qquad\square$

### D.3 PROOF OF THEOREM 3.2

**Theorem D.4** (Theorem 3.2). *Let Assumption 2.1 holds. Let problem (4) be solved by Algorithm 2 with operators and Q, which satisfy Definition 3.1. Then for $\tau = 1 - p$ and*

$$\gamma = \frac{1}{4}\min\left\{1; \frac{1}{L_r}; \frac{1}{L_\ell}; \sqrt{\frac{1-\tau}{\omega[\lambda_{\max}(AA^T) + \mathbb{I}(\textit{diff. seed})\max_i\{\lambda_{\max}(A_i A_i^T)\}]}}; \sqrt{\frac{1-\tau}{\omega\lambda_{\max}(AA^T)}};\right\},$$

*it holds that*

$$\mathbb{E}gap(\bar{x}^K, \bar{z}^K, \bar{y}^K) = \mathcal{O}\Big([1 + \sqrt{\tfrac{\omega}{p}}(\sqrt{\lambda_{\max}(AA^T)}) + L_\ell + L_r]\cdot\frac{D^2}{K}$$

$$+ \sqrt{\tfrac{\omega}{p}}\mathbb{I}(\textit{diff. seed})\max_{i=1,\dots,n}\{\sqrt{\lambda_{\max}(A_i A_i^T)}\}\cdot\frac{D^2}{K}\Big),$$

*where the indicator function $\mathbb{I}(\textit{diff. seed})$ is responsible for whether the different or same random seed is used on all devices, $\bar{x}^K := \frac{1}{K}\sum_{k=0}^{K-1} x^{k+1/2}$, $\bar{z}^K := \frac{1}{K}\sum_{k=0}^{K-1} z^{k+1/2}$, $\bar{y}^K := \frac{1}{K}\sum_{k=0}^{K-1} y^{k+1/2}$ and $D^2 := \max_{x,z,y\in\mathcal{X},\mathcal{Z},\mathcal{Y}}\left[\|x^0 - x\|^2 + \|z^0 - z\|^2 + \|y^0 - y\|^2\right]$.*

*Proof.* We start the proof with the following equations on the variables $x_i^{k+1}$, $x_i^{k+1/2}$, $x_i^k$ and any $x_i \in \mathbb{R}^{d_i}$:

$$\|x_i^{k+1} - x_i\|^2 = \|x_i^k - x_i\|^2 + 2\langle x_i^{k+1} - x_i^k, x_i^{k+1} - x_i\rangle - \|x_i^{k+1} - x_i^k\|^2,$$

$$\|x_i^{k+1/2} - x_i^{k+1}\|^2 = \|x_i^k - x_i^{k+1}\|^2 + 2\langle x_i^{k+1/2} - x_i^k, x_i^{k+1/2} - x_i^{k+1}\rangle - \|x_i^{k+1/2} - x_i^k\|^2.$$

Summing up two previous inequalities and making small rearrangements, we get

$$\|x_i^{k+1} - x_i\|^2 = \|x_i^k - x_i\|^2 - \|x_i^{k+1/2} - x_i^k\|^2 - \|x_i^{k+1/2} - x_i^{k+1}\|^2$$

$$+ 2\langle x_i^{k+1} - x_i^k, x_i^{k+1} - x_i\rangle + 2\langle x_i^{k+1/2} - x_i^k, x_i^{k+1/2} - x_i^{k+1}\rangle.$$

Using that $x_i^{k+1} - x_i^k = (1-\tau)(w_i^k - x_i^k) - \gamma(A_i^T[Q(y^{k+1/2} - u^k) + u^k] + \nabla r_i(x_i^{k+1/2}))$ and $x_i^{k+1/2} - x_i^k = (1-\tau)(w_i^k - x_i^k) - \gamma(A_i^T u^k + \nabla r_i(x_i^k))$, we obtain

$$
\begin{aligned}
\|x_i^{k+1} - x_i\|^2 =& \|x_i^k - x_i\|^2 - \|x_i^{k+1/2} - x_i^k\|^2 - \|x_i^{k+1/2} - x_i^{k+1}\|^2 \\
& + 2(1-\tau)\langle w_i^k - x_i^k, x_i^{k+1} - x_i\rangle \\
& - 2\gamma\langle A_i^T[Q(y^{k+1/2} - u^k) + u^k] + \nabla r_i(x_i^{k+1/2}), x_i^{k+1} - x_i\rangle \\
& + 2(1-\tau)\langle w_i^k - x_i^k, x_i^{k+1/2} - x_i^{k+1}\rangle \\
& - 2\gamma\langle A_i^T u^k + \nabla r_i(x_i^k), x_i^{k+1/2} - x_i^{k+1}\rangle \\
=& \|x_i^k - x_i\|^2 - \|x_i^{k+1/2} - x_i^k\|^2 - \|x_i^{k+1/2} - x_i^{k+1}\|^2 \\
& + 2(1-\tau)\langle w_i^k - x_i^k, x_i^{k+1/2} - x_i\rangle \\
& - 2\gamma\langle A_i^T[Q(y^{k+1/2} - u^k) + u^k] + \nabla r_i(x_i^{k+1/2}), x_i^{k+1/2} - x_i\rangle \\
& + 2\gamma\langle A_i^T[Q(y^{k+1/2} - u^k)] + \nabla r_i(x_i^{k+1/2}) - \nabla r_i(x_i^k), x_i^{k+1/2} - x_i^{k+1}\rangle \\
=& \|x_i^k - x_i\|^2 - \|x_i^{k+1/2} - x_i^k\|^2 - \|x_i^{k+1/2} - x_i^{k+1}\|^2 \\
& + 2(1-\tau)\langle w_i^k - x_i^{k+1/2}, x_i^{k+1/2} - x_i\rangle \\
& + 2(1-\tau)\langle x_i^{k+1/2} - x_i^k, x_i^{k+1/2} - x_i\rangle \\
& - 2\gamma\langle A_i^T[Q(y^{k+1/2} - u^k) + u^k] + \nabla r_i(x_i^{k+1/2}), x_i^{k+1/2} - x_i\rangle \\
& + 2\gamma\langle A_i^T[Q(y^{k+1/2} - u^k)] + \nabla r_i(x_i^{k+1/2}) - \nabla r_i(x_i^k), x_i^{k+1/2} - x_i^{k+1}\rangle.
\end{aligned}
$$

For the second and third lines we use identity $2\langle a, b\rangle = \|a + b\|^2 - \|a\|^2 - \|b\|^2$, and get

$$
\begin{aligned}
\|x_i^{k+1} - x_i\|^2 =& \|x_i^k - x_i\|^2 - \|x_i^{k+1/2} - x_i^k\|^2 - \|x_i^{k+1/2} - x_i^{k+1}\|^2 \\
& + (1-\tau)(\|w_i^k - x_i\|^2 - \|w_i^k - x_i^{k+1/2}\|^2 - \|x_i^{k+1/2} - x_i\|^2) \\
& + (1-\tau)(\|x_i^{k+1/2} - x_i^k\|^2 + \|x_i^{k+1/2} - x_i\|^2 - \|x_i^k - x_i\|^2) \\
& - 2\gamma\langle A_i^T[Q(y^{k+1/2} - u^k) + u^k] + \nabla r_i(x_i^{k+1/2}), x_i^{k+1/2} - x_i\rangle \\
& + 2\gamma\langle A_i^T[Q(y^{k+1/2} - u^k)] + \nabla r_i(x_i^{k+1/2}) - \nabla r_i(x_i^k), x_i^{k+1/2} - x_i^{k+1}\rangle \quad (19) \\
=& \tau\|x_i^k - x_i\|^2 + (1-\tau)\|w_i^k - x_i\|^2 \\
& - \tau\|x_i^{k+1/2} - x_i^k\|^2 - (1-\tau)\|w_i^k - x_i^{k+1/2}\|^2 - \|x_i^{k+1/2} - x_i^{k+1}\|^2 \\
& - 2\gamma\langle A_i(x_i^{k+1/2} - x_i), y^{k+1/2}\rangle - 2\gamma\langle \nabla r_i(x_i^{k+1/2}), x_i^{k+1/2} - x_i\rangle \\
& - 2\gamma\langle A_i(x_i^{k+1/2} - x_i), Q(y^{k+1/2} - u^k) - y^{k+1/2} + u^k\rangle \\
& + 2\gamma\langle A_i(x_i^{k+1/2} - x_i^{k+1}), Q(y^{k+1/2} - u^k)\rangle \\
& + 2\gamma\langle \nabla r_i(x_i^{k+1/2}) - \nabla r_i(x_i^k), x_i^{k+1/2} - x_i^{k+1}\rangle.
\end{aligned}
$$

Summing over all $i$ from 1 to $n$ and using the notation of $A = [A_1, \ldots, A_i, \ldots, A_n]$, $x = [x_1^T, \ldots, x_i^T, \ldots, x_n^T]^T$, $w = [w_1^T, \ldots, w_i^T, \ldots, w_n^T]^T$, we deduce

$$
\begin{aligned}
\|x^{k+1} - x\|^2 =& \tau\|x^k - x\|^2 + (1-\tau)\|w^k - x\|^2 \\
& - \tau\|x^{k+1/2} - x^k\|^2 - (1-\tau)\|w^k - x^{k+1/2}\|^2 - \|x^{k+1/2} - x^{k+1}\|^2 \\
& - 2\gamma\langle A(x^{k+1/2} - x), y^{k+1/2}\rangle - 2\gamma\sum_{i=1}^n \langle \nabla r_i(x_i^{k+1/2}), x_i^{k+1/2} - x_i\rangle \\
& - 2\gamma\langle A(x^{k+1/2} - x), Q(y^{k+1/2} - u^k) - y^{k+1/2} + u^k\rangle \\
& + 2\gamma\langle A(x^{k+1/2} - x^{k+1}), Q(y^{k+1/2} - u^k)\rangle \\
& + 2\gamma\sum_{i=1}^n \langle \nabla r_i(x_i^{k+1/2}) - \nabla r_i(x_i^k), x_i^{k+1/2} - x_i^{k+1}\rangle
\end{aligned}
$$

$$
\begin{aligned}
=&\tau\|x^k - x\|^2 + (1-\tau)\|w^k - x\|^2 \\
&- \tau\|x^{k+1/2} - x^k\|^2 - (1-\tau)\|w^k - x^{k+1/2}\|^2 - \|x^{k+1/2} - x^{k+1}\|^2 \\
&- 2\gamma\langle A(x^{k+1/2} - x), y^{k+1/2}\rangle - 2\gamma\sum_{i=1}^{n}\langle\nabla r_i(x_i^{k+1/2}), x_i^{k+1/2} - x_i\rangle \\
&- 2\gamma\langle A(x^{k+1/2} - x), Q(y^{k+1/2} - u^k) - y^{k+1/2} + u^k\rangle \\
&+ 2\gamma\langle A^T Q(y^{k+1/2} - u^k), x^{k+1/2} - x^{k+1}\rangle \\
&+ 2\gamma\sum_{i=1}^{n}\langle\nabla r_i(x_i^{k+1/2}) - \nabla r_i(x_i^k), x_i^{k+1/2} - x_i^{k+1}\rangle.
\end{aligned}
$$

By simple fact: $2\langle a, b\rangle \le \eta\|a\|^2 + \frac{1}{\eta}\|b\|^2$ with $a = A^T Q(y^{k+1/2} - u^k)$, $b = x^{k+1/2} - x^{k+1}$, $\eta = 2\gamma$ and $a = \nabla r_i(x_i^{k+1/2}) - \nabla r_i(x_i^k)$, $b = x_i^{k+1/2} - x_i^{k+1}$, $\eta = 2\gamma$, we get

$$
\begin{aligned}
\|x^{k+1} - x\|^2 \le &\tau\|x^k - x\|^2 + (1-\tau)\|w^k - x\|^2 \\
&- \tau\|x^{k+1/2} - x^k\|^2 - (1-\tau)\|w^k - x^{k+1/2}\|^2 - \|x^{k+1/2} - x^{k+1}\|^2 \\
&- 2\gamma\langle A(x^{k+1/2} - x), y^{k+1/2}\rangle - 2\gamma\sum_{i=1}^{n}\langle\nabla r_i(x_i^{k+1/2}), x_i^{k+1/2} - x_i\rangle \\
&- 2\gamma\langle A(x^{k+1/2} - x), Q(y^{k+1/2} - u^k) - y^{k+1/2} + u^k\rangle \\
&+ 2\gamma^2\|A^T Q(y^{k+1/2} - u^k)\|^2 + \frac{1}{2}\|x^{k+1/2} - x^{k+1}\|^2 \\
&+ 2\gamma^2\sum_{i=1}^{n}\|\nabla r_i(x_i^{k+1/2}) - \nabla r_i(x_i^k)\|^2 + \frac{1}{2}\|x^{k+1/2} - x^{k+1}\|^2.
\end{aligned}
$$

Adding to the both sides $\|w^{k+1} - x\|^2$, one can obtain

$$
\begin{aligned}
\|x^{k+1} - x\|^2 &+ \|w^{k+1} - x\|^2 \\
\le &\|x^k - x\|^2 + \|w^k - x\|^2 \\
&- (1-\tau)\|x^k - x\|^2 - \tau\|w^k - x\|^2 + \|w^{k+1} - x\|^2 \\
&- \tau\|x^{k+1/2} - x^k\|^2 - (1-\tau)\|w^k - x^{k+1/2}\|^2 \\
&- 2\gamma\langle A(x^{k+1/2} - x), y^{k+1/2}\rangle - 2\gamma\sum_{i=1}^{n}\langle\nabla r_i(x_i^{k+1/2}), x_i^{k+1/2} - x_i\rangle \\
&- 2\gamma\langle A(x^{k+1/2} - x), Q(y^{k+1/2} - u^k) - y^{k+1/2} + u^k\rangle \\
&+ 2\gamma^2\|A^T Q(y^{k+1/2} - u^k)\|^2 \\
&+ 2\gamma^2\sum_{i=1}^{n}\|\nabla r_i(x_i^{k+1/2}) - \nabla r_i(w_i^k)\|^2 \\
= &\|x^k - x\|^2 + \|w^k - x\|^2 \\
&- \tau\|x^{k+1/2} - x^k\|^2 - (1-\tau)\|w^k - x^{k+1/2}\|^2 \\
&- 2\gamma\langle A(x^{k+1/2} - x), y^{k+1/2}\rangle - 2\gamma\sum_{i=1}^{n}\langle\nabla r_i(x_i^{k+1/2}), x_i^{k+1/2} - x_i\rangle \\
&- (1-\tau)\|x^k\|^2 - \tau\|w^k\|^2 + \|w^{k+1}\|^2 \\
&+ 2\langle(1-\tau)x^k + \tau w^k - w^{k+1}, x\rangle \\
&- 2\gamma\langle A(x^{k+1/2} - x^0), Q(y^{k+1/2} - u^k) - y^{k+1/2} + u^k\rangle \\
&- 2\gamma\langle A(x^0 - x), Q(y^{k+1/2} - u^k) - y^{k+1/2} + u^k\rangle \\
&+ 2\gamma^2\|A^T Q(y^{k+1/2} - u^k)\|^2
\end{aligned}
$$

$$+ 2\gamma^2 \sum_{i=1}^{n} \|\nabla r_i(x_i^{k+1/2}) - \nabla r_i(x_i^k)\|^2. \tag{20}$$

Using the same steps, one can obtain for $z \in \mathbb{R}^s$,

$$\begin{aligned}
\|z^{k+1} - z\|^2 \leq &\|z^k - z\|^2 - \|z^{k+1/2} - z^k\|^2 \\
&+ 2\gamma\langle y^{k+1/2}, z^{k+1/2} - z\rangle - 2\gamma\langle \nabla\ell(z^{k+1/2}, b), z^{k+1/2} - z\rangle \\
&+ 2\gamma^2\|y^{k+1/2} - y^k\|^2 + 2\gamma^2\|\nabla\ell(z^{k+1/2}, b) - \nabla\ell(z^k, b)\|^2. \tag{21}
\end{aligned}$$

and for all $y \in \mathbb{R}^s$,

$$\begin{aligned}
\|y^{k+1} - y\|^2 &+ \|u^{k+1} - y\|^2 \\
&\leq \|y^k - y\|^2 + \|u^k - y\|^2 \\
&\quad - \tau\|y^{k+1/2} - y^k\|^2 - (1-\tau)\|u^k - y^{k+1/2}\|^2 \\
&\quad - 2\gamma\langle z^{k+1/2}, y^{k+1/2} - y\rangle + 2\gamma\langle \sum_{i=1}^{n} A_i x_i^{k+1/2}, y^{k+1/2} - y\rangle \\
&\quad + 2\gamma\langle \sum_{i=1}^{n} [Q(A_i x_i^{k+1/2} - A_i w_i^k) + A_i w_i^k - A_i x_i^{k+1/2}], y^{k+1/2} - y^0\rangle \\
&\quad + 2\gamma\langle \sum_{i=1}^{n} [Q(A_i x_i^{k+1/2} - A_i w_i^k) + A_i w_i^k - A_i x_i^{k+1/2}], y^0 - y\rangle \\
&\quad - (1-\tau)\|y^k\|^2 - \tau\|u^k\|^2 + \|y^{k+1}\|^2 \\
&\quad + 2\langle (1-\tau)y^k + \tau u^k - u^{k+1}, y\rangle \\
&\quad + 2\gamma^2\|\sum_{i=1}^{n} Q(A_i x_i^{k+1/2} - A_i w_i^k)\|^2 + 2\gamma^2\|z^{k+1/2} - z^k\|^2. \tag{22}
\end{aligned}$$

Summing up (20), (21) and (22), we obtain

$$\begin{aligned}
\|x^{k+1} - x\|^2 &+ \|w^{k+1} - x\|^2 + \|z^{k+1} - z\|^2 + \|y^{k+1} - y\|^2 + \|u^{k+1} - y\|^2 \\
&\leq \|x^k - x\|^2 + \|w^k - x\|^2 + \|z^k - z\|^2 + \|y^k - y\|^2 + \|u^k - y\|^2 \\
&\quad - \tau\|x^{k+1/2} - x^k\|^2 - (1-\tau)\|w^k - x^{k+1/2}\|^2 - \|z^{k+1/2} - z^k\|^2 \\
&\quad - \tau\|y^{k+1/2} - y^k\|^2 - (1-\tau)\|u^k - y^{k+1/2}\|^2 \\
&\quad - 2\gamma\langle A(x^{k+1/2} - x), y^{k+1/2}\rangle + 2\gamma\langle y^{k+1/2}, z^{k+1/2} - z\rangle \\
&\quad - 2\gamma\langle z^{k+1/2}, y^{k+1/2} - y\rangle + 2\gamma\langle \sum_{i=1}^{n} A_i x_i^{k+1/2}, y^{k+1/2} - y\rangle \\
&\quad - 2\gamma\sum_{i=1}^{n}\langle \nabla r_i(x_i^{k+1/2}), x_i^{k+1/2} - x_i\rangle - 2\gamma\langle \nabla\ell(z^{k+1/2}, b), z^{k+1/2} - z\rangle \\
&\quad - (1-\tau)\|x^k\|^2 - \tau\|w^k\|^2 + \|w^{k+1}\|^2 \\
&\quad + 2\langle (1-\tau)x^k + \tau w^k - w^{k+1}, x\rangle \\
&\quad - (1-\tau)\|y^k\|^2 - \tau\|u^k\|^2 + \|y^{k+1}\|^2 \\
&\quad + 2\langle (1-\tau)y^k + \tau u^k - u^{k+1}, y\rangle \\
&\quad - 2\gamma\langle A(x^{k+1/2} - x^0), Q(y^{k+1/2} - u^k) - y^{k+1/2} + u^k\rangle \\
&\quad - 2\gamma\langle A(x^0 - x), Q(y^{k+1/2} - u^k) - y^{k+1/2} + u^k\rangle \\
&\quad + 2\gamma\langle \sum_{i=1}^{n} [Q(A_i x_i^{k+1/2} - A_i w_i^k) + A_i w_i^k - A_i x_i^{k+1/2}], y^{k+1/2} - y^0\rangle \\
&\quad + 2\gamma\langle \sum_{i=1}^{n} [Q(A_i x_i^{k+1/2} - A_i w_i^k) + A_i w_i^k - A_i x_i^{k+1/2}], y^0 - y\rangle
\end{aligned}$$

$$+ 2\gamma^2 \|A^T Q(y^{k+1/2} - u^k)\|^2 + 2\gamma^2 \|\sum_{i=1}^n Q(A_i x_i^{k+1/2} - A_i w_i^k)\|^2$$

$$+ 2\gamma^2 \sum_{i=1}^n \|\nabla r_i(x_i^{k+1/2}) - \nabla r_i(x_i^k)\|^2 + 2\gamma^2 \|y^{k+1/2} - y^k\|^2$$

$$+ 2\gamma^2 \|z^{k+1/2} - z^k\|^2 + 2\gamma^2 \|\nabla\ell(z^{k+1/2}, b) - \nabla\ell(z^k, b)\|^2.$$

After small rearrangements, we have

$$2\gamma\Big[\langle\nabla\ell(z^{k+1/2}, b), z^{k+1/2} - z\rangle + \sum_{i=1}^n \langle\nabla r_i(x_i^{k+1/2}), x_i^{k+1/2} - x_i\rangle$$

$$+ \langle Ax^{k+1/2} - z^{k+1/2}, y\rangle - \langle Ax - z, y^{k+1/2}\rangle\Big]$$

$$\leq \|x^k - x\|^2 + \|w^k - x\|^2 + \|z^k - z\|^2 + \|y^k - y\|^2 + \|u^k - y\|^2$$

$$- \left(\|x^{k+1} - x\|^2 + \|w^{k+1} - x\|^2 + \|z^{k+1} - z\|^2 + \|y^{k+1} - y\|^2 + \|u^{k+1} - y\|^2\right)$$

$$- \tau\|x^{k+1/2} - x^k\|^2 - (1-\tau)\|w^k - x^{k+1/2}\|^2 - \|z^{k+1/2} - z^k\|^2$$

$$- \tau\|y^{k+1/2} - y^k\|^2 - (1-\tau)\|u^k - y^{k+1/2}\|^2$$

$$- (1-\tau)\|x^k\|^2 - \tau\|w^k\|^2 + \|w^{k+1}\|^2$$

$$+ 2\langle(1-\tau)x^k + \tau w^k - w^{k+1}, x\rangle$$

$$- (1-\tau)\|y^k\|^2 - \tau\|u^k\|^2 + \|y^{k+1}\|^2$$

$$+ 2\langle(1-\tau)y^k + \tau u^k - u^{k+1}, y\rangle$$

$$- 2\gamma\langle A(x^{k+1/2} - x^0), Q(y^{k+1/2} - u^k) - y^{k+1/2} + u^k\rangle$$

$$- 2\gamma\langle A(x^0 - x), Q(y^{k+1/2} - u^k) - y^{k+1/2} + u^k\rangle$$

$$+ 2\gamma\langle\sum_{i=1}^n [Q(A_i x_i^{k+1/2} - A_i w_i^k) + A_i w_i^k - A_i x_i^{k+1/2}], y^{k+1/2} - y^0\rangle$$

$$+ 2\gamma\langle\sum_{i=1}^n [Q(A_i x_i^{k+1/2} - A_i w_i^k) + A_i w_i^k - A_i x_i^{k+1/2}], y^0 - y\rangle$$

$$+ 2\gamma^2 \|A^T Q(y^{k+1/2} - u^k)\|^2 + 2\gamma^2 \|\sum_{i=1}^n Q(A_i x_i^{k+1/2} - A_i w_i^k)\|^2$$

$$+ 2\gamma^2 \sum_{i=1}^n \|\nabla r_i(x_i^{k+1/2}) - \nabla r_i(x_i^k)\|^2 + 2\gamma^2 \|y^{k+1/2} - y^k\|^2$$

$$+ 2\gamma^2 \|z^{k+1/2} - z^k\|^2 + 2\gamma^2 \|\nabla\ell(z^{k+1/2}, b) - \nabla\ell(z^k, b)\|^2.$$

Using convexity and $L_r$-smoothness of the function $r_i$ with convexity and $L_\ell$-smoothness of the function $\ell$ (Assumption 2.1), we have

$$2\gamma\Big[\ell(z^{k+1/2}, b) - \ell(z, b) + \sum_{i=1}^n r_i(x_i^{k+1/2}) - r_i(x_i)$$

$$+ \langle Ax^{k+1/2} - z^{k+1/2}, y\rangle - \langle Ax - z, y^{k+1/2}\rangle\Big]$$

$$\leq \|x^k - x\|^2 + \|w^k - x\|^2 + \|z^k - z\|^2 + \|y^k - y\|^2 + \|u^k - y\|^2$$

$$- \left(\|x^{k+1} - x\|^2 + \|w^{k+1} - x\|^2 + \|z^{k+1} - z\|^2 + \|y^{k+1} - y\|^2 + \|u^{k+1} - y\|^2\right)$$

$$- \tau\|x^{k+1/2} - x^k\|^2 - (1-\tau)\|w^k - x^{k+1/2}\|^2 - \|z^{k+1/2} - z^k\|^2$$

$$- \tau\|y^{k+1/2} - y^k\|^2 - (1-\tau)\|u^k - y^{k+1/2}\|^2$$

$$- (1-\tau)\|x^k\|^2 - \tau\|w^k\|^2 + \|w^{k+1}\|^2$$

$$+ 2\langle (1-\tau)x^k + \tau w^k - w^{k+1}, x\rangle$$

$$- (1-\tau)\|y^k\|^2 - \tau\|u^k\|^2 + \|y^{k+1}\|^2$$

$$+ 2\langle (1-\tau)y^k + \tau u^k - u^{k+1}, y\rangle$$

$$- 2\gamma\langle A(x^{k+1/2} - x^0), Q(y^{k+1/2} - u^k) - y^{k+1/2} + u^k\rangle$$

$$- 2\gamma\langle A(x^0 - x), Q(y^{k+1/2} - u^k) - y^{k+1/2} + u^k\rangle$$

$$+ 2\gamma\langle \sum_{i=1}^n [Q(A_i x_i^{k+1/2} - A_i w_i^k) + A_i w_i^k - A_i x_i^{k+1/2}], y^{k+1/2} - y^0\rangle$$

$$+ 2\gamma\langle \sum_{i=1}^n [Q(A_i x_i^{k+1/2} - A_i w_i^k) + A_i w_i^k - A_i x_i^{k+1/2}], y^0 - y\rangle$$

$$+ 2\gamma^2\|A^T Q(y^{k+1/2} - u^k)\|^2 + 2\gamma^2\|\sum_{i=1}^n Q(A_i x_i^{k+1/2} - A_i w_i^k)\|^2$$

$$+ 2\gamma^2 L_r^2\|x^{k+1/2} - x^k\|^2 + 2\gamma^2\|y^{k+1/2} - y^k\|^2$$

$$+ 2\gamma^2\|z^{k+1/2} - z^k\|^2 + 2\gamma^2 L_\ell^2\|z^{k+1/2} - z^k\|^2.$$

Then we sum all over $k$ from 0 to $K-1$, divide by $K$, use Jensen inequality for convex functions $\ell$ and $r_i$ with notation $\bar{x}_i^K = \frac{1}{K}\sum_{k=0}^{K-1} x_i^{k+1/2}$, $\bar{z}^K = \frac{1}{K}\sum_{k=0}^{K-1} z^{k+1/2}$, $\bar{y}^K = \frac{1}{K}\sum_{k=0}^{K-1} y^{k+1/2}$, and have

$$2\gamma\left[\ell(\bar{z}^K, b) - \ell(z, b) + \sum_{i=1}^n \left(r_i(\bar{x}_i^K) - r_i(x_i)\right) + \langle A\bar{x}^K - \bar{z}^K, y\rangle - \langle Ax - z, \bar{y}^K\rangle\right]$$

$$\leq \frac{1}{K}\left(\|x^0 - x\|^2 + \|w^0 - x\|^2 + \|z^0 - z\|^2 + \|y^0 - y\|^2 + \|u^0 - y\|^2\right)$$

$$- \frac{1}{K}\left(\|x^K - x\|^2 + \|w^K - x\|^2 + \|z^K - z\|^2 + \|y^K - y\|^2 + \|u^K - y\|^2\right)$$

$$- \frac{\tau}{K}\sum_{k=0}^{K-1}\|x^{k+1/2} - x^k\|^2 - \frac{1-\tau}{K}\sum_{k=0}^{K-1}\|w^k - x^{k+1/2}\|^2 - \frac{1}{K}\sum_{k=0}^{K-1}\|z^{k+1/2} - z^k\|^2$$

$$- \frac{\tau}{K}\sum_{k=0}^{K-1}\|y^{k+1/2} - y^k\|^2 - \frac{1-\tau}{K}\sum_{k=0}^{K-1}\|u^k - y^{k+1/2}\|^2$$

$$+ \frac{1}{K}\sum_{k=0}^{K-1}\left[\|w^{k+1}\|^2 - (1-\tau)\|x^k\|^2 - \tau\|w^k\|^2\right]$$

$$+ \frac{2}{K}\sum_{k=0}^{K-1}\langle (1-\tau)x^k + \tau w^k - w^{k+1}, x\rangle$$

$$+ \frac{1}{K}\sum_{k=0}^{K-1}\left[\|y^{k+1}\|^2 - (1-\tau)\|y^k\|^2 - \tau\|u^k\|^2\right]$$

$$+ \frac{2}{K}\sum_{k=0}^{K-1}\langle (1-\tau)y^k + \tau u^k - u^{k+1}, y\rangle$$

$$- \frac{2\gamma}{K}\sum_{k=0}^{K-1}\langle A(x^{k+1/2} - x^0), Q(y^{k+1/2} - u^k) - y^{k+1/2} + u^k\rangle$$

$$- \frac{2\gamma}{K}\sum_{k=0}^{K-1}\langle A(x^0 - x), Q(y^{k+1/2} - u^k) - y^{k+1/2} + u^k\rangle$$

$$+ \frac{2\gamma}{K}\sum_{k=0}^{K-1}\langle \sum_{i=1}^n [Q(A_i x_i^{k+1/2} - A_i w_i^k) + A_i w_i^k - A_i x_i^{k+1/2}], y^{k+1/2} - y^0\rangle$$

$$+ \frac{2\gamma}{K} \sum_{k=0}^{K-1} \langle \sum_{i=1}^{n} [Q(A_i x_i^{k+1/2} - A_i w_i^k) + A_i w_i^k - A_i x_i^{k+1/2}], y^0 - y \rangle$$

$$+ \frac{2\gamma^2}{K} \sum_{k=0}^{K-1} \|A^T Q(y^{k+1/2} - u^k)\|^2 + \frac{2\gamma^2}{K} \sum_{k=0}^{K-1} \| \sum_{i=1}^{n} Q(A_i x_i^{k+1/2} - A_i w_i^k)\|^2$$

$$+ \frac{2\gamma^2 L_r^2}{K} \sum_{k=0}^{K-1} \|x^{k+1/2} - x^k\|^2 + \frac{2\gamma^2}{K} \sum_{k=0}^{K-1} \|y^{k+1/2} - y^k\|^2$$

$$+ \frac{2\gamma^2}{K} \sum_{k=0}^{K-1} \|z^{k+1/2} - z^k\|^2 + \frac{2\gamma^2 L_\ell^2}{K} \sum_{k=0}^{K-1} \|z^{k+1/2} - z^k\|^2.$$

As in (15) we pass to the gap criterion by taking the maximum in $y \in \mathcal{Y}$ and the minimum in $x \in \mathcal{X}$ and $z \in \mathcal{Z}$. Additionally, we also take the mathematical expectation

$$2\gamma \mathbb{E} \text{gap}(\bar{x}^K, \bar{z}^K, \bar{y}^K)$$

$$\leq \frac{1}{K} \left( \max_{x \in \mathcal{X}} \|x^0 - x\|^2 + \max_{x \in \mathcal{X}} \|w^0 - x\|^2 + \max_{z \in \mathcal{Z}} \|z^0 - z\|^2 \right.$$

$$\left. + \max_{y \in \mathcal{Y}} \|y^0 - y\|^2 + \max_{y \in \mathcal{Y}} \|u^0 - y\|^2 \right)$$

$$- \frac{\tau}{K} \sum_{k=0}^{K-1} \mathbb{E}\|x^{k+1/2} - x^k\|^2 - \frac{1-\tau}{K} \sum_{k=0}^{K-1} \mathbb{E}\|w^k - x^{k+1/2}\|^2$$

$$- \frac{1}{K} \sum_{k=0}^{K-1} \mathbb{E}\|z^{k+1/2} - z^k\|^2$$

$$- \frac{\tau}{K} \sum_{k=0}^{K-1} \mathbb{E}\|y^{k+1/2} - y^k\|^2 - \frac{1-\tau}{K} \sum_{k=0}^{K-1} \mathbb{E}\|u^k - y^{k+1/2}\|^2$$

$$+ \frac{1}{K} \sum_{k=0}^{K-1} \mathbb{E}\|w^{k+1}\|^2 - (1-\tau)\|x^k\|^2 - \tau\|w^k\|^2$$

$$+ \frac{2}{K} \mathbb{E} \max_{x \in \mathcal{X}} \sum_{k=0}^{K-1} \langle (1-\tau)x^k + \tau w^k - w^{k+1}, x \rangle$$

$$+ \frac{1}{K} \sum_{k=0}^{K-1} \mathbb{E}\|y^{k+1}\|^2 - (1-\tau)\|y^k\|^2 - \tau\|u^k\|^2$$

$$+ \frac{2}{K} \mathbb{E} \max_{y \in \mathcal{Y}} \sum_{k=0}^{K-1} \langle (1-\tau)y^k + \tau u^k - u^{k+1}, y \rangle$$

$$- \frac{2\gamma}{K} \sum_{k=0}^{K-1} \mathbb{E} \langle A(x^{k+1/2} - x^0), Q(y^{k+1/2} - u^k) - y^{k+1/2} + u^k \rangle$$

$$+ \frac{2\gamma}{K} \cdot \mathbb{E} \max_{x \in \mathcal{X}} \sum_{k=0}^{K-1} \langle A(x - x^0), Q(y^{k+1/2} - u^k) - y^{k+1/2} + u^k \rangle$$

$$+ \frac{2\gamma}{K} \sum_{k=0}^{K-1} \mathbb{E} \langle \sum_{i=1}^{n} [Q(A_i x_i^{k+1/2} - A_i w_i^k) + A_i w_i^k - A_i x_i^{k+1/2}], y^{k+1/2} - y^0 \rangle$$

$$+ \frac{2\gamma}{K} \cdot \mathbb{E} \max_{y \in \mathcal{Y}} \sum_{k=0}^{K-1} \langle \sum_{i=1}^{n} [Q(A_i x_i^{k+1/2} - A_i w_i^k) + A_i w_i^k - A_i x_i^{k+1/2}], y^0 - y \rangle$$

$$+ \frac{2\gamma^2}{K} \sum_{k=0}^{K-1} \mathbb{E}\|A^T Q(y^{k+1/2} - u^k)\|^2 + \frac{2\gamma^2}{K} \sum_{k=0}^{K-1} \mathbb{E}\| \sum_{i=1}^{n} Q(A_i x_i^{k+1/2} - A_i w_i^k)\|^2$$

$$+ \frac{2\gamma^2 L_r^2}{K} \sum_{k=0}^{K-1} \mathbb{E}\|x^{k+1/2} - x^k\|^2 + \frac{2\gamma^2}{K} \sum_{k=0}^{K-1} \mathbb{E}\|y^{k+1/2} - y^k\|^2$$

$$+ \frac{2\gamma^2}{K} \sum_{k=0}^{K-1} \mathbb{E}\|z^{k+1/2} - z^k\|^2 + \frac{2\gamma^2 L_\ell^2}{K} \sum_{k=0}^{K-1} \mathbb{E}\|z^{k+1/2} - z^k\|^2. \tag{23}$$

Next, we work with the terms of (23) separately. Using that $1 - \tau = p$ and lines $11 - 19$, we get

$$\mathbb{E}\|w^{k+1}\|^2 - (1-\tau)\|x^k\|^2 - \tau\|w^k\|^2$$
$$= \mathbb{E}\mathbb{E}_{b_k}\left[\|w^{k+1}\|^2\right] - (1-\tau)\|x^k\|^2 - \tau\|w^k\|^2$$
$$= \mathbb{E}p\|x^k\|^2 + (1-p)\|w^k\|^2 - (1-\tau)\|x^k\|^2 - \tau\|w^k\|^2 = 0. \tag{24}$$

The same way we can obtain

$$\mathbb{E}\|u^{k+1}\|^2 - (1-\tau)\|y^k\|^2 - \tau\|u^k\|^2 = 0. \tag{25}$$

With $1 - \tau = p$, one can also note

$$\mathbb{E}\max_{x \in \mathcal{X}} \sum_{k=0}^{K-1} \langle (1-\tau)x^k + \tau w^k - w^{k+1}, x \rangle = \mathbb{E}\max_{x \in \mathcal{X}} \sum_{k=0}^{K-1} \langle (1-\tau)x^k + \tau w^k - w^{k+1}, x \rangle + 0$$

$$= \mathbb{E}\max_{x \in \mathcal{X}} \sum_{k=0}^{K-1} \langle (1-\tau)x^k + \tau w^k - w^{k+1}, x \rangle$$

$$+ \mathbb{E}\sum_{k=0}^{K-1} \langle (1-\tau)x^k + \tau w^k - \mathbb{E}_{b_k}[w^{k+1}], -x^0 \rangle$$

$$= \mathbb{E}\max_{x \in \mathcal{X}} \sum_{k=0}^{K-1} \langle (1-\tau)x^k + \tau w^k - w^{k+1}, x - x^0 \rangle.$$

By Cauchy Schwartz inequality: $2\langle a, b \rangle \leq \eta\|a\|^2 + \frac{1}{\eta}\|b\|^2$ with $a = \sum_{k=0}^{K-1}[(1-\tau)x^k + \tau w^k - w^{k+1}]$, $b = x - x^0$ and $\eta = \frac{1}{4}$, one can obtain

$$\mathbb{E}\max_{x \in \mathcal{X}} \sum_{k=0}^{K-1} \langle (1-\tau)x^k + \tau w^k - w^{k+1}, x \rangle$$

$$\leq \mathbb{E}\max_{x \in \mathcal{X}} \left[ \frac{1}{8}\| \sum_{k=0}^{K-1}[(1-\tau)x^k + \tau w^k - w^{k+1}]\|^2 + 2\|x - x^0\|^2 \right]$$

$$= \mathbb{E}\max_{x \in \mathcal{X}} 2\|x - x^0\|^2 + \mathbb{E}\frac{1}{8}\| \sum_{k=0}^{K-1}[(1-\tau)x^k + \tau w^k - w^{k+1}]\|^2$$

$$= \mathbb{E}\max_{x \in \mathcal{X}} 2\|x - x^0\|^2 + \frac{1}{8} \sum_{k=0}^{K-1} \mathbb{E}\|(1-\tau)x^k + \tau w^k - w^{k+1}\|^2$$

$$+ \frac{1}{4} \sum_{k_1 < k_2} \mathbb{E}\langle (1-\tau)x^{k_1} + \tau w^{k_1} - w^{k_1+1}, (1-\tau)x^{k_2} + \tau w^{k_2} - w^{k_2+1} \rangle$$

$$= \mathbb{E}\max_{x \in \mathcal{X}} 2\|x - x^0\|^2 + \frac{1}{8} \sum_{k=0}^{K-1} \mathbb{E}\|(1-\tau)x^k + \tau w^k - w^{k+1}\|^2$$

$$+ \frac{1}{4} \sum_{k_1 < k_2} \mathbb{E}\langle (1-\tau)x^{k_1} + \tau w^{k_1} - w^{k_1+1}, \mathbb{E}_{b_{k_2}}[(1-\tau)x^{k_2} + \tau w^{k_2} - w^{k_2+1}] \rangle$$

$$= \mathbb{E}\max_{x \in \mathcal{X}} 2\|x - x^0\|^2 + \frac{1}{8} \sum_{k=0}^{K-1} \mathbb{E}\|(1-\tau)x^k + \tau w^k - w^{k+1}\|^2$$

$$= \mathbb{E}\max_{x \in \mathcal{X}} 2\|x - x^0\|^2 + \frac{1}{8} \sum_{k=0}^{K-1} \mathbb{E}\|\mathbb{E}_{b_k}[w^{k+1}] - w^{k+1}\|^2$$

$$= \mathbb{E}\max_{x \in \mathcal{X}} 2\|x - x^0\|^2 + \frac{1}{8} \sum_{k=0}^{K-1} \mathbb{E}\mathbb{E}_{b_k}[\|w^{k+1}\|^2] - \|\mathbb{E}_{b_k}[w^{k+1}]\|^2$$

$$= \mathbb{E}\max_{x \in \mathcal{X}} 2\|x - x^0\|^2 + \frac{1}{8} \sum_{k=0}^{K-1} \mathbb{E}\mathbb{E}_{b_k}[\|w^{k+1}\|^2] - \|\mathbb{E}_{b_k}[w^{k+1}]\|^2$$

$$= \mathbb{E}\max_{x \in \mathcal{X}} 2\|x - x^0\|^2 + \frac{1}{8} \sum_{k=0}^{K-1} \mathbb{E}\tau\|w^k\|^2 + (1 - \tau)\|x^k\|^2 - \|(1 - \tau)x^k + \tau w^k\|^2$$

$$= \mathbb{E}\max_{x \in \mathcal{X}} 2\|x - x^0\|^2 + \frac{1}{8} \sum_{k=0}^{K-1} \mathbb{E}\tau(1 - \tau)\|w^k - x^k\|^2. \tag{26}$$

Making the same steps, one can get

$$\mathbb{E}\max_{y \in \mathcal{Y}} \sum_{k=0}^{K-1} \langle (1 - \tau)y^k + \tau u^k - u^{k+1}, y \rangle$$

$$\leq \mathbb{E}\max_{y \in \mathcal{Y}} 2\|y - y^0\|^2 + \frac{1}{8} \sum_{k=0}^{K-1} \mathbb{E}\tau(1 - \tau)\|u^k - y^k\|^2. \tag{27}$$

With unbiasedness of $Q$, we have

$$\mathbb{E}\langle A(x^{k+1/2} - x^0), Q(y^{k+1/2} - u^k) - y^{k+1/2} + u^k \rangle$$
$$= \mathbb{E}\langle A(x^{k+1/2} - x^0), \mathbb{E}_Q[Q(y^{k+1/2} - u^k)] - y^{k+1/2} + u^k \rangle = 0. \tag{28}$$

And

$$\mathbb{E}\langle \sum_{i=1}^n [Q(A_i x_i^{k+1/2} - A_i w_i^k) + A_i w_i^k - A_i x_i^{k+1/2}], y^{k+1/2} - y^0 \rangle = 0. \tag{29}$$

Also with Cauchy Schwartz inequality: $2\langle a, b \rangle \leq \eta\|a\|^2 + \frac{1}{\eta}\|b\|^2$ with $a = \sum_{k=0}^{K-1} A^T[Q(y^{k+1/2} - u^k) - y^{k+1/2} + u^k]$, $b = x - x^0$ and $\eta = \gamma$, one can obtain

$$\mathbb{E}\max_{x \in \mathcal{X}} \sum_{k=0}^{K-1} \langle x - x^0, A^T Q(y^{k+1/2} - u^k) - y^{k+1/2} + u^k \rangle$$

$$\leq \mathbb{E}\max_{x \in \mathcal{X}} \frac{1}{2\gamma}\|x^0 - x\|^2 + \mathbb{E}\frac{\gamma}{2}\|\sum_{k=0}^{K-1} A^T[Q(y^{k+1/2} - u^k) - y^{k+1/2} + u^k]\|^2$$

$$= \mathbb{E}\max_{x \in \mathcal{X}} \frac{1}{2\gamma}\|x^0 - x\|^2 + \mathbb{E}\frac{\gamma}{2} \sum_{k=0}^{K-1} \|A^T[Q(y^{k+1/2} - u^k) - y^{k+1/2} + u^k]\|^2$$

$$+ \mathbb{E}\gamma \sum_{k_1 < k_2} \langle A^T[Q(y^{k_1+1/2} - u^{k_1}) - y^{k_1+1/2} + u^{k_1}], A^T[Q(y^{k_2+1/2} - u^{k_2}) - y^{k_2+1/2} + u^{k_2}] \rangle$$

$$= \mathbb{E}\max_{x \in \mathcal{X}} \frac{1}{2\gamma}\|x^0 - x\|^2 + \mathbb{E}\frac{\gamma}{2} \sum_{k=0}^{K-1} \|A^T[Q(y^{k+1/2} - u^k) - y^{k+1/2} + u^k]\|^2$$

$$+ \mathbb{E}\gamma \sum_{k_1 < k_2} \langle A^T[Q(y^{k_1+1/2} - u^{k_1}) - y^{k_1+1/2} + u^{k_1}], A^T\mathbb{E}_{Q_{k_2}}[Q(y^{k_2+1/2} - u^{k_2}) - y^{k_2+1/2} + u^{k_2}] \rangle$$

$$= \mathbb{E}\max_{x \in \mathcal{X}} \frac{1}{2\gamma}\|x^0 - x\|^2 + \mathbb{E}\frac{\gamma}{2} \sum_{k=0}^{K-1} \|A^T[Q(y^{k+1/2} - u^k) - y^{k+1/2} + u^k]\|^2$$

$$= \mathbb{E}\max_{x \in \mathcal{X}} \frac{1}{2\gamma}\|x^0 - x\|^2 + \mathbb{E}\frac{\gamma}{2} \sum_{k=0}^{K-1} \mathbb{E}_Q \left[ \|A^T[Q(y^{k+1/2} - u^k)] - \mathbb{E}_Q[A^T[Q(y^{k+1/2} - u^k)]]\|^2 \right]$$

$$\leq \mathbb{E}\max_{x \in \mathcal{X}} \frac{1}{2\gamma} \|x^0 - x\|^2 + \mathbb{E}\frac{\gamma}{2} \sum_{k=0}^{K-1} \|A^T[Q(y^{k+1/2} - u^k)]\|^2. \tag{30}$$

The same way one can note that

$$\mathbb{E}\max_{y \in \mathcal{Y}} \sum_{k=0}^{K-1} \sum_{i=1}^{n} \langle \sum_{i=1}^{n} [Q(A_i x_i^{k+1/2} - A_i w_i^k) + A_i w_i^k - A_i x_i^{k+1/2}], y^0 - y \rangle$$

$$\leq \mathbb{E}\max_{y \in \mathcal{Y}} \frac{1}{2\gamma} \|y^0 - y\|^2 + \mathbb{E}\frac{\gamma}{2} \sum_{k=0}^{K-1} \|\sum_{i=1}^{n} Q(A_i x_i^{k+1/2} - A_i w_i^k)\|^2. \tag{31}$$

Combining (23) with (24), (25), (26), (27), (28), (29), (30), (31), we obtain

$$2\gamma \mathbb{E}\text{gap}(\bar{x}^K, \bar{z}^K, \bar{y}^K)$$

$$\leq \frac{1}{K} \left( \max_{x \in \mathcal{X}} \|x^0 - x\|^2 + \max_{x \in \mathcal{X}} \|w^0 - x\|^2 + \max_{z \in \mathcal{Z}} \|z^0 - z\|^2 \right.$$

$$\left. + \max_{y \in \mathcal{Y}} \|y^0 - y\|^2 + \max_{y \in \mathcal{Y}} \|u^0 - y\|^2 \right)$$

$$- \frac{\tau}{K} \sum_{k=0}^{K-1} \mathbb{E}\|x^{k+1/2} - x^k\|^2 - \frac{1-\tau}{K} \sum_{k=0}^{K-1} \mathbb{E}\|w^k - x^{k+1/2}\|^2$$

$$- \frac{1}{K} \sum_{k=0}^{K-1} \mathbb{E}\|z^{k+1/2} - z^k\|^2$$

$$- \frac{\tau}{K} \sum_{k=0}^{K-1} \mathbb{E}\|y^{k+1/2} - y^k\|^2 - \frac{1-\tau}{K} \sum_{k=0}^{K-1} \mathbb{E}\|u^k - y^{k+1/2}\|^2$$

$$+ \frac{4}{K} \mathbb{E}\max_{x \in \mathcal{X}} \|x - x^0\|^2 + \frac{1}{4K} \sum_{k=0}^{K-1} \mathbb{E}\tau(1-\tau)\|w^k - x^k\|^2$$

$$+ \frac{4}{K} \mathbb{E}\max_{y \in \mathcal{Y}} \|y - y^0\|^2 + \frac{1}{4K} \sum_{k=0}^{K-1} \mathbb{E}\tau(1-\tau)\|u^k - y^k\|^2$$

$$+ \frac{1}{K} \mathbb{E}\max_{x \in \mathcal{X}} \|x^0 - x\|^2 + \frac{\gamma^2}{K} \mathbb{E}\sum_{k=0}^{K-1} \|A^T[Q(y^{k+1/2} - u^k)]\|^2$$

$$+ \frac{1}{K} \mathbb{E}\max_{y \in \mathcal{Y}} \|y^0 - y\|^2 + \frac{\gamma^2}{K} \mathbb{E}\sum_{k=0}^{K-1} \|\sum_{i=1}^{n} Q(A_i x_i^{k+1/2} - A_i w_i^k)\|^2$$

$$+ \frac{3\gamma^2}{K} \sum_{k=0}^{K-1} \mathbb{E}\|A^T Q(y^{k+1/2} - u^k)\|^2 + \frac{3\gamma^2}{K} \sum_{k=0}^{K-1} \mathbb{E}\|\sum_{i=1}^{n} Q(A_i x_i^{k+1/2} - A_i w_i^k)\|^2$$

$$+ \frac{2\gamma^2 L_r^2}{K} \sum_{k=0}^{K-1} \mathbb{E}\|x^{k+1/2} - x^k\|^2 + \frac{2\gamma^2}{K} \sum_{k=0}^{K-1} \mathbb{E}\|y^{k+1/2} - y^k\|^2$$

$$+ \frac{2\gamma^2}{K} \sum_{k=0}^{K-1} \mathbb{E}\|z^{k+1/2} - z^k\|^2 + \frac{2\gamma^2 L_\ell^2}{K} \sum_{k=0}^{K-1} \mathbb{E}\|z^{k+1/2} - z^k\|^2$$

$$\leq \frac{1}{K} \left( 6\max_{x \in \mathcal{X}} \|x^0 - x\|^2 + \max_{x \in \mathcal{X}} \|w^0 - x\|^2 + \max_{z \in \mathcal{Z}} \|z^0 - z\|^2 \right.$$

$$\left. + 6\max_{y \in \mathcal{Y}} \|y^0 - y\|^2 + \max_{y \in \mathcal{Y}} \|u^0 - y\|^2 \right)$$

$$- \frac{\tau}{K} \sum_{k=0}^{K-1} \mathbb{E}\|x^{k+1/2} - x^k\|^2 - \frac{1-\tau}{K} \sum_{k=0}^{K-1} \mathbb{E}\|w^k - x^{k+1/2}\|^2$$

$$- \frac{1}{K} \sum_{k=0}^{K-1} \mathbb{E}\|z^{k+1/2} - z^k\|^2$$

$$- \frac{\tau}{K} \sum_{k=0}^{K-1} \mathbb{E}\|y^{k+1/2} - y^k\|^2 - \frac{1-\tau}{K} \sum_{k=0}^{K-1} \mathbb{E}\|u^k - y^{k+1/2}\|^2$$

$$+ \frac{1}{4K} \sum_{k=0}^{K-1} \mathbb{E}\tau(1-\tau)\|w^k - x^k\|^2 + \frac{1}{4K} \sum_{k=0}^{K-1} \mathbb{E}\tau(1-\tau)\|u^k - y^k\|^2$$

$$+ \frac{3\gamma^2}{K} \sum_{k=0}^{K-1} \mathbb{E}\|A^T Q(y^{k+1/2} - u^k)\|^2 + \frac{3\gamma^2}{K} \sum_{k=0}^{K-1} \mathbb{E}\|\sum_{i=1}^{n} Q(A_i x_i^{k+1/2} - A_i w_i^k)\|^2$$

$$+ \frac{2\gamma^2 L_r^2}{K} \sum_{k=0}^{K-1} \mathbb{E}\|x^{k+1/2} - x^k\|^2 + \frac{2\gamma^2}{K} \sum_{k=0}^{K-1} \mathbb{E}\|y^{k+1/2} - y^k\|^2$$

$$+ \frac{2\gamma^2(1 + L_\ell^2)}{K} \sum_{k=0}^{K-1} \mathbb{E}\|z^{k+1/2} - z^k\|^2.$$

Applying Cauchy Schwartz inequality and using that $\tau \leq 1$, we get

$$2\gamma\mathbb{E}\text{gap}(\bar{x}^K, \bar{z}^K, \bar{y}^K)$$

$$\leq \frac{1}{K} \left( 6 \max_{x \in \mathcal{X}} \|x^0 - x\|^2 + \max_{x \in \mathcal{X}} \|w^0 - x\|^2 + \max_{z \in \mathcal{Z}} \|z^0 - z\|^2 \right.$$

$$\left. + 6 \max_{y \in \mathcal{Y}} \|y^0 - y\|^2 + \max_{y \in \mathcal{Y}} \|u^0 - y\|^2 \right)$$

$$- \frac{\tau}{K} \sum_{k=0}^{K-1} \mathbb{E}\|x^{k+1/2} - x^k\|^2 - \frac{1-\tau}{K} \sum_{k=0}^{K-1} \mathbb{E}\|w^k - x^{k+1/2}\|^2$$

$$- \frac{1}{K} \sum_{k=0}^{K-1} \mathbb{E}\|z^{k+1/2} - z^k\|^2$$

$$- \frac{\tau}{K} \sum_{k=0}^{K-1} \mathbb{E}\|y^{k+1/2} - y^k\|^2 - \frac{1-\tau}{K} \sum_{k=0}^{K-1} \mathbb{E}\|u^k - y^{k+1/2}\|^2$$

$$+ \frac{1}{2K} \sum_{k=0}^{K-1} \mathbb{E}(1-\tau)\|w^k - x^{k+1/2}\|^2 + \frac{1}{2K} \sum_{k=0}^{K-1} \mathbb{E}(1-\tau)\|x^{k+1/2} - x^k\|^2$$

$$+ \frac{1}{2K} \sum_{k=0}^{K-1} \mathbb{E}(1-\tau)\|u^k - y^{k+1/2}\|^2 + \frac{1}{2K} \sum_{k=0}^{K-1} \mathbb{E}(1-\tau)\|y^{k+1/2} - y^k\|^2$$

$$+ \frac{3\gamma^2}{K} \sum_{k=0}^{K-1} \mathbb{E}\|A^T Q(y^{k+1/2} - u^k)\|^2 + \frac{3\gamma^2}{K} \sum_{k=0}^{K-1} \mathbb{E}\|\sum_{i=1}^{n} Q(A_i x_i^{k+1/2} - A_i w_i^k)\|^2$$

$$+ \frac{2\gamma^2 L_r^2}{K} \sum_{k=0}^{K-1} \mathbb{E}\|x^{k+1/2} - x^k\|^2 + \frac{2\gamma^2}{K} \sum_{k=0}^{K-1} \mathbb{E}\|y^{k+1/2} - y^k\|^2$$

$$+ \frac{2\gamma^2(1 + L_\ell^2)}{K} \sum_{k=0}^{K-1} \mathbb{E}\|z^{k+1/2} - z^k\|^2$$

$$= \frac{1}{K} \left( 6 \max_{x \in \mathcal{X}} \|x^0 - x\|^2 + \max_{x \in \mathcal{X}} \|w^0 - x\|^2 + \max_{z \in \mathcal{Z}} \|z^0 - z\|^2 \right.$$

$$\left. + 6 \max_{y \in \mathcal{Y}} \|y^0 - y\|^2 + \max_{y \in \mathcal{Y}} \|u^0 - y\|^2 \right)$$

$$- \left( \frac{3\tau - 1}{2} - 2\gamma^2 L_r^2 \right) \cdot \frac{1}{K} \sum_{k=0}^{K-1} \mathbb{E} \|x^{k+1/2} - x^k\|^2 - \frac{1-\tau}{2K} \sum_{k=0}^{K-1} \mathbb{E} \|w^k - x^{k+1/2}\|^2$$

$$- \left( 1 - 2\gamma^2 (1 + L_\ell^2) \right) \frac{1}{K} \sum_{k=0}^{K-1} \mathbb{E} \|z^{k+1/2} - z^k\|^2$$

$$- \left( \frac{3\tau - 1}{2} - 2\gamma^2 \right) \frac{1}{K} \sum_{k=0}^{K-1} \mathbb{E} \|y^{k+1/2} - y^k\|^2 - \frac{1-\tau}{2K} \sum_{k=0}^{K-1} \mathbb{E} \|u^k - y^{k+1/2}\|^2$$

$$+ \frac{3\gamma^2}{K} \sum_{k=0}^{K-1} \mathbb{E} \|A^T Q(y^{k+1/2} - u^k)\|^2 + \frac{3\gamma^2}{K} \sum_{k=0}^{K-1} \mathbb{E} \|\sum_{i=1}^{n} Q(A_i x_i^{k+1/2} - A_i w_i^k)\|^2.$$

$$(32)$$

Using the notation of $\lambda_{\max}(\cdot)$ as a maximum eigenvalue and the definition of unbiased compression, we get

$$\mathbb{E} \|A^T Q(y^{k+1/2} - u^k)\|^2 \leq \lambda_{\max}(AA^T) \mathbb{E} \|Q(y^{k+1/2} - u^k)\|^2$$
$$\leq \lambda_{\max}(AA^T) \omega \mathbb{E} \|y^{k+1/2} - u^k\|^2.$$

For $\mathbb{E} \|\sum_{i=1}^{n} Q(A_i x_i^{k+1/2} - A_i w_i^k)\|^2$ we have two options. If $\sum_{i=1}^{n} Q(A_i x_i^{k+1/2} - A_i w_i^k) = Q(\sum_{i=1}^{n} [A_i x_i^{k+1/2} - A_i w_i^k]) = Q(Ax^{k+1/2} - Aw^k)$, then

$$\mathbb{E} \|\sum_{i=1}^{n} Q(A_i x_i^{k+1/2} - A_i w_i^k)\|^2 = \mathbb{E} \|Q(Ax^{k+1/2} - Aw^k)\|^2$$
$$\leq \omega \mathbb{E} \|A(x^{k+1/2} - w^k)\|^2$$
$$\leq \lambda_{\max}(A^T A) \omega \mathbb{E} \|x^{k+1/2} - w^k\|^2.$$

If $\sum_{i=1}^{n} Q(A_i x_i^{k+1/2} - A_i w_i^k) \neq Q(\sum_{i=1}^{n} [A_i x_i^{k+1/2} - A_i w_i^k])$, but $Q$ are independent, then

$$\mathbb{E} \|\sum_{i=1}^{n} Q(A_i x_i^{k+1/2} - A_i w_i^k)\|^2$$

$$= \sum_{i=1}^{n} \mathbb{E} \|Q(A_i x_i^{k+1/2} - A_i w_i^k)\|^2$$
$$+ \sum_{i \neq j} \mathbb{E} \langle Q(A_i x_i^{k+1/2} - A_i w_i^k), Q(A_j x_j^{k+1/2} - A_j w_j^k) \rangle$$
$$= \sum_{i=1}^{n} \mathbb{E} \|Q(A_i x_i^{k+1/2} - A_i w_i^k)\|^2$$
$$+ \sum_{i \neq j} \mathbb{E} \langle \mathbb{E}_{Q_i}[Q(A_i x_i^{k+1/2} - A_i w_i^k)], \mathbb{E}_{Q_j}[Q(A_j x_j^{k+1/2} - A_j w_j^k)] \rangle$$
$$= \sum_{i=1}^{n} \mathbb{E} \|Q(A_i x_i^{k+1/2} - A_i w_i^k)\|^2$$
$$+ \sum_{i \neq j} \mathbb{E} \langle A_i x_i^{k+1/2} - A_i w_i^k, A_j x_j^{k+1/2} - A_j w_j^k \rangle$$
$$= \sum_{i=1}^{n} \mathbb{E} \|Q(A_i x_i^{k+1/2} - A_i w_i^k)\|^2$$
$$+ \mathbb{E} \|\sum_{i=1}^{n} [A_i x_i^{k+1/2} - A_i w_i^k]\|^2 - \sum_{i=1}^{n} \mathbb{E} \|A_i x_i^{k+1/2} - A_i w_i^k\|^2$$
$$\leq \omega \sum_{i=1}^{n} \mathbb{E} \|A_i x_i^{k+1/2} - A_i w_i^k\|^2 + \mathbb{E} \|A(x^{k+1/2} - w^k)\|^2$$

$$\leq \omega \sum_{i=1}^{n} \lambda_{\max}(A_i^T A_i) \mathbb{E}\|x_i^{k+1/2} - w_i^k\|^2 + \lambda_{\max}(A^T A)\mathbb{E}\|x^{k+1/2} - w^k\|^2$$

$$\leq \omega \max_i \left\{\lambda_{\max}(A_i^T A_i)\right\} \sum_{i=1}^{n} \mathbb{E}\|x_i^{k+1/2} - w_i^k\|^2 + \lambda_{\max}(A^T A)\mathbb{E}\|x^{k+1/2} - w^k\|^2$$

$$= \left(\omega \max_i \left\{\lambda_{\max}(A_i^T A_i)\right\} + \lambda_{\max}(A^T A)\right) \mathbb{E}\|x^{k+1/2} - w^k\|^2.$$

Let us introduce

$$\chi_{\text{compress}} = \begin{cases} \omega \lambda_{\max}(A^T A), \\ \omega \max_i \left\{\lambda_{\max}(A_i^T A_i)\right\} + \lambda_{\max}(A^T A), \end{cases}$$

depending on the case $Q$ we consider. Let us return to (32) and obtain

$$2\gamma \mathbb{E}\text{gap}(\bar{x}^K, \bar{z}^K, \bar{y}^K)$$

$$\leq \frac{1}{K}\left(6 \max_{x \in \mathcal{X}}\|x^0 - x\|^2 + \max_{x \in \mathcal{X}}\|w^0 - x\|^2 + \max_{z \in \mathcal{Z}}\|z^0 - z\|^2\right.$$

$$\left. + 6 \max_{y \in \mathcal{Y}}\|y^0 - y\|^2 + \max_{y \in \mathcal{Y}}\|u^0 - y\|^2\right)$$

$$- \left(\frac{3\tau - 1}{2} - 2\gamma^2 L_r^2\right) \cdot \frac{1}{K}\sum_{k=0}^{K-1}\mathbb{E}\|x^{k+1/2} - x^k\|^2$$

$$- \left(\frac{1-\tau}{2} - 3\chi_{\text{compress}}\gamma^2\right)\frac{1}{K}\sum_{k=0}^{K-1}\mathbb{E}\|w^k - x^{k+1/2}\|^2$$

$$- \left(1 - 2\gamma^2(1 + L_\ell^2)\right)\frac{1}{K}\sum_{k=0}^{K-1}\mathbb{E}\|z^{k+1/2} - z^k\|^2$$

$$- \left(\frac{3\tau - 1}{2} - 2\gamma^2\right)\frac{1}{K}\sum_{k=0}^{K-1}\mathbb{E}\|y^{k+1/2} - y^k\|^2$$

$$- \left(\frac{1-\tau}{2} - 3\lambda_{\max}(AA^T)\omega\gamma^2\right)\frac{1}{K}\sum_{k=0}^{K-1}\mathbb{E}\|u^k - y^{k+1/2}\|^2.$$

If we choose $\tau \geq \frac{1}{2}$ and $\gamma$ as follows

$$\gamma \leq \frac{1}{4}\min\left\{1; \frac{1}{L_r}; \frac{1}{L_\ell}; \sqrt{\frac{1-\tau}{\chi_{\text{compress}}}}; \sqrt{\frac{1-\tau}{\omega\lambda_{\max}(AA^T)}};\right\},$$

then one can obtain

$$\mathbb{E}\text{gap}(\bar{x}^K, \bar{z}^K, \bar{y}^K) \leq \frac{1}{2\gamma K}\left(6 \max_{x \in \mathcal{X}}\|x^0 - x\|^2 + \max_{x \in \mathcal{X}}\|w^0 - x\|^2 + \max_{z \in \mathcal{Z}}\|z^0 - z\|^2\right.$$

$$\left. + 6 \max_{y \in \mathcal{Y}}\|y^0 - y\|^2 + \max_{y \in \mathcal{Y}}\|u^0 - y\|^2\right).$$

With $\gamma = \frac{1}{4}\min\left\{1; \frac{1}{L_r}; \frac{1}{L_\ell}; \sqrt{\frac{1-\tau}{\chi_{\text{compress}}}}; \sqrt{\frac{1-\tau}{\omega\lambda_{\max}(AA^T)}};\right\}$, we finish the proof. $\square$

### D.4 PROOF OF THEOREM 3.4

**Theorem D.5** (Theorem 3.4). *Let Assumption 2.1 holds. Let problem (4) be solved by Algorithm 3 (Appendix A) with operators and C, which satisfy Definition 3.3. Then for $\tau = 1 - p$ and*

$$\gamma = \frac{1}{4}\min\left\{1; \frac{1}{L_r}; \frac{1}{L_\ell}; \sqrt{\frac{1-\tau}{\delta^2[\lambda_{\max}(AA^T) + n\cdot\max_i\{\lambda_{\max}(A_i A_i^T)\}]}}; \sqrt{\frac{1-\tau}{\omega\lambda_{\max}(AA^T)}};\right\},$$

*it holds that*

$$\mathbb{E}gap(\bar{x}^K, \bar{z}^K, \bar{y}^K) = \mathcal{O}([1 + \tfrac{\delta}{\sqrt{p}}\left(\sqrt{\lambda_{\max}(AA^T)} + n \cdot \max_{i=1,\dots,n}\{\sqrt{\lambda_{\max}(A_iA_i^T)}\}\right) + L_\ell + L_r] \cdot \tfrac{D^2}{K}),$$

*where* $\bar{x}^K := \frac{1}{K}\sum_{k=0}^{K-1} x^{k+1/2}$, $\bar{z}^K := \frac{1}{K}\sum_{k=0}^{K-1} z^{k+1/2}$, $\bar{y}^K := \frac{1}{K}\sum_{k=0}^{K-1} y^{k+1/2}$ *and* $D^2 :=$ $\max_{x,z,y\in\mathcal{X},\mathcal{Z},\mathcal{Y}}\left[\|x^0 - x\|^2 + \|z^0 - z\|^2 + \|y^0 - y\|^2\right].$

To begin with, let us introduce the useful notation for the further proof:

$$\hat{x}_i^k = x_i^k - \gamma A_i^T e^k, \quad \hat{x}_i^{k+1/2} = x_i^{k+1/2} - \gamma A_i^T e^k,$$
$$\hat{y}^k = y^k - \gamma \sum_{i=1}^n e_i^k, \quad \hat{y}^{k+1/2} = y^{k+1/2} - \gamma \sum_{i=1}^n e_i^k. \tag{33}$$

It is easy to verify that such sequences have useful properties:

$$\begin{aligned}
\hat{x}_i^{k+1} =& x_i^{k+1} - \gamma A_i^T e^{k+1} \\
=& \tau x_i^k + (1-\tau)w_i^k - \gamma(A_i^T[C(y^{k+1/2} - u^k + e^k) + u^k] + \nabla r_i(x_i^{k+1/2})) - \gamma A_i^T e^k \\
& - \gamma A_i^T\left(y^{k+1/2} - u^k + e^k - C(y^{k+1/2} - u^k)\right) \\
=& \tau x_i^k + (1-\tau)w_i^k - \gamma(A_i^T u^k + \nabla r_i(x_i^k)) \\
& - \gamma A_i^T\left(y^{k+1/2} - u^k + e^k\right) - \gamma(\nabla r_i(x_i^{k+1/2}) - \nabla r_i(x_i^k)) \\
=& \hat{x}_i^{k+1/2} - \gamma A_i^T\left(y^{k+1/2} - u^k\right) - \gamma(\nabla r_i(x_i^{k+1/2}) - \nabla r_i(x_i^k)), \tag{34}
\end{aligned}$$

and

$$\begin{aligned}
\hat{y}^{k+1} =& y^{k+1} + \gamma \sum_{i=1}^n e_i^{k+1} \\
=& \tau y^k + (1-\tau)u^k + \gamma\left(\sum_{i=1}^n[C(A_i x_i^{k+1/2} - A_i w_i^k + e_i^k) + A_i w_i^k] - z^{k+1/2}\right) \\
& + \gamma \sum_{i=1}^n[A_i x_i^{k+1/2} - A_i w_i^k + e_i^k - C(A_i x_i^{k+1/2} - A_i w_i^k + e_i^k)] \\
=& \tau y^k + (1-\tau)u^k + \gamma\left(\sum_{i=1}^n A_i w_i^k - z^k\right) - \gamma \sum_{i=1}^n e_i^k \\
& + \gamma \sum_{i=1}^n[A_i x_i^{k+1/2} - A_i w_i^k] - \gamma(z^{k+1/2} - z^k) \\
=& \hat{y}^{k+1/2} + \gamma \sum_{i=1}^n[A_i x_i^{k+1/2} - A_i w_i^k] - \gamma(z^{k+1/2} - z^k).
\end{aligned}$$

Now we are ready to start the proof.

*Proof.* We start the proof with the following equations on the variables $\hat{x}_i^{k+1}$, $x_i^{k+1/2}$, $\hat{x}_i^k$ and any $x_i \in \mathbb{R}^{d_i}$:

$$\|\hat{x}_i^{k+1} - x_i\|^2 = \|x_i^{k+1/2} - x_i\|^2 + 2\langle \hat{x}_i^{k+1} - x_i^{k+1/2}, x_i^{k+1/2} - x_i\rangle + \|\hat{x}_i^{k+1} - x_i^{k+1/2}\|^2,$$
$$\|x_i^{k+1/2} - x_i\|^2 = \|\hat{x}_i^k - x_i\|^2 + 2\langle x_i^{k+1/2} - \hat{x}_i^k, x_i^{k+1/2} - x_i\rangle - \|x_i^{k+1/2} - \hat{x}_i^k\|^2.$$

Summing up two previous inequalities and making small rearrangements, we get

$$\begin{aligned}
\|\hat{x}_i^{k+1} - x_i\|^2 =& \|\hat{x}_i^k - x_i\|^2 + 2\langle \hat{x}_i^{k+1} - \hat{x}_i^k, x_i^{k+1/2} - x_i\rangle \\
& + \|\hat{x}_i^{k+1} - x_i^{k+1/2}\|^2 - \|x_i^{k+1/2} - \hat{x}_i^k\|^2. \tag{35}
\end{aligned}$$

Using the definitions (33) and (34), one can obtain

$$
\begin{aligned}
\|\hat{x}_i^{k+1} - x_i^{k+1/2}\|^2 \leq & 2\|\hat{x}_i^{k+1} - \hat{x}_i^{k+1/2}\|^2 + 2\|\hat{x}_i^{k+1/2} - x_i^{k+1/2}\|^2 \\
= & 2\gamma^2\|A_i^T(y^{k+1/2} - u^k) - (\nabla r_i(x_i^{k+1/2}) - \nabla r_i(x_i^k))\|^2 + 2\gamma^2\|A_i^T e^k\|^2 \\
\leq & 4\gamma^2\|A_i^T(y^{k+1/2} - u^k)\|^2 + 4\gamma^2\|\nabla r_i(x_i^{k+1/2}) - \nabla r_i(x_i^k)\|^2 \\
& + 2\gamma^2\|A_i^T e^k\|^2.
\end{aligned}
\tag{36}
$$

With (33), (34) and the update for $x_i^{k+1/2}$, we have

$$
\begin{aligned}
\hat{x}_i^{k+1} - \hat{x}_i^k = & \hat{x}_i^{k+1} - \hat{x}_i^{k+1/2} + \hat{x}_i^{k+1/2} - \hat{x}_i^k \\
= & \hat{x}_i^{k+1} - \hat{x}_i^{k+1/2} + x_i^{k+1/2} - x_i^k \\
= & -\gamma A_i^T\left(y^{k+1/2} - u^k\right) - \gamma(\nabla r_i(x_i^{k+1/2}) - \nabla r_i(x_i^k)) \\
& + (1-\tau)(w_i^k - x_i^k) - \gamma\left(A_i^T u^k + \nabla r_i(x_i^k)\right) \\
= & -\gamma(A_i^T y^{k+1/2} + \nabla r_i(x_i^{k+1/2})) + (1-\tau)(w_i^k - x_i^k).
\end{aligned}
\tag{37}
$$

Combining (35), (36), (37), we get

$$
\begin{aligned}
\|\hat{x}_i^{k+1} - x_i\|^2 \leq & \|\hat{x}_i^k - x_i\|^2 - 2\langle\gamma(A_i^T y^{k+1/2} + \nabla r_i(x_i^{k+1/2})) - (1-\tau)(w_i^k - x_i^k), x_i^{k+1/2} - x_i\rangle \\
& + 4\gamma^2\|A_i^T(y^{k+1/2} - u^k)\|^2 + 4\gamma^2\|\nabla r_i(x_i^{k+1/2}) - \nabla r_i(x_i^k)\|^2 \\
& + 2\gamma^2\|A_i^T e^k\|^2 - \|x_i^{k+1/2} - \hat{x}_i^k\|^2 \\
\leq & \|\hat{x}_i^k - x_i\|^2 - 2\langle\gamma(A_i^T y^{k+1/2} + \nabla r_i(x_i^{k+1/2})), x_i^{k+1/2} - x_i\rangle \\
& + 2(1-\tau)\langle w_i^k - x_i^{k+1/2}, x_i^{k+1/2} - x_i\rangle \\
& + 2(1-\tau)\langle x_i^{k+1/2} - x_i^k, x_i^{k+1/2} - x_i\rangle \\
& + 4\gamma^2\|A_i^T(y^{k+1/2} - u^k)\|^2 + 4\gamma^2\|\nabla r_i(x_i^{k+1/2}) - \nabla r_i(x_i^k)\|^2 \\
& + 2\gamma^2\|A_i^T e^k\|^2 - \frac{1}{2}\|x_i^{k+1/2} - x_i^k\|^2 + \|\hat{x}_i^k - x_i^k\|^2 \\
= & \|\hat{x}_i^k - x_i\|^2 - 2\langle\gamma(A_i^T y^{k+1/2} + \nabla r_i(x_i^{k+1/2})), x_i^{k+1/2} - x_i\rangle \\
& + 2(1-\tau)\langle w_i^k - x_i^{k+1/2}, x_i^{k+1/2} - x_i\rangle \\
& + 2(1-\tau)\langle x_i^{k+1/2} - x_i^k, x_i^{k+1/2} - x_i\rangle \\
& + 4\gamma^2\|A_i^T(y^{k+1/2} - u^k)\|^2 + 4\gamma^2\|\nabla r_i(x_i^{k+1/2}) - \nabla r_i(x_i^k)\|^2 \\
& + 2\gamma^2\|A_i^T e^k\|^2 - \frac{1}{2}\|x_i^{k+1/2} - x_i^k\|^2 + \gamma^2\|A_i^T e^k\|^2.
\end{aligned}
$$

In the last two steps we use (33) and Cauchy Schwartz inequality in the form $-\|a\|^2 \leq -\frac{1}{2}\|a + b\|^2 + \|b\|^2$ with $a = x_i^{k+1/2} - \hat{x}_i^k$ and $b = \hat{x}_i^k - x_i^k$. For the second and third lines we use identity $2\langle a, b\rangle = \|a + b\|^2 - \|a\|^2 - \|b\|^2$, and have

$$
\begin{aligned}
\|\hat{x}_i^{k+1} - x_i\|^2 \leq & \|\hat{x}_i^k - x_i\|^2 - 2\gamma\langle A_i^T y^{k+1/2} + \nabla r_i(x_i^{k+1/2}), x_i^{k+1/2} - x_i\rangle \\
& + (1-\tau)(\|w_i^k - x_i\|^2 - \|w_i^k - x_i^{k+1/2}\|^2 - \|x_i^{k+1/2} - x_i\|^2) \\
& + (1-\tau)(\|x_i^{k+1/2} - x_i^k\|^2 + \|x_i^{k+1/2} - x_i\|^2 - \|x_i^k - x_i\|^2) \\
& + 4\gamma^2\|A_i^T(y^{k+1/2} - u^k)\|^2 + 4\gamma^2\|\nabla r_i(x_i^{k+1/2}) - \nabla r_i(x_i^k)\|^2 \\
& + 3\gamma^2\|A_i^T e^k\|^2 - \frac{1}{2}\|x_i^{k+1/2} - x_i^k\|^2 \\
= & \|\hat{x}_i^k - x_i\|^2 - (1-\tau)\|x_i^k - x_i\|^2 + (1-\tau)\|w_i^k - x_i\|^2 \\
& - 2\gamma\langle A_i^T y^{k+1/2} + \nabla r_i(x_i^{k+1/2}), x_i^{k+1/2} - x_i\rangle \\
& + 4\gamma^2\|A_i^T(y^{k+1/2} - u^k)\|^2 + 4\gamma^2\|\nabla r_i(x_i^{k+1/2}) - \nabla r_i(x_i^k)\|^2
\end{aligned}
$$

$$+ 3\gamma^2 \|A_i^T e^k\|^2 - \left(\tau - \frac{1}{2}\right) \|x_i^{k+1/2} - x_i^k\|^2 - (1-\tau)\|w_i^k - x_i^{k+1/2}\|^2.$$

Summing over all $i$ from 1 to $n$, we deduce

$$\sum_{i=1}^n \|\hat{x}_i^{k+1} - x_i\|^2 \leq \sum_{i=1}^n \|\hat{x}_i^k - x_i\|^2 - (1-\tau)\sum_{i=1}^n \|x_i^k - x_i\|^2 + (1-\tau)\sum_{i=1}^n \|w_i^k - x_i\|^2$$

$$- 2\gamma\sum_{i=1}^n \langle A_i^T y^{k+1/2} + \nabla r_i(x_i^{k+1/2}), x_i^{k+1/2} - x_i\rangle$$

$$+ 4\gamma^2 \sum_{i=1}^n \|A_i^T(y^{k+1/2} - u^k)\|^2 + 4\gamma^2 \sum_{i=1}^n \|\nabla r_i(x_i^{k+1/2}) - \nabla r_i(x_i^k)\|^2$$

$$+ 3\gamma^2 \sum_{i=1}^n \|A_i^T e^k\|^2 - \left(\tau - \frac{1}{2}\right)\sum_{i=1}^n \|x_i^{k+1/2} - x_i^k\|^2$$

$$- (1-\tau)\sum_{i=1}^n \|w_i^k - x_i^{k+1/2}\|^2.$$

With notation of $A = [A_1, \ldots, A_i, \ldots, A_n]$, $x = [x_1^T, \ldots, x_i^T, \ldots, x_n^T]^T$, $\hat{x} = [\hat{x}_1^T, \ldots, \hat{x}_i^T, \ldots, \hat{x}_n^T]^T$ and $w = [w_1^T, \ldots, w_i^T, \ldots, w_n^T]^T$, one can obtain that $\sum_{i=1}^n A_i x_i = Ax$, $\sum_{i=1}^n \|A_i^T e^k\| = \|A^T e^k\|^2$ and $\sum_{i=1}^n \|A_i^T(y^{k+1/2} - u^k)\|^2 = \|A^T(y^{k+1/2} - u^k)\|^2$:

$$\|\hat{x}^{k+1} - x\|^2 \leq \|\hat{x}^k - x\|^2 - (1-\tau)\|x^k - x\|^2 + (1-\tau)\|w^k - x\|^2$$

$$- 2\gamma\langle y^{k+1/2}, A(x^{k+1/2} - x)\rangle - 2\gamma\sum_{i=1}^n \langle \nabla r_i(x_i^{k+1/2}), x_i^{k+1/2} - x_i\rangle$$

$$+ 4\gamma^2 \|A^T(y^{k+1/2} - u^k)\|^2 + 4\gamma^2 \sum_{i=1}^n \|\nabla r_i(x_i^{k+1/2}) - \nabla r_i(x_i^k)\|^2$$

$$+ 3\gamma^2 \|A^T e^k\|^2 - \left(\tau - \frac{1}{2}\right)\|x^{k+1/2} - x^k\|^2 - (1-\tau)\|w^k - x^{k+1/2}\|^2. \quad (38)$$

One can note that the updates for the variable $z$ from lines 4 and 11 of Algorithm 3 are the same as those from lines 6 and 11 of Algorithm 1. Therefore, we can simply use (13), i.e. for $z \in \mathbb{R}^s$ it holds

$$\|z^{k+1} - z\|^2 \leq \|z^k - z\|^2 - \|z^{k+1/2} - z^k\|^2$$

$$+ 2\gamma\langle y^{k+1/2}, z^{k+1/2} - z\rangle - 2\gamma\langle \nabla\ell(z^{k+1/2}, b), z^{k+1/2} - z\rangle$$

$$+ 2\gamma^2 \|y^{k+1/2} - y^k\|^2 + 2\gamma^2 \|\nabla\ell(z^{k+1/2}, b) - \nabla\ell(z^k, b)\|^2. \quad (39)$$

For the updates of the variable $y$ from lines (5), (12) and from (33), we can repeat the same steps as in obtaining (38). In particular, for all $y \in \mathbb{R}^s$, we get

$$\|\hat{y}^{k+1} - y\|^2 \leq \|\hat{y}^k - y\|^2 - (1-\tau)\|y^k - y\|^2 + (1-\tau)\|u^k - y\|^2$$

$$+ 2\gamma\langle \sum_{i=1}^n A_i x_i^{k+1/2} - z^{k+1/2}, y^{k+1/2} - y\rangle$$

$$+ 4\gamma^2 \left\|\sum_{i=1}^n [A_i x_i^{k+1/2} - A_i w_i^k]\right\|^2 + 4\gamma^2 \|z^{k+1/2} - z^k\|^2$$

$$+ 3\gamma^2 \left\|\sum_{i=1}^n e_i^k\right\|^2 - \left(\tau - \frac{1}{2}\right)\|y^{k+1/2} - y^k\|^2 - (1-\tau)\|u^k - y^{k+1/2}\|^2$$

$$= \|\hat{y}^k - y\|^2 - (1-\tau)\|y^k - y\|^2 + (1-\tau)\|u^k - y\|^2$$

$$+ 2\gamma\langle Ax^{k+1/2} - z^{k+1/2}, y^{k+1/2} - y\rangle$$

$$+ 4\gamma^2 \left\|Ax^{k+1/2} - Aw^k\right\|^2 + 4\gamma^2 \|z^{k+1/2} - z^k\|^2$$

$$+ 3\gamma^2 \left\| \sum_{i=1}^{n} e_i^k \right\|^2 - \left( \tau - \frac{1}{2} \right) \|y^{k+1/2} - y^k\|^2 - (1-\tau)\|u^k - y^{k+1/2}\|^2. \quad (40)$$

Here we also use the notation of $A$ and $x$. Summing up (38), (39) and (40), we obtain

$$\|\hat{x}^{k+1} - x\|^2 + \|z^{k+1} - z\|^2 + \|\hat{y}^{k+1} - y\|^2$$
$$\leq \|\hat{x}^k - x\|^2 + \|z^k - z\|^2 + \|\hat{y}^k - y\|^2$$
$$- (1-\tau)\|x^k - x\|^2 + (1-\tau)\|w^k - x\|^2 - (1-\tau)\|y^k - y\|^2 + (1-\tau)\|u^k - y\|^2$$
$$- 2\gamma\langle y^{k+1/2}, A(x^{k+1/2} - x)\rangle - 2\gamma \sum_{i=1}^{n} \langle \nabla r_i(x_i^{k+1/2}), x_i^{k+1/2} - x_i\rangle$$
$$+ 2\gamma\langle y^{k+1/2}, z^{k+1/2} - z\rangle - 2\gamma\langle \nabla\ell(z^{k+1/2}, b), z^{k+1/2} - z\rangle$$
$$+ 2\gamma\langle Ax^{k+1/2} - z^{k+1/2}, y^{k+1/2} - y\rangle$$
$$- \left( \tau - \frac{1}{2} \right) \|x^{k+1/2} - x^k\|^2 - (1-\tau)\|w^k - x^{k+1/2}\|^2 - \|z^{k+1/2} - z^k\|^2$$
$$- \left( \tau - \frac{1}{2} \right) \|y^{k+1/2} - y^k\|^2 - (1-\tau)\|u^k - y^{k+1/2}\|^2$$
$$+ 4\gamma^2\|A^T(y^{k+1/2} - u^k)\|^2 + 4\gamma^2 \sum_{i=1}^{n} \|\nabla r_i(x_i^{k+1/2}) - \nabla r_i(x_i^k)\|^2$$
$$+ 2\gamma^2\|y^{k+1/2} - y^k\|^2 + 2\gamma^2\|\nabla\ell(z^{k+1/2}, b) - \nabla\ell(z^k, b)\|^2$$
$$+ 4\gamma^2 \left\| Ax^{k+1/2} - Aw^k \right\|^2 + 4\gamma^2\|z^{k+1/2} - z^k\|^2$$
$$+ 3\gamma^2\|A^T e^k\|^2 + 3\gamma^2 \left\| \sum_{i=1}^{n} e_i^k \right\|^2.$$

Using convexity and $L_r$-smoothness of the function $r_i$ with convexity and $L_\ell$-smoothness of the function $\ell$, we have

$$\|\hat{x}^{k+1} - x\|^2 + \|z^{k+1} - z\|^2 + \|\hat{y}^{k+1} - y\|^2$$
$$\leq \|\hat{x}^k - x\|^2 + \|z^k - z\|^2 + \|\hat{y}^k - y\|^2$$
$$- (1-\tau)\|x^k - x\|^2 + (1-\tau)\|w^k - x\|^2 - (1-\tau)\|y^k - y\|^2 + (1-\tau)\|u^k - y\|^2$$
$$+ 2\gamma\langle y^{k+1/2}, Ax - z\rangle - 2\gamma\langle Ax^{k+1/2} - z^{k+1/2}, y\rangle$$
$$- 2\gamma(\ell(z^{k+1/2}, b) - \ell(z, b)) - 2\gamma \sum_{i=1}^{n} (r_i(x_i^{k+1/2}) - r_i(x_i))$$
$$- \left( \tau - \frac{1}{2} \right) \|x^{k+1/2} - x^k\|^2 - (1-\tau)\|w^k - x^{k+1/2}\|^2 - \|z^{k+1/2} - z^k\|^2$$
$$- \left( \tau - \frac{1}{2} \right) \|y^{k+1/2} - y^k\|^2 - (1-\tau)\|u^k - y^{k+1/2}\|^2$$
$$+ 4\gamma^2\lambda_{\max}(AA^T)\|y^{k+1/2} - u^k\|^2 + 4\gamma^2 L_r^2\|x^{k+1/2} - x^k\|^2$$
$$+ 2\gamma^2\|y^{k+1/2} - y^k\|^2 + 2\gamma^2 L_\ell^2\|z^{k+1/2} - z^k\|^2$$
$$+ 4\gamma^2\lambda_{\max}(A^T A)\|x^{k+1/2} - w^k\|^2 + 4\gamma^2\|z^{k+1/2} - z^k\|^2$$
$$+ 3\gamma^2\lambda_{\max}(AA^T)\|e^k\|^2 + 3\gamma^2 \left\| \sum_{i=1}^{n} e_i^k \right\|^2.$$

Also here we apply the definition of $\lambda_{\max}(\cdot)$ as a maximum eigenvalue. With Cauchy Schwartz inequality for $n$ summands: $\|\sum_{i=1}^{n} e_i^k\|^2 \leq n \sum_{i=1}^{n} \|e_i^k\|^2$ and after small rearrangements, we obtain

$$2\gamma \left[ \ell(z^{k+1/2}, b) - \ell(z, b)) + \sum_{i=1}^{n} (r_i(x_i^{k+1/2}) - r_i(x_i)) \right.$$

$$+ \langle Ax^{k+1/2} - z^{k+1/2}, y \rangle - \langle Ax - z, y^{k+1/2} \rangle \Big]$$

$$\leq \|\hat{x}^k - x\|^2 + \|z^k - z\|^2 + \|\hat{y}^k - y\|^2$$
$$- \left( \|\hat{x}^{k+1} - x\|^2 + \|z^{k+1} - z\|^2 + \|\hat{y}^{k+1} - y\|^2 \right)$$
$$- (1-\tau)\|x^k - x\|^2 + (1-\tau)\|w^k - x\|^2 - (1-\tau)\|y^k - y\|^2 + (1-\tau)\|u^k - y\|^2$$
$$- \left( \tau - \frac{1}{2} - 4\gamma^2 L_r^2 \right) \|x^{k+1/2} - x^k\|^2 - (1 - \tau - 4\gamma^2 \lambda_{\max}(A^T A))\|w^k - x^{k+1/2}\|^2$$
$$- (1 - 4\gamma^2 - 2\gamma^2 L_\ell^2)\|z^{k+1/2} - z^k\|^2$$
$$- \left( \tau - \frac{1}{2} - 2\gamma^2 \right) \|y^{k+1/2} - y^k\|^2 - (1 - \tau - 4\gamma^2 \lambda_{\max}(AA^T))\|u^k - y^{k+1/2}\|^2$$
$$+ 3\gamma^2 \lambda_{\max}(AA^T)\|e^k\|^2 + 3\gamma^2 n \sum_{i=1}^n \|e_i^k\|^2.$$

Then we sum all over $k$ from 0 to $K-1$, divide by $K$, use Jensen inequality for convex functions $\ell$ and $r_i$ with notation $\bar{x}_i^K = \frac{1}{K}\sum_{k=0}^{K-1} x_i^{k+1/2}$, $\bar{z}^K = \frac{1}{K}\sum_{k=0}^{K-1} z^{k+1/2}$, $\bar{y}^K = \frac{1}{K}\sum_{k=0}^{K-1} y^{k+1/2}$, and have

$$2\gamma \left[ \ell(\bar{z}^K, b) - \ell(z, b) + \sum_{i=1}^n \left( r_i(\bar{x}_i^K) - r_i(x_i) \right) + \langle A\bar{x}^K - \bar{z}^K, y \rangle - \langle Ax - z, \bar{y}^K \rangle \right]$$

$$\leq \frac{1}{K} \left( \|\hat{x}^0 - x\|^2 + \|z^0 - z\|^2 + \|\hat{y}^0 - y\|^2 \right)$$
$$- (1-\tau) \cdot \frac{1}{K} \sum_{k=0}^{K-1} \|x^k - x\|^2 + (1-\tau) \cdot \frac{1}{K} \sum_{k=0}^{K-1} \|w^k - x\|^2$$
$$- (1-\tau) \cdot \frac{1}{K} \sum_{k=0}^{K-1} \|y^k - y\|^2 + (1-\tau) \cdot \frac{1}{K} \sum_{k=0}^{K-1} \|u^k - y\|^2$$
$$- \left( \tau - \frac{1}{2} - 4\gamma^2 L_r^2 \right) \cdot \frac{1}{K} \sum_{k=0}^{K-1} \|x^{k+1/2} - x^k\|^2$$
$$- (1 - \tau - 4\gamma^2 \lambda_{\max}(A^T A)) \cdot \frac{1}{K} \sum_{k=0}^{K-1} \|w^k - x^{k+1/2}\|^2$$
$$- (1 - 4\gamma^2 - 2\gamma^2 L_\ell^2)\frac{1}{K} \sum_{k=0}^{K-1} \|z^{k+1/2} - z^k\|^2$$
$$- \left( \tau - \frac{1}{2} - 2\gamma^2 \right) \cdot \frac{1}{K} \sum_{k=0}^{K-1} \|y^{k+1/2} - y^k\|^2$$
$$- (1 - \tau - 4\gamma^2 \lambda_{\max}(AA^T)) \cdot \frac{1}{K} \sum_{k=0}^{K-1} \|u^k - y^{k+1/2}\|^2$$
$$+ 3\gamma^2 \lambda_{\max}(AA^T) \cdot \frac{1}{K} \sum_{k=0}^{K-1} \|e^k\|^2 + 3\gamma^2 n \cdot \frac{1}{K} \sum_{i=1}^n \sum_{k=0}^{K-1} \|e_i^k\|^2. \qquad (41)$$

Using small rearrangements, we can deduce

$$- (1-\tau) \cdot \frac{1}{K} \sum_{k=0}^{K-1} \|x^k - x\|^2 + (1-\tau) \cdot \frac{1}{K} \sum_{k=0}^{K-1} \|w^k - x\|^2$$
$$= \frac{1}{K} \sum_{k=0}^{K-1} \|w^k - x\|^2 - \frac{1}{K} \sum_{k=0}^{K-1} [(1-\tau)\|x^k - x\|^2 + \tau\|w^k - x\|^2]$$

$$= \frac{1}{K}\|w^0 - x\|^2 - \frac{1}{K}\|w^K - x\|^2$$

$$+ \frac{1}{K}\sum_{k=0}^{K-1}[\|w^{k+1} - x\|^2 - (1-\tau)\|x^k - x\|^2 - \tau\|w^k - x\|^2]$$

$$= \frac{1}{K}\|w^0 - x\|^2 - \frac{1}{K}\|w^K - x\|^2 + \frac{1}{K}\sum_{k=0}^{K-1}[\|w^{k+1}\|^2 - (1-\tau)\|x^k\|^2 - \tau\|w^k\|^2]$$

$$+ \frac{2}{K}\sum_{k=0}^{K-1}\langle (1-\tau)x^k + \tau w^k - w^{k+1}, x\rangle. \tag{42}$$

The same way we can make

$$-(1-\tau)\cdot\frac{1}{K}\sum_{k=0}^{K-1}\|y^k - y\|^2 + (1-\tau)\cdot\frac{1}{K}\sum_{k=0}^{K-1}\|u^k - y\|^2$$

$$= \frac{1}{K}\|u^0 - y\|^2 - \frac{1}{K}\|u^K - y\|^2 + \frac{1}{K}\sum_{k=0}^{K-1}[\|u^{k+1}\|^2 - (1-\tau)\|y^k\|^2 - \tau\|u^k\|^2]$$

$$+ \frac{2}{K}\sum_{k=0}^{K-1}\langle (1-\tau)y^k + \tau u^k - u^{k+1}, y\rangle. \tag{43}$$

Substituting (42) and (43) to (41), we obtain

$$2\gamma\left[\ell(\bar{z}^K, b) - \ell(z, b) + \sum_{i=1}^{n}\left(r_i(\bar{x}_i^K) - r_i(x_i)\right) + \langle A\bar{x}^K - \bar{z}^K, y\rangle - \langle Ax - z, \bar{y}^K\rangle\right]$$

$$\leq \frac{1}{K}\left(\|\hat{x}^0 - x\|^2 + \|z^0 - z\|^2 + \|\hat{y}^0 - y\|^2\right)$$

$$+ \frac{1}{K}\|w^0 - x\|^2 - \frac{1}{K}\|w^K - x\|^2 + \frac{1}{K}\sum_{k=0}^{K-1}[\|w^{k+1}\|^2 - (1-\tau)\|x^k\|^2 - \tau\|w^k\|^2]$$

$$+ \frac{2}{K}\sum_{k=0}^{K-1}\langle (1-\tau)x^k + \tau w^k - w^{k+1}, x\rangle$$

$$+ \frac{1}{K}\|u^0 - y\|^2 - \frac{1}{K}\|u^K - y\|^2 + \frac{1}{K}\sum_{k=0}^{K-1}[\|u^{k+1}\|^2 - (1-\tau)\|y^k\|^2 - \tau\|u^k\|^2]$$

$$+ \frac{2}{K}\sum_{k=0}^{K-1}\langle (1-\tau)y^k + \tau u^k - u^{k+1}, y\rangle$$

$$- \left(\tau - \frac{1}{2} - 4\gamma^2 L_r^2\right)\cdot\frac{1}{K}\sum_{k=0}^{K-1}\|x^{k+1/2} - x^k\|^2$$

$$- (1 - \tau - 4\gamma^2\lambda_{\max}(A^TA))\cdot\frac{1}{K}\sum_{k=0}^{K-1}\|w^k - x^{k+1/2}\|^2$$

$$- (1 - 4\gamma^2 - 2\gamma^2 L_\ell^2)\frac{1}{K}\sum_{k=0}^{K-1}\|z^{k+1/2} - z^k\|^2$$

$$- \left(\tau - \frac{1}{2} - 2\gamma^2\right)\cdot\frac{1}{K}\sum_{k=0}^{K-1}\|y^{k+1/2} - y^k\|^2$$

$$- (1 - \tau - 4\gamma^2\lambda_{\max}(AA^T))\cdot\frac{1}{K}\sum_{k=0}^{K-1}\|u^k - y^{k+1/2}\|^2$$

$$+ 3\gamma^2 \lambda_{\max}(AA^T) \cdot \frac{1}{K} \sum_{k=0}^{K-1} \|e^k\|^2 + 3\gamma^2 n \cdot \frac{1}{K} \sum_{i=1}^{n} \sum_{k=0}^{K-1} \|e_i^k\|^2.$$

As in (15) we pass to the gap criterion by taking the maximum in $y \in \mathcal{Y}$ and the minimum in $x \in \mathcal{X}$ and $z \in \mathcal{Z}$. Additionally, we also take the mathematical expectation

$$2\gamma \mathbb{E}\text{gap}(\bar{x}^K, \bar{z}^K, \bar{y}^K)$$

$$\leq \frac{1}{K}\left( \max_{x \in \mathcal{X}} \|\hat{x}^0 - x\|^2 + \max_{x \in \mathcal{X}} \|w^0 - x\|^2 + \max_{z \in \mathcal{X}} \|z^0 - z\|^2 \right.$$

$$\left. + \max_{y \in \mathcal{Y}} \|\hat{y}^0 - y\|^2 + \max_{y \in \mathcal{Y}} \|u^0 - y\|^2 \right)$$

$$+ \frac{1}{K} \sum_{k=0}^{K-1} \mathbb{E}\|w^{k+1}\|^2 - (1-\tau)\|x^k\|^2 - \tau\|w^k\|^2$$

$$+ \frac{2}{K} \mathbb{E}\max_{x \in \mathcal{X}} \sum_{k=0}^{K-1} \langle (1-\tau)x^k + \tau w^k - w^{k+1}, x \rangle$$

$$+ \frac{1}{K} \sum_{k=0}^{K-1} \mathbb{E}\|u^{k+1}\|^2 - (1-\tau)\|y^k\|^2 - \tau\|u^k\|^2$$

$$+ \frac{2}{K} \mathbb{E}\max_{y \in \mathcal{Y}} \sum_{k=0}^{K-1} \langle (1-\tau)y^k + \tau u^k - u^{k+1}, y \rangle$$

$$- \left( \tau - \frac{1}{2} - 4\gamma^2 L_r^2 \right) \cdot \frac{1}{K} \sum_{k=0}^{K-1} \mathbb{E}\|x^{k+1/2} - x^k\|^2$$

$$- (1 - \tau - 4\gamma^2 \lambda_{\max}(A^T A)) \cdot \frac{1}{K} \sum_{k=0}^{K-1} \mathbb{E}\|w^k - x^{k+1/2}\|^2$$

$$- (1 - 4\gamma^2 - 2\gamma^2 L_\ell^2)\frac{1}{K} \sum_{k=0}^{K-1} \mathbb{E}\|z^{k+1/2} - z^k\|^2$$

$$- \left( \tau - \frac{1}{2} - 2\gamma^2 \right) \cdot \frac{1}{K} \sum_{k=0}^{K-1} \mathbb{E}\|y^{k+1/2} - y^k\|^2$$

$$- (1 - \tau - 4\gamma^2 \lambda_{\max}(AA^T)) \cdot \frac{1}{K} \sum_{k=0}^{K-1} \mathbb{E}\|u^k - y^{k+1/2}\|^2$$

$$+ 3\gamma^2 \lambda_{\max}(AA^T) \cdot \frac{1}{K} \sum_{k=0}^{K-1} \mathbb{E}\|e^k\|^2 + 3\gamma^2 n \cdot \frac{1}{K} \sum_{i=1}^{n} \sum_{k=0}^{K-1} \mathbb{E}\|e_i^k\|^2.$$

Since lines 13–21 of Algorithm 3 are equivalent to lines 11–19 of Algorithm 2. Then, we can use (26), (27), (28), (29) and get

$$2\gamma \mathbb{E}\text{gap}(\bar{x}^K, \bar{z}^K, \bar{y}^K)$$

$$\leq \frac{1}{K}\left( 5\max_{x \in \mathcal{X}} \|\hat{x}^0 - x\|^2 + \max_{x \in \mathcal{X}} \|w^0 - x\|^2 + \max_{z \in \mathcal{X}} \|z^0 - z\|^2 \right.$$

$$\left. + 5\max_{y \in \mathcal{Y}} \|\hat{y}^0 - y\|^2 + \max_{y \in \mathcal{Y}} \|u^0 - y\|^2 \right)$$

$$+ \frac{1}{4} \sum_{k=0}^{K-1} \mathbb{E}\tau(1-\tau)\|w^k - x^k\|^2 + \frac{1}{4} \sum_{k=0}^{K-1} \mathbb{E}\tau(1-\tau)\|u^k - y^k\|^2$$

$$- \left( \tau - \frac{1}{2} - 4\gamma^2 L_r^2 \right) \cdot \frac{1}{K} \sum_{k=0}^{K-1} \mathbb{E}\|x^{k+1/2} - x^k\|^2$$

$$- (1 - \tau - 4\gamma^2 \lambda_{\max}(A^T A)) \cdot \frac{1}{K} \sum_{k=0}^{K-1} \mathbb{E}\|w^k - x^{k+1/2}\|^2$$

$$- (1 - 4\gamma^2 - 2\gamma^2 L_\ell^2) \frac{1}{K} \sum_{k=0}^{K-1} \mathbb{E}\|z^{k+1/2} - z^k\|^2$$

$$- \left(\tau - \frac{1}{2} - 2\gamma^2\right) \cdot \frac{1}{K} \sum_{k=0}^{K-1} \mathbb{E}\|y^{k+1/2} - y^k\|^2$$

$$- (1 - \tau - 4\gamma^2 \lambda_{\max}(AA^T)) \cdot \frac{1}{K} \sum_{k=0}^{K-1} \mathbb{E}\|u^k - y^{k+1/2}\|^2$$

$$+ 3\gamma^2 \lambda_{\max}(AA^T) \cdot \frac{1}{K} \sum_{k=0}^{K-1} \mathbb{E}\|e^k\|^2 + 3\gamma^2 n \cdot \frac{1}{K} \sum_{i=1}^{n} \sum_{k=0}^{K-1} \mathbb{E}\|e_i^k\|^2.$$

Next we work with error feedback terms:

$$\mathbb{E}\|e^{k+1}\|^2 = \mathbb{E}\|y^{k+1/2} - u^k + e^k - C(y^{k+1/2} - u^k + e^k)\|^2$$

$$\leq \left(1 - \frac{1}{\delta}\right) \mathbb{E}\|y^{k+1/2} - u^k + e^k\|^2.$$

With Cauchy Schwartz inequality in the form $\|a + b\|^2 \leq \left(1 + \frac{1}{\eta}\right)\|a\|^2 + (1+\eta)\|b\|^2$ with $a = e^k$, $b = y^{k+1/2} - u^k$ and $\eta = 2\delta$, we get

$$\mathbb{E}\|e^{k+1}\|^2 \leq \left(1 - \frac{1}{\delta}\right)\left(1 + \frac{1}{2\delta}\right) \mathbb{E}\|e^k\|^2 + (2\delta + 1)\left(1 - \frac{1}{\delta}\right) \mathbb{E}\|y^{k+1/2} - u^k\|^2$$

$$\leq \left(1 - \frac{1}{2\delta}\right) \mathbb{E}\|e^k\|^2 + 3\delta \mathbb{E}\|y^{k+1/2} - u^k\|^2.$$

Running the recursion and using that $e_0 = 0$, we have

$$\mathbb{E}\|e^{k+1}\|^2 \leq 3\delta \sum_{j=0}^{k} \left(1 - \frac{1}{2\delta}\right)^{k-j} \mathbb{E}\|y^{j+1/2} - u^j\|^2.$$

Then we sum all over $k$ from 0 to $K - 1$, divide by $K$.

$$\frac{1}{K} \sum_{k=0}^{K-1} \mathbb{E}\|e^k\|^2 \leq 3\delta \cdot \frac{1}{K} \sum_{k=0}^{K-1} \sum_{j=0}^{k-1} \left(1 - \frac{1}{2\delta}\right)^{k-1-j} \mathbb{E}\|y^{j+1/2} - u^j\|^2$$

$$\leq 3\delta \cdot \frac{1}{K} \sum_{k=0}^{K-1} \mathbb{E}\|y^{k+1/2} - u^k\|^2 \sum_{j=0}^{\infty} \left(1 - \frac{1}{2\delta}\right)^j$$

$$\leq 6\delta^2 \cdot \frac{1}{K} \sum_{k=0}^{K-1} \mathbb{E}\|y^{k+1/2} - u^k\|^2. \tag{44}$$

The same way we can make the following estimate:

$$\frac{1}{K} \sum_{k=0}^{K-1} \mathbb{E}\|e_i^k\|^2 \leq 6\delta^2 \cdot \frac{1}{K} \sum_{k=0}^{K-1} \mathbb{E}\|A_i x_i^{k+1/2} - A_i w_i^k\|^2. \tag{45}$$

Combining (41) with (44) and (45), we have

$$2\gamma \mathbb{E}\text{gap}(\bar{x}^K, \bar{z}^K, \bar{y}^K)$$

$$\leq \frac{1}{K}\left(5 \max_{x \in \mathcal{X}} \|\hat{x}^0 - x\|^2 + \max_{x \in \mathcal{X}} \|w^0 - x\|^2 + \max_{z \in \mathcal{X}} \|z^0 - z\|^2\right.$$

$$\left. + 5 \max_{y \in \mathcal{Y}} \|\hat{y}^0 - y\|^2 + \max_{y \in \mathcal{Y}} \|u^0 - y\|^2\right)$$

$$+ \frac{1}{4} \sum_{k=0}^{K-1} \mathbb{E}\tau(1-\tau)\|w^k - x^k\|^2 + \frac{1}{4} \sum_{k=0}^{K-1} \mathbb{E}\tau(1-\tau)\|u^k - y^k\|^2$$

$$- \left(\tau - \frac{1}{2} - 4\gamma^2 L_r^2\right) \cdot \frac{1}{K} \sum_{k=0}^{K-1} \mathbb{E}\|x^{k+1/2} - x^k\|^2$$

$$- (1 - \tau - 4\gamma^2 \lambda_{\max}(A^T A)) \cdot \frac{1}{K} \sum_{k=0}^{K-1} \mathbb{E}\|w^k - x^{k+1/2}\|^2$$

$$- (1 - 4\gamma^2 - 2\gamma^2 L_\ell^2)\frac{1}{K} \sum_{k=0}^{K-1} \mathbb{E}\|z^{k+1/2} - z^k\|^2$$

$$- \left(\tau - \frac{1}{2} - 2\gamma^2\right) \cdot \frac{1}{K} \sum_{k=0}^{K-1} \mathbb{E}\|y^{k+1/2} - y^k\|^2$$

$$- (1 - \tau - 4\gamma^2 \lambda_{\max}(AA^T)) \cdot \frac{1}{K} \sum_{k=0}^{K-1} \mathbb{E}\|u^k - y^{k+1/2}\|^2$$

$$+ 18\gamma^2\delta^2\lambda_{\max}(AA^T) \cdot \frac{1}{K} \sum_{k=0}^{K-1} \mathbb{E}\|y^{k+1/2} - u^k\|^2$$

$$+ 18\gamma^2\delta^2 n \cdot \frac{1}{K} \sum_{k=0}^{K-1} \sum_{i=1}^{n} \mathbb{E}\|A_i x_i^{k+1/2} - A_i w_i^k\|^2.$$

For $\sum_{i=1}^{n} \mathbb{E}\|A_i x_i^{k+1/2} - A_i w_i^k\|^2$ we get

$$\sum_{i=1}^{n} \mathbb{E}\|A_i x_i^{k+1/2} - A_i w_i^k\|^2 \leq \sum_{i=1}^{n} \lambda_{\max}(A_i^T A_i)\mathbb{E}\|x_i^{k+1/2} - w_i^k\|^2$$

$$\leq \max_i \lambda_{\max}(A_i^T A_i) \sum_{i=1}^{n} \mathbb{E}\|x_i^{k+1/2} - w_i^k\|^2$$

$$\leq \max_i[\lambda_{\max}(A_i^T A_i)]\mathbb{E}\|x^{k+1/2} - w^k\|^2.$$

Then one can deduce

$$2\gamma\mathbb{E}\text{gap}(\bar{x}^K, \bar{z}^K, \bar{y}^K)$$

$$\leq \frac{1}{K}\left(5 \max_{x \in \mathcal{X}} \|\hat{x}^0 - x\|^2 + \max_{x \in \mathcal{X}} \|w^0 - x\|^2 + \max_{z \in \mathcal{X}} \|z^0 - z\|^2\right.$$

$$\left. + 5 \max_{y \in \mathcal{Y}} \|\hat{y}^0 - y\|^2 + \max_{y \in \mathcal{Y}} \|u^0 - y\|^2\right)$$

$$+ \frac{1}{4} \sum_{k=0}^{K-1} \mathbb{E}\tau(1-\tau)\|w^k - x^k\|^2 + \frac{1}{4} \sum_{k=0}^{K-1} \mathbb{E}\tau(1-\tau)\|u^k - y^k\|^2$$

$$- \left(\tau - \frac{1}{2} - 4\gamma^2 L_r^2\right) \cdot \frac{1}{K} \sum_{k=0}^{K-1} \mathbb{E}\|x^{k+1/2} - x^k\|^2$$

$$- (1 - \tau - 4\gamma^2 \lambda_{\max}(A^T A) - 18\gamma^2\delta^2 n \max_i[\lambda_{\max}(A_i^T A_i)]) \cdot \frac{1}{K} \sum_{k=0}^{K-1} \mathbb{E}\|w^k - x^{k+1/2}\|^2$$

$$- (1 - 4\gamma^2 - 2\gamma^2 L_\ell^2)\frac{1}{K} \sum_{k=0}^{K-1} \mathbb{E}\|z^{k+1/2} - z^k\|^2$$

$$- \left(\tau - \frac{1}{2} - 2\gamma^2\right) \cdot \frac{1}{K} \sum_{k=0}^{K-1} \mathbb{E}\|y^{k+1/2} - y^k\|^2$$

$$- (1 - \tau - 4\gamma^2\lambda_{\max}(AA^T)(1 + 5\delta^2)) \cdot \frac{1}{K} \sum_{k=0}^{K-1} \mathbb{E}\|u^k - y^{k+1/2}\|^2.$$

With $\tau \leq 1$ and Cauchy Schwartz inequality, we have

$$2\gamma\mathbb{E}\text{gap}(\bar{x}^K, \bar{z}^K, \bar{y}^K)$$

$$\leq \frac{1}{K}\left( 5\max_{x\in\mathcal{X}}\|\hat{x}^0 - x\|^2 + \max_{x\in\mathcal{X}}\|w^0 - x\|^2 + \max_{z\in\mathcal{X}}\|z^0 - z\|^2 \right.$$

$$\left. + 5\max_{y\in\mathcal{Y}}\|\hat{y}^0 - y\|^2 + \max_{y\in\mathcal{Y}}\|u^0 - y\|^2 \right)$$

$$- \left( \frac{3\tau - 2}{2} - 4\gamma^2 L_r^2 \right) \cdot \frac{1}{K} \sum_{k=0}^{K-1} \mathbb{E}\|x^{k+1/2} - x^k\|^2$$

$$- \left( \frac{1 - \tau}{2} - 4\gamma^2\lambda_{\max}(A^T A) - 18\gamma^2\delta^2 n \max_i[\lambda_{\max}(A_i^T A_i)] \right) \cdot \frac{1}{K} \sum_{k=0}^{K-1} \mathbb{E}\|w^k - x^{k+1/2}\|^2$$

$$- (1 - 4\gamma^2 - 2\gamma^2 L_\ell^2)\frac{1}{K} \sum_{k=0}^{K-1} \mathbb{E}\|z^{k+1/2} - z^k\|^2$$

$$- \left( \frac{3\tau - 2}{2} - 2\gamma^2 \right) \cdot \frac{1}{K} \sum_{k=0}^{K-1} \mathbb{E}\|y^{k+1/2} - y^k\|^2$$

$$- \left( \frac{1 - \tau}{2} - 4\gamma^2\lambda_{\max}(AA^T)(1 + 5\delta^2) \right) \cdot \frac{1}{K} \sum_{k=0}^{K-1} \mathbb{E}\|u^k - y^{k+1/2}\|^2.$$

If we choose $\tau \geq \frac{1}{2}$ and $\gamma$ as follows

$$\gamma \leq \frac{1}{4}\min\left\{ 1; \frac{1}{L_r}; \frac{1}{L_\ell}; \sqrt{\frac{1 - \tau}{5\delta^2 n \max_i[\lambda_{\max}(A_i^T A_i)]}}; \sqrt{\frac{1 - \tau}{3\delta^2\lambda_{\max}(AA^T)}}; \right\},$$

then one can obtain

$$\mathbb{E}\text{gap}(\bar{x}^K, \bar{z}^K, \bar{y}^K) \leq \frac{1}{2\gamma K}\left( 5\max_{x\in\mathcal{X}}\|x^0 - x\|^2 + \max_{x\in\mathcal{X}}\|w^0 - x\|^2 + \max_{z\in\mathcal{Z}}\|z^0 - z\|^2 \right.$$

$$\left. + 5\max_{y\in\mathcal{Y}}\|y^0 - y\|^2 + \max_{y\in\mathcal{Y}}\|u^0 - y\|^2 \right).$$

With $\gamma = \frac{1}{4}\min\left\{ 1; \frac{1}{L_r}; \frac{1}{L_\ell}; \sqrt{\frac{1-\tau}{5\delta^2 n \max_i[\lambda_{\max}(A_i^T A_i)]}}; \sqrt{\frac{1-\tau}{3\delta^2\lambda_{\max}(AA^T)}}; \right\}$, we finish the proof. $\quad\square$

## D.5 PROOF OF THEOREM 3.5

**Theorem D.6** (Theorem 3.5). *Let Assumption 2.1 holds. Let problem (4) be solved by Algorithm 4 (Appendix A). Then for $\tau = 1 - p$ and*

$$\gamma = \tfrac{1}{4}\min\left\{ 1; \tfrac{1}{L_r}; \tfrac{1}{L_\ell}; \sqrt{\tfrac{1-\tau}{\lambda_{\max}(AA^T) + n\cdot\max_i\{\lambda_{\max}(A_i A_i^T)\}}} \right\},$$

*it holds that*

$$\mathbb{E}gap(\bar{x}^K, \bar{z}^K, \bar{y}^K) = \mathcal{O}\left( \left[ 1 + \tfrac{1}{\sqrt{p}}\left( \sqrt{\lambda_{\max}(AA^T)} + n\cdot\max_{i=1,\ldots,n}\left\{ \sqrt{\lambda_{\max}(A_i A_i^T)} \right\} \right) + L_\ell + L_r \right] \cdot \tfrac{D^2}{K} \right),$$

*where $\bar{x}^K := \frac{1}{K}\sum_{k=0}^{K-1} x^{k+1/2}$, $\bar{z}^K := \frac{1}{K}\sum_{k=0}^{K-1} z^{k+1/2}$, $\bar{y}^K := \frac{1}{K}\sum_{k=0}^{K-1} y^{k+1/2}$ and $D^2 := \max_{x,z,y\in\mathcal{X},\mathcal{Z},\mathcal{Y}}\left[ \|x^0 - x\|^2 + \|z^0 - z\|^2 + \|y^0 - y\|^2 \right]$.*

*Proof.* The proof repeats almost the same steps as the proof of Theorem 3.2 (Section D.3). In particular, in the proof of Theorem 3.2 we need to replace $Q(y^{k+1/2} - u^k)$ by $y^{k+1/2} - u^k$ and

$\sum_{i=1}^n Q(A_i[x_i^{k+1/2} - w_i^k])$ by $n \cdot A_{i_k}[x_{i_k}^{k+1/2} - w_{i_k}^k]$, and use. In the end, we arrive at the analogue of (32).

$$2\gamma\mathbb{E}\text{gap}(\bar{x}^K, \bar{z}^K, \bar{y}^K)$$

$$\leq \frac{1}{K}\left(6\max_{x\in\mathcal{X}}\|x^0 - x\|^2 + \max_{x\in\mathcal{X}}\|w^0 - x\|^2 + \max_{z\in\mathcal{Z}}\|z^0 - z\|^2\right.$$

$$\left. + 6\max_{y\in\mathcal{Y}}\|y^0 - y\|^2 + \max_{y\in\mathcal{Y}}\|u^0 - y\|^2\right)$$

$$- \left(\frac{3\tau - 1}{2} - 2\gamma^2 L_r^2\right) \cdot \frac{1}{K}\sum_{k=0}^{K-1}\mathbb{E}\|x^{k+1/2} - x^k\|^2 - \frac{1-\tau}{2K}\sum_{k=0}^{K-1}\mathbb{E}\|w^k - x^{k+1/2}\|^2$$

$$- \left(1 - 2\gamma^2(1 + L_\ell^2)\right)\frac{1}{K}\sum_{k=0}^{K-1}\mathbb{E}\|z^{k+1/2} - z^k\|^2$$

$$- \left(\frac{3\tau - 1}{2} - 2\gamma^2\right)\frac{1}{K}\sum_{k=0}^{K-1}\mathbb{E}\|y^{k+1/2} - y^k\|^2 - \frac{1-\tau}{2K}\sum_{k=0}^{K-1}\mathbb{E}\|u^k - y^{k+1/2}\|^2$$

$$+ \frac{3\gamma^2}{K}\sum_{k=0}^{K-1}\mathbb{E}\|A^T(y^{k+1/2} - u^k)\|^2 + \frac{3\gamma^2 n^2}{K}\sum_{k=0}^{K-1}\mathbb{E}\|A_{i_k}(x_{i_k}^{k+1/2} - w_{i_k}^k)\|^2.$$

Using the notation of $\lambda_{\max}(\cdot)$ as a maximum eigenvalue and the random choice of $i_k$, we get

$$\mathbb{E}\|A_{i_k}(x_{i_k}^{k+1/2} - w_{i_k}^k)\|^2 = \mathbb{E}\mathbb{E}_{i_k}\left[\|A_{i_k}(x_{i_k}^{k+1/2} - w_{i_k}^k)\|^2\right]$$

$$= \frac{1}{n}\sum_{i=1}^n\mathbb{E}\|A_i(x_i^{k+1/2} - w_i^k)\|^2$$

$$\leq \frac{1}{n}\sum_{i=1}^n\lambda_{\max}(A_i^T A_i)\mathbb{E}\|x_i^{k+1/2} - w_i^k\|^2$$

$$\leq \frac{\max_i\{\lambda_{\max}(A_i^T A_i)\}}{n}\mathbb{E}\|x^{k+1/2} - w^k\|^2.$$

Therefore, we obtain

$$2\gamma\mathbb{E}\text{gap}(\bar{x}^K, \bar{z}^K, \bar{y}^K)$$

$$\leq \frac{1}{K}\left(6\max_{x\in\mathcal{X}}\|x^0 - x\|^2 + \max_{x\in\mathcal{X}}\|w^0 - x\|^2 + \max_{z\in\mathcal{Z}}\|z^0 - z\|^2\right.$$

$$\left. + 6\max_{y\in\mathcal{Y}}\|y^0 - y\|^2 + \max_{y\in\mathcal{Y}}\|u^0 - y\|^2\right)$$

$$- \left(\frac{3\tau - 1}{2} - 2\gamma^2 L_r^2\right) \cdot \frac{1}{K}\sum_{k=0}^{K-1}\mathbb{E}\|x^{k+1/2} - x^k\|^2$$

$$- \left(\frac{1-\tau}{2} - 3\gamma^2 n\max_i\{\lambda_{\max}(A_i^T A_i)\}\right)\frac{1}{K}\sum_{k=0}^{K-1}\mathbb{E}\|w^k - x^{k+1/2}\|^2$$

$$- \left(1 - 2\gamma^2(1 + L_\ell^2)\right)\frac{1}{K}\sum_{k=0}^{K-1}\mathbb{E}\|z^{k+1/2} - z^k\|^2$$

$$- \left(\frac{3\tau - 1}{2} - 2\gamma^2\right)\frac{1}{K}\sum_{k=0}^{K-1}\mathbb{E}\|y^{k+1/2} - y^k\|^2$$

$$- \left(\frac{1-\tau}{2} - 3\lambda_{\max}(AA^T)\gamma^2\right)\frac{1}{K}\sum_{k=0}^{K-1}\mathbb{E}\|u^k - y^{k+1/2}\|^2.$$

If we choose $\tau \geq \frac{1}{2}$ and $\gamma$ as follows

$$\gamma \leq \frac{1}{4} \min \left\{ 1; \frac{1}{L_r}; \frac{1}{L_\ell}; \sqrt{\frac{1-\tau}{n \max_i \{\lambda_{\max}(A_i^T A_i)\}}}; \sqrt{\frac{1-\tau}{\lambda_{\max}(AA^T)}}; \right\},$$

then one can obtain

$$\mathbb{E}\mathrm{gap}(\bar{x}^K, \bar{z}^K, \bar{y}^K) \leq \frac{1}{2\gamma K} \left( 6 \max_{x \in \mathcal{X}} \|x^0 - x\|^2 + \max_{x \in \mathcal{X}} \|w^0 - x\|^2 + \max_{z \in \mathcal{Z}} \|z^0 - z\|^2 \right.$$

$$\left. + 6 \max_{y \in \mathcal{Y}} \|y^0 - y\|^2 + \max_{y \in \mathcal{Y}} \|u^0 - y\|^2 \right).$$

With $\gamma = \frac{1}{4} \min \left\{ 1; \frac{1}{L_r}; \frac{1}{L_\ell}; \sqrt{\frac{1-\tau}{n \max_i \{\lambda_{\max}(A_i^T A_i)\}}}; \sqrt{\frac{1-\tau}{\lambda_{\max}(AA^T)}}; \right\}$, we finish the proof. $\square$

### D.6  PROOF OF THEOREM 3.6

**Theorem D.7** (Theorem 3.6). *Let Assumption 2.1 holds. Let problem (4) be solved by Algorithm 5 (Appendix A). Then for $\tau = 1 - p$ and*

$$\gamma = \frac{1}{4} \min \left\{ 1; \frac{1}{L_r}; \frac{1}{L_\ell}; \sqrt{\frac{1-\tau}{s(\lambda_{\max}(A^T A) + \mathbb{I}(\textit{diff. seed}) \max_i \{\lambda_{\max}(A_i^T A_i)\})}}; \sqrt{\frac{1-\tau}{d \max_i \{\lambda_{\max}(A_i^T A_i)\}}}; \right\},$$

*it holds that*

$$\mathbb{E}\mathrm{gap}(\bar{x}^K, \bar{z}^K, \bar{y}^K) = \mathcal{O}\left( \left[ \frac{1}{\sqrt{p}} \left( s \sqrt{\lambda_{\max}(A^T A) + \mathbb{I}(\textit{diff. seed}) \max_{i=1,\dots,n} \{\lambda_{\max}(A_i^T A_i)\}} + \right) + L_\ell + L_r \right] \cdot \frac{D^2}{K} \right.$$

$$\left. + \left[ \frac{1}{\sqrt{p}} \left( d \cdot \max_{i=1,\dots,n} \left\{ \sqrt{\lambda_{\max}(A_i A_i^T)} \right\} \right) \right] \cdot \frac{D^2}{K} \right),$$

*where $\bar{x}^K := \frac{1}{K} \sum_{k=0}^{K-1} x^{k+1/2}$, $\bar{z}^K := \frac{1}{K} \sum_{k=0}^{K-1} z^{k+1/2}$, $\bar{y}^K := \frac{1}{K} \sum_{k=0}^{K-1} y^{k+1/2}$ and $D^2 := \max_{x,z,y \in \mathcal{X}, \mathcal{Z}, \mathcal{Y}} \left[ \|x^0 - x\|^2 + \|z^0 - z\|^2 + \|y^0 - y\|^2 \right]$.*

*Proof.* Most of the proof is the same as that of Theorem 3.2. We note only some main steps of the proof and changes regarding Section D.4 with the proof of Theorem 3.2. We start with an analogue of (19) and get

$$\|x_i^{k+1} - x_i\|^2 = \|x_i^k - x_i\|^2 - \|x_i^{k+1/2} - x_i^k\|^2 - \|x_i^{k+1/2} - x_i^{k+1}\|^2$$

$$+ (1-\tau)(\|w_i^k - x_i\|^2 - \|w_i^k - x_i^{k+1/2}\|^2 - \|x_i^{k+1/2} - x_i\|^2)$$

$$+ (1-\tau)(\|x_i^{k+1/2} - x_i^k\|^2 + \|x_i^{k+1/2} - x_i\|^2 - \|x_i^k - x_i\|^2)$$

$$- 2\gamma \langle d_i \cdot \langle A_i^T (y^{k+1/2} - u^k), e_{j_i^k} \rangle e_{j_i^k} + A_i^T u^k + \nabla r_i(x_i^{k+1/2}), x_i^{k+1/2} - x_i \rangle$$

$$+ 2\gamma \langle d_i \cdot \langle A_i^T (y^{k+1/2} - u^k), e_{j_i^k} \rangle e_{j_i^k} + \nabla r_i(x_i^{k+1/2}) - \nabla r_i(x_i^k), x_i^{k+1/2} - x_i^{k+1} \rangle$$

$$= \tau \|x_i^k - x_i\|^2 + (1-\tau)\|w_i^k - x_i\|^2$$

$$- \tau \|x_i^{k+1/2} - x_i^k\|^2 - (1-\tau)\|w_i^k - x_i^{k+1/2}\|^2 - \|x_i^{k+1/2} - x_i^{k+1}\|^2$$

$$- 2\gamma \langle d_i \cdot \langle A_i^T (y^{k+1/2} - u^k), e_{j_i^k} \rangle e_{j_i^k} + A_i^T u^k + \nabla r_i(x_i^{k+1/2}), x_i^{k+1/2} - x_i \rangle$$

$$+ 2\gamma \langle d_i \cdot \langle A_i^T (y^{k+1/2} - u^k), e_{j_i^k} \rangle e_{j_i^k} + \nabla r_i(x_i^{k+1/2}) - \nabla r_i(x_i^k), x_i^{k+1/2} - x_i^{k+1} \rangle.$$

Summing over all $i$ from 1 to $n$ and using the notation of $A = [A_1, \dots, A_i, \dots, A_n]$, $x = [x_1^T, \dots, x_i^T, \dots, x_n^T]^T$, $w = [w_1^T, \dots, w_i^T, \dots, w_n^T]^T$, we deduce

$$\|x^{k+1} - x\|^2 = \tau \|x^k - x\|^2 + (1-\tau)\|w^k - x\|^2$$

$$- \tau \|x^{k+1/2} - x^k\|^2 - (1-\tau)\|w^k - x^{k+1/2}\|^2 - \|x^{k+1/2} - x^{k+1}\|^2$$

$$- 2\gamma \langle A(x^{k+1/2} - x), y^{k+1/2} \rangle - 2\gamma \sum_{i=1}^n \langle \nabla r_i(x_i^{k+1/2}), x_i^{k+1/2} - x_i \rangle$$

$$- 2\gamma \sum_{i=1}^{n} \langle d_i \cdot \langle A_i^T(y^{k+1/2} - u^k), e_{j_i^k} \rangle e_{j_i^k} - A_i^T(y^{k+1/2} - u^k), x_i^{k+1/2} - x_i \rangle$$

$$+ 2\gamma \sum_{i=1}^{n} \langle d_i \cdot \langle A_i^T(y^{k+1/2} - u^k), e_{j_i^k} \rangle e_{j_i^k}, x_i^{k+1/2} - x_i^{k+1} \rangle$$

$$+ 2\gamma \sum_{i=1}^{n} \langle \nabla r_i(x_i^{k+1/2}) - \nabla r_i(x_i^k), x_i^{k+1/2} - x_i^{k+1} \rangle.$$

By simple fact: $2\langle a, b \rangle \le \eta \|a\|^2 + \frac{1}{\eta}\|b\|^2$ with $a = d_i \cdot [A_i^T(y^{k+1/2} - u^k)]_{(j_i^k)}$, $b = x_i^{k+1/2} - x_i^{k+1}$, $\eta = 2\gamma$ and $a = \nabla r_i(x_i^{k+1/2}) - \nabla r_i(x_i^k)$, $b = x_i^{k+1/2} - x_i^{k+1}$, $\eta = 2\gamma$, we get

$$\|x^{k+1} - x\|^2 = \tau \|x^k - x\|^2 + (1 - \tau)\|w^k - x\|^2$$

$$- \tau \|x^{k+1/2} - x^k\|^2 - (1 - \tau)\|w^k - x^{k+1/2}\|^2 - \|x^{k+1/2} - x^{k+1}\|^2$$

$$- 2\gamma \langle A(x^{k+1/2} - x), y^{k+1/2} \rangle - 2\gamma \sum_{i=1}^{n} \langle \nabla r_i(x_i^{k+1/2}), x_i^{k+1/2} - x_i \rangle$$

$$- 2\gamma \sum_{i=1}^{n} \langle d_i \cdot \langle A_i^T(y^{k+1/2} - u^k), e_{j_i^k} \rangle e_{j_i^k} - A_i^T(y^{k+1/2} - u^k), x_i^{k+1/2} - x_i \rangle$$

$$+ 2\gamma^2 \sum_{i=1}^{n} \|d_i \cdot \langle A_i^T(y^{k+1/2} - u^k), e_{j_i^k} \rangle e_{j_i^k}\|^2 + \frac{1}{2}\|x^{k+1/2} - x^{k+1}\|^2$$

$$+ 2\gamma^2 \sum_{i=1}^{n} \|\nabla r_i(x_i^{k+1/2}) - \nabla r_i(x_i^k)\|^2 + \frac{1}{2}\|x^{k+1/2} - x^{k+1}\|^2$$

$$= \tau \|x^k - x\|^2 + (1 - \tau)\|w^k - x\|^2$$

$$- \tau \|x^{k+1/2} - x^k\|^2 - (1 - \tau)\|w^k - x^{k+1/2}\|^2$$

$$- 2\gamma \langle A(x^{k+1/2} - x), y^{k+1/2} \rangle - 2\gamma \sum_{i=1}^{n} \langle \nabla r_i(x_i^{k+1/2}), x_i^{k+1/2} - x_i \rangle$$

$$- 2\gamma \sum_{i=1}^{n} \langle d_i \cdot \langle A_i^T(y^{k+1/2} - u^k), e_{j_i^k} \rangle e_{j_i^k} - A_i^T(y^{k+1/2} - u^k), x_i^{k+1/2} - x_i \rangle$$

$$+ 2\gamma^2 \sum_{i=1}^{n} \|d_i \cdot \langle A_i^T(y^{k+1/2} - u^k), e_{j_i^k} \rangle e_{j_i^k}\|^2$$

$$+ 2\gamma^2 \sum_{i=1}^{n} \|\nabla r_i(x_i^{k+1/2}) - \nabla r_i(x_i^k)\|^2.$$

The analogue of (20) is

$$\|x^{k+1} - x\|^2 + \|w^{k+1} - x\|^2$$

$$\le \|x^k - x\|^2 + \|w^k - x\|^2$$

$$- \tau \|x^{k+1/2} - x^k\|^2 - (1 - \tau)\|w^k - x^{k+1/2}\|^2$$

$$- 2\gamma \langle A(x^{k+1/2} - x), y^{k+1/2} \rangle - 2\gamma \sum_{i=1}^{n} \langle \nabla r_i(x_i^{k+1/2}), x_i^{k+1/2} - x_i \rangle$$

$$- (1 - \tau)\|x^k\|^2 - \tau \|w^k\|^2 + \|w^{k+1}\|^2$$

$$+ 2\langle (1 - \tau)x^k + \tau w^k - w^{k+1}, x \rangle$$

$$- 2\gamma \sum_{i=1}^{n} \langle d_i \cdot \langle A_i^T(y^{k+1/2} - u^k), e_{j_i^k} \rangle e_{j_i^k} - A_i^T(y^{k+1/2} - u^k), x_i^{k+1/2} - x_i^0 \rangle$$

$$- 2\gamma \sum_{i=1}^{n} \langle d_i \cdot \langle A_i^T(y^{k+1/2} - u^k), e_{j_i^k} \rangle e_{j_i^k} - A_i^T(y^{k+1/2} - u^k), x_i^0 - x_i \rangle$$

$$+ 2\gamma^2 \sum_{i=1}^{n} \|d_i \cdot \langle A_i^T(y^{k+1/2} - u^k), e_{j_i^k}\rangle e_{j_i^k}\|^2$$

$$+ 2\gamma^2 \sum_{i=1}^{n} \|\nabla r_i(x_i^{k+1/2}) - \nabla r_i(x_i^k)\|^2.$$

(21) is absolutely the same. The analogue of (22) is

$$\|y^{k+1} - y\|^2 + \|u^{k+1} - y\|^2$$

$$\leq \|y^k - y\|^2 + \|u^k - y\|^2$$

$$- \tau\|y^{k+1/2} - y^k\|^2 - (1-\tau)\|u^k - y^{k+1/2}\|^2$$

$$- 2\gamma\langle z^{k+1/2}, y^{k+1/2} - y\rangle + 2\gamma\langle \sum_{i=1}^{n} A_i x_i^{k+1/2}, y^{k+1/2} - y\rangle$$

$$+ 2\gamma\langle \sum_{i=1}^{n} [s \cdot \langle A_i(x_i^{k+1/2} - w_i^k), e_{c_i^k}\rangle e_{c_i^k} + A_i w_i^k - A_i x_i^{k+1/2}], y^{k+1/2} - y^0\rangle$$

$$+ 2\gamma\langle \sum_{i=1}^{n} [s \cdot \langle A_i(x_i^{k+1/2} - w_i^k), e_{c_i^k}\rangle e_{c_i^k} + A_i w_i^k - A_i x_i^{k+1/2}], y^0 - y\rangle$$

$$- (1-\tau)\|y^k\|^2 - \tau\|u^k\|^2 + \|y^{k+1}\|^2$$

$$+ 2\langle (1-\tau)y^k + \tau u^k - u^{k+1}, y\rangle$$

$$+ 2\gamma^2 \|\sum_{i=1}^{n} s \cdot \langle A_i(x_i^{k+1/2} - w_i^k), e_{c_i^k}\rangle e_{c_i^k}\|^2 + 2\gamma^2\|z^{k+1/2} - z^k\|^2.$$

The analogue of (23) is

$$2\gamma\mathbb{E}\mathrm{gap}(\bar{x}^K, \bar{z}^K, \bar{y}^K)$$

$$\leq \frac{1}{K}\left(\max_{x\in\mathcal{X}}\|x^0 - x\|^2 + \max_{x\in\mathcal{X}}\|w^0 - x\|^2 + \max_{z\in\mathcal{Z}}\|z^0 - z\|^2\right.$$

$$\left. + \max_{y\in\mathcal{Y}}\|y^0 - y\|^2 + \max_{y\in\mathcal{Y}}\|u^0 - y\|^2\right)$$

$$- \frac{\tau}{K}\sum_{k=0}^{K-1}\mathbb{E}\|x^{k+1/2} - x^k\|^2 - \frac{1-\tau}{K}\sum_{k=0}^{K-1}\mathbb{E}\|w^k - x^{k+1/2}\|^2$$

$$- \frac{1}{K}\sum_{k=0}^{K-1}\mathbb{E}\|z^{k+1/2} - z^k\|^2$$

$$- \frac{\tau}{K}\sum_{k=0}^{K-1}\mathbb{E}\|y^{k+1/2} - y^k\|^2 - \frac{1-\tau}{K}\sum_{k=0}^{K-1}\mathbb{E}\|u^k - y^{k+1/2}\|^2$$

$$+ \frac{1}{K}\sum_{k=0}^{K-1}\mathbb{E}\|w^{k+1}\|^2 - (1-\tau)\|x^k\|^2 - \tau\|w^k\|^2$$

$$+ \frac{2}{K}\mathbb{E}\max_{x\in\mathcal{X}}\sum_{k=0}^{K-1}\langle (1-\tau)x^k + \tau w^k - w^{k+1}, x\rangle$$

$$+ \frac{1}{K}\sum_{k=0}^{K-1}\mathbb{E}\|y^{k+1}\|^2 - (1-\tau)\|y^k\|^2 - \tau\|u^k\|^2$$

$$+ \frac{2}{K}\mathbb{E}\max_{y\in\mathcal{Y}}\sum_{k=0}^{K-1}\langle (1-\tau)y^k + \tau u^k - u^{k+1}, y\rangle$$

$$- \frac{2\gamma}{K}\sum_{k=0}^{K-1}\sum_{i=1}^{n}\mathbb{E}\langle d_i \cdot [A_i^T(y^{k+1/2} - u^k)]_{(j_i^k)} - A_i^T(y^{k+1/2} - u^k), x_i^{k+1/2} - x_i^0\rangle$$

$$+ \frac{2\gamma}{K} \cdot \mathbb{E}\max_{x \in \mathcal{X}} \sum_{k=0}^{K-1} \sum_{i=1}^{n} \langle d_i \cdot [A_i^T(y^{k+1/2} - u^k)]_{(j_i^k)} - A_i^T(y^{k+1/2} - u^k), x_i - x_i^0 \rangle$$

$$+ \frac{2\gamma}{K} \sum_{k=0}^{K-1} \mathbb{E}\langle \sum_{i=1}^{n}[s \cdot \langle A_i(x_i^{k+1/2} - w_i^k), e_{c_i^k}\rangle e_{c_i^k} + A_i w_i^k - A_i x_i^{k+1/2}], y^{k+1/2} - y^0 \rangle$$

$$+ \frac{2\gamma}{K} \cdot \mathbb{E}\max_{y \in \mathcal{Y}} \sum_{k=0}^{K-1} \langle \sum_{i=1}^{n}[s \cdot \langle A_i(x_i^{k+1/2} - w_i^k), e_{c_i^k}\rangle e_{c_i^k} + A_i w_i^k - A_i x_i^{k+1/2}], y^0 - y \rangle$$

$$+ \frac{2\gamma^2}{K} \sum_{k=0}^{K-1} \sum_{i=1}^{n} \mathbb{E}\|d_i \cdot [A_i^T(y^{k+1/2} - u^k)]_{(j_i^k)}\|^2$$

$$+ \frac{2\gamma^2}{K} \sum_{k=0}^{K-1} \mathbb{E}\| \sum_{i=1}^{n} s \cdot \langle A_i(x_i^{k+1/2} - w_i^k), e_{c_i^k}\rangle e_{c_i^k}\|^2$$

$$+ \frac{2\gamma^2 L_r^2}{K} \sum_{k=0}^{K-1} \mathbb{E}\|x^{k+1/2} - x^k\|^2 + \frac{2\gamma^2}{K} \sum_{k=0}^{K-1} \mathbb{E}\|y^{k+1/2} - y^k\|^2$$

$$+ \frac{2\gamma^2}{K} \sum_{k=0}^{K-1} \mathbb{E}\|z^{k+1/2} - z^k\|^2 + \frac{2\gamma^2 L_\ell^2}{K} \sum_{k=0}^{K-1} \mathbb{E}\|z^{k+1/2} - z^k\|^2.$$

(24), (25), (26), (27) are absolutely the same. The analogue of (28) is

$$\mathbb{E}\langle d_i \cdot \langle A_i^T(y^{k+1/2} - u^k), e_{j_i^k}\rangle e_{j_i^k} - A_i^T(y^{k+1/2} - u^k), x_i^{k+1/2} - x_i^0 \rangle$$
$$= \mathbb{E}\langle \mathbb{E}_{j_i^k}\left[ d_i \cdot \langle A_i^T(y^{k+1/2} - u^k), e_{j_i^k}\rangle e_{j_i^k}\right] - A_i^T(y^{k+1/2} - u^k), x_i^{k+1/2} - x_i^0 \rangle = 0.$$

The analogue of (29) is

$$\mathbb{E}\langle \sum_{i=1}^{n}[s \cdot \langle A_i(x_i^{k+1/2} - w_i^k), e_{c_i^k}\rangle e_{c_i^k} + A_i w_i^k - A_i x_i^{k+1/2}], y^{k+1/2} - y^0 \rangle$$
$$= \mathbb{E}\langle \sum_{i=1}^{n} \mathbb{E}_{c_i^k}\left[ s \cdot \langle A_i(x_i^{k+1/2} - w_i^k), e_{c_i^k}\rangle e_{c_i^k}\right] + A_i w_i^k - A_i x_i^{k+1/2}, y^{k+1/2} - y^0 \rangle = 0.$$

The analogue of (30) is

$$\mathbb{E}\max_{x \in \mathcal{X}} \sum_{i=1}^{n} \sum_{k=0}^{K-1} \langle d_i \cdot \langle A_i^T(y^{k+1/2} - u^k), e_{j_i^k}\rangle e_{j_i^k} - A_i^T(y^{k+1/2} - u^k), x_i - x_i^0 \rangle$$

$$\leq \mathbb{E}\max_{x \in \mathcal{X}} \frac{1}{2\gamma} \sum_{i=1}^{n} \|x_i^0 - x_i\|^2$$

$$+ \mathbb{E}\frac{\gamma}{2} \sum_{i=1}^{n} \| \sum_{k=0}^{K-1} d_i \cdot \langle A_i^T(y^{k+1/2} - u^k), e_{j_i^k}\rangle e_{j_i^k} - A_i^T(y^{k+1/2} - u^k)\|^2$$

$$= \mathbb{E}\max_{x \in \mathcal{X}} \frac{1}{2\gamma} \|x^0 - x\|^2 + \mathbb{E}\frac{\gamma}{2} \sum_{i=1}^{n} \sum_{k=0}^{K-1} \|d_i \cdot \langle A_i^T(y^{k+1/2} - u^k), e_{j_i^k}\rangle e_{j_i^k} - A_i^T(y^{k+1/2} - u^k)\|^2$$

$$+ \mathbb{E}\Bigg[ \gamma \sum_{i=1}^{n} \sum_{k_1 < k_2} \langle d_i \cdot \langle A_i^T(y^{k+1/2} - u^k), e_{j_i^k}\rangle e_{j_i^k} - A_i^T(y^{k_1+1/2} - u^{k_1}),$$

$$d_i \cdot [A_i^T(y^{k_2+1/2} - u^{k_2})]_{(j_i^{k_2})} - A_i^T(y^{k_2+1/2} - u^k)\rangle \Bigg]$$

$$= \mathbb{E}\max_{x \in \mathcal{X}} \frac{1}{2\gamma} \|x^0 - x\|^2 + \mathbb{E}\frac{\gamma}{2} \sum_{i=1}^{n} \sum_{k=0}^{K-1} \|d_i \cdot \langle A_i^T(y^{k+1/2} - u^k), e_{j_i^k}\rangle e_{j_i^k} - A_i^T(y^{k+1/2} - u^k)\|^2$$

$$+ \mathbb{E}\Bigg[\gamma \sum_{i=1}^{n} \sum_{k_1 < k_2} \langle d_i \cdot \langle A_i^T(y^{k+1/2} - u^k), e_{j_i^k}\rangle e_{j_i^k} - A_i^T(y^{k_1+1/2} - u^{k_1}),$$

$$\mathbb{E}_{j_i^{k_2}}\big[d_i \cdot [A_i^T(y^{k_2+1/2} - u^{k_2})]_{(j_i^{k_2})}\big] - A_i^T(y^{k_2+1/2} - u^k)\rangle\Bigg]$$

$$= \mathbb{E}\max_{x \in \mathcal{X}} \frac{1}{2\gamma}\|x^0 - x\|^2 + \mathbb{E}\frac{\gamma}{2}\sum_{i=1}^{n}\sum_{k=0}^{K-1}\|d_i \cdot \langle A_i^T(y^{k+1/2} - u^k), e_{j_i^k}\rangle e_{j_i^k} - A_i^T(y^{k+1/2} - u^k)\|^2$$

$$= \mathbb{E}\max_{x \in \mathcal{X}} \frac{1}{2\gamma}\|x^0 - x\|^2$$

$$+ \mathbb{E}\frac{\gamma}{2}\sum_{i=1}^{n}\sum_{k=0}^{K-1}\|d_i \cdot [A_i^T(y^{k+1/2} - u^k)]_{(j_i^k)} - \mathbb{E}_{j_i^k}\big[d_i \cdot \langle A_i^T(y^{k+1/2} - u^k), e_{j_i^k}\rangle e_{j_i^k}\big]\|^2$$

$$\leq \mathbb{E}\max_{x \in \mathcal{X}} \frac{1}{2\gamma}\|x^0 - x\|^2 + \mathbb{E}\frac{\gamma}{2}\sum_{i=1}^{n}\sum_{k=0}^{K-1}\|d_i \cdot \langle A_i^T(y^{k+1/2} - u^k), e_{j_i^k}\rangle e_{j_i^k}\|^2.$$

The analogue of (31) is

$$\mathbb{E}\max_{y \in \mathcal{Y}} \sum_{k=0}^{K-1}\langle\sum_{i=1}^{n}[s \cdot \langle A_i(x_i^{k+1/2} - w_i^k), e_{c_i^k}\rangle e_{c_i^k} + A_i w_i^k - A_i x_i^{k+1/2}], y^0 - y\rangle$$

$$\leq \mathbb{E}\max_{y \in \mathcal{Y}} \frac{1}{2\gamma}\|y^0 - y\|^2 + \mathbb{E}\frac{\gamma}{2}\sum_{k=0}^{K-1}\|\sum_{i=1}^{n} s \cdot \langle A_i(x_i^{k+1/2} - w_i^k), e_{c_i^k}\rangle e_{c_i^k}\|^2.$$

The analogue of (32) is

$$2\gamma\mathbb{E}\text{gap}(\bar{x}^K, \bar{z}^K, \bar{y}^K)$$

$$\leq \frac{1}{K}\Bigg(6\max_{x \in \mathcal{X}}\|x^0 - x\|^2 + \max_{x \in \mathcal{X}}\|w^0 - x\|^2 + \max_{z \in \mathcal{Z}}\|z^0 - z\|^2$$

$$+ 6\max_{y \in \mathcal{Y}}\|y^0 - y\|^2 + \max_{y \in \mathcal{Y}}\|u^0 - y\|^2\Bigg)$$

$$- \left(\frac{3\tau - 1}{2} - 2\gamma^2 L_r^2\right) \cdot \frac{1}{K}\sum_{k=0}^{K-1}\mathbb{E}\|x^{k+1/2} - x^k\|^2 - \frac{1-\tau}{2K}\sum_{k=0}^{K-1}\mathbb{E}\|w^k - x^{k+1/2}\|^2$$

$$- \left(1 - 2\gamma^2(1 + L_\ell^2)\right)\frac{1}{K}\sum_{k=0}^{K-1}\mathbb{E}\|z^{k+1/2} - z^k\|^2$$

$$- \left(\frac{3\tau - 1}{2} - 2\gamma^2\right)\frac{1}{K}\sum_{k=0}^{K-1}\mathbb{E}\|y^{k+1/2} - y^k\|^2 - \frac{1-\tau}{2K}\sum_{k=0}^{K-1}\mathbb{E}\|u^k - y^{k+1/2}\|^2$$

$$+ \frac{3\gamma^2}{K}\sum_{k=0}^{K-1}\sum_{i=1}^{n}\mathbb{E}\|d_i \cdot \langle A_i^T(y^{k+1/2} - u^k), e_{j_i^k}\rangle e_{j_i^k}\|^2$$

$$+ \frac{3\gamma^2}{K}\sum_{k=0}^{K-1}\mathbb{E}\|\sum_{i=1}^{n} s \cdot \langle A_i(x_i^{k+1/2} - w_i^k), e_{c_i^k}\rangle e_{c_i^k}\|^2. \tag{46}$$

Let us estimate two last lines. Here we use that coordinates $j_i$ and $c_i$ are chosen uniformly and independently.

$$\mathbb{E}\|d_i \cdot \langle A_i^T(y^{k+1/2} - u^k), e_{j_i^k}\rangle e_{j_i^k}\|^2 = d_i^2 \mathbb{E}\mathbb{E}_{e_{j_i^k}}\left[\|\langle A_i^T(y^{k+1/2} - u^k), e_{j_i^k}\rangle e_{j_i^k}\|^2\right]$$

$$= d_i^2 \mathbb{E}\frac{1}{d_i}\sum_{r=1}^{d_i}\left[\|\langle A_i^T(y^{k+1/2} - u^k), e_r\rangle e_r\|^2\right]$$

$$= d_i \mathbb{E}\|A_i^T(y^{k+1/2} - u^k)\|^2$$

$$\leq d_i \lambda_{\max}(A_i A_i^T) \mathbb{E}\|y^{k+1/2} - u^k\|^2.$$

For $\mathbb{E}\|\sum_{i=1}^n s \cdot \langle A_i(x_i^{k+1/2} - w_i^k), e_{c_i^k}\rangle e_{c_i^k}\|^2$ we have two options. If $c_i^k = c^k$ for all $i$, then $\sum_{i=1}^n \sum_{i=1}^n s \cdot \langle A_i(x_i^{k+1/2} - w_i^k), e_{c^k}\rangle e_{c^k} = s\langle \sum_{i=1}^n [A_i(x_i^{k+1/2} - w_i^k)], e_{c^k}\rangle e_{c^k} = s\langle A(x^{k+1/2} - w^k), e_{c^k}\rangle e_{c^k}$, then

$$\mathbb{E}\|\sum_{i=1}^n s \cdot \langle A_i(x_i^{k+1/2} - w_i^k), e_{c_i^k}\rangle e_{c_i^k}\|^2 = \mathbb{E}\|s\langle A(x^{k+1/2} - w^k), e_{c^k}\rangle e_{c^k}\|^2$$

$$= s^2 \mathbb{E}\mathbb{E}_{c^k}\left[\|\langle A(x^{k+1/2} - w^k), e_{c^k}\rangle e_{c^k}\|^2\right]$$

$$= s^2 \mathbb{E}\frac{1}{s}\sum_{r=1}^s \|\langle A(x^{k+1/2} - w^k), e_r\rangle e_r\|^2$$

$$= s\mathbb{E}\left[\|A(x^{k+1/2} - w^k)\|^2\right]$$

$$\leq s\lambda_{\max}(A^T A)\mathbb{E}\|x^{k+1/2} - w^k\|^2.$$

If $c_i^k$ are chosen independently (i.e. $c_i^k \neq c_j^k$), then

$$\mathbb{E}\|\sum_{i=1}^n s \cdot \langle A_i(x_i^{k+1/2} - w_i^k), e_{c_i^k}\rangle e_{c_i^k}\|^2$$

$$= \sum_{i=1}^n \mathbb{E}\|s \cdot \langle A_i(x_i^{k+1/2} - w_i^k), e_{c_i^k}\rangle e_{c_i^k}\|^2$$

$$+ \sum_{i \neq j} \mathbb{E}\langle s \cdot \langle A_i(x_i^{k+1/2} - w_i^k), e_{c_i^k}\rangle e_{c_i^k}, s \cdot \langle A_j(x_j^{k+1/2} - w_j^k), e_{c_j^k}\rangle e_{c_j^k}\rangle$$

$$= \sum_{i=1}^n \mathbb{E}\|s \cdot \langle A_i(x_i^{k+1/2} - w_i^k), e_{c_i^k}\rangle e_{c_i^k}\|^2$$

$$+ \sum_{i \neq j} \mathbb{E}\langle \mathbb{E}_{c_i^k}\left[s \cdot \langle A_i(x_i^{k+1/2} - w_i^k), e_{c_i^k}\rangle e_{c_i^k}\right], \mathbb{E}_{c_j^k}\left[s \cdot \langle A_j(x_j^{k+1/2} - w_j^k), e_{c_j^k}\rangle e_{c_j^k}\rangle\right]$$

$$= \sum_{i=1}^n \mathbb{E}\|s \cdot \langle A_i(x_i^{k+1/2} - w_i^k), e_{c_i^k}\rangle e_{c_i^k}\|^2$$

$$+ \sum_{i \neq j} \mathbb{E}\langle A_i(x_i^{k+1/2} - w_i^k), A_j(x_j^{k+1/2} - w_j^k)\rangle$$

$$= s^2 \sum_{i=1}^n \mathbb{E}\|\langle A_i(x_i^{k+1/2} - w_i^k), e_{c_i^k}\rangle e_{c_i^k}\|^2$$

$$+ \mathbb{E}\|\sum_{i=1}^n [A_i x_i^{k+1/2} - A_i w_i^k]\|^2 - \sum_{i=1}^n \mathbb{E}\|A_i x_i^{k+1/2} - A_i w_i^k\|^2$$

$$\leq s\sum_{i=1}^n \mathbb{E}\|A_i x_i^{k+1/2} - A_i w_i^k\|^2 + \mathbb{E}\|A(x^{k+1/2} - w^k)\|^2$$

$$\leq s\sum_{i=1}^n \lambda_{\max}(A_i^T A_i)\mathbb{E}\|x_i^{k+1/2} - w_i^k\|^2 + \lambda_{\max}(A^T A)\mathbb{E}\|x^{k+1/2} - w^k\|^2$$

$$\leq s\max_i\left\{\lambda_{\max}(A_i^T A_i)\right\}\sum_{i=1}^n \mathbb{E}\|x_i^{k+1/2} - w_i^k\|^2 + \lambda_{\max}(A^T A)\mathbb{E}\|x^{k+1/2} - w^k\|^2$$

$$= \left(s\max_i\left\{\lambda_{\max}(A_i^T A_i)\right\} + \lambda_{\max}(A^T A)\right)\mathbb{E}\|x^{k+1/2} - w^k\|^2.$$

Let us introduce

$$\chi_{\text{coord}} = \begin{cases} s\lambda_{\max}(A^T A), \\ s\max_i\left\{\lambda_{\max}(A_i^T A_i)\right\} + \lambda_{\max}(A^T A), \end{cases}$$

depending on the case $c_i$ we consider. It remains to come back to (46) and get

$$2\gamma\mathbb{E}\text{gap}(\bar{x}^K, \bar{z}^K, \bar{y}^K)$$

$$\leq \frac{1}{K}\left(6\max_{x\in\mathcal{X}}\|x^0 - x\|^2 + \max_{x\in\mathcal{X}}\|w^0 - x\|^2 + \max_{z\in\mathcal{Z}}\|z^0 - z\|^2\right.$$

$$\left. + 6\max_{y\in\mathcal{Y}}\|y^0 - y\|^2 + \max_{y\in\mathcal{Y}}\|u^0 - y\|^2\right)$$

$$- \left(\frac{3\tau - 1}{2} - 2\gamma^2 L_r^2\right)\cdot\frac{1}{K}\sum_{k=0}^{K-1}\mathbb{E}\|x^{k+1/2} - x^k\|^2 - \frac{1-\tau}{2K}\sum_{k=0}^{K-1}\mathbb{E}\|w^k - x^{k+1/2}\|^2$$

$$- \left(1 - 2\gamma^2(1 + L_\ell^2)\right)\frac{1}{K}\sum_{k=0}^{K-1}\mathbb{E}\|z^{k+1/2} - z^k\|^2$$

$$- \left(\frac{3\tau - 1}{2} - 2\gamma^2\right)\frac{1}{K}\sum_{k=0}^{K-1}\mathbb{E}\|y^{k+1/2} - y^k\|^2 - \frac{1-\tau}{2K}\sum_{k=0}^{K-1}\mathbb{E}\|u^k - y^{k+1/2}\|^2$$

$$+ \frac{3\gamma^2}{K}\sum_{k=0}^{K-1}\sum_{i=1}^{n}d_i\lambda_{\max}(A_iA_i^T)\mathbb{E}\|y^{k+1/2} - u^k\|^2$$

$$+ \frac{3\gamma^2\chi_{\text{coord}}}{K}\sum_{k=0}^{K-1}\mathbb{E}\|x^{k+1/2} - w^k\|^2$$

$$\leq \frac{1}{K}\left(6\max_{x\in\mathcal{X}}\|x^0 - x\|^2 + \max_{x\in\mathcal{X}}\|w^0 - x\|^2 + \max_{z\in\mathcal{Z}}\|z^0 - z\|^2\right.$$

$$\left. + 6\max_{y\in\mathcal{Y}}\|y^0 - y\|^2 + \max_{y\in\mathcal{Y}}\|u^0 - y\|^2\right)$$

$$- \left(\frac{3\tau - 1}{2} - 2\gamma^2 L_r^2\right)\cdot\frac{1}{K}\sum_{k=0}^{K-1}\mathbb{E}\|x^{k+1/2} - x^k\|^2 - \frac{1-\tau}{2K}\sum_{k=0}^{K-1}\mathbb{E}\|w^k - x^{k+1/2}\|^2$$

$$- \left(1 - 2\gamma^2(1 + L_\ell^2)\right)\frac{1}{K}\sum_{k=0}^{K-1}\mathbb{E}\|z^{k+1/2} - z^k\|^2$$

$$- \left(\frac{3\tau - 1}{2} - 2\gamma^2\right)\frac{1}{K}\sum_{k=0}^{K-1}\mathbb{E}\|y^{k+1/2} - y^k\|^2 - \frac{1-\tau}{2K}\sum_{k=0}^{K-1}\mathbb{E}\|u^k - y^{k+1/2}\|^2$$

$$+ \frac{3\gamma^2}{K}\cdot d\max_i\left\{\lambda_{\max}(A_i^T A_i)\right\}\sum_{k=0}^{K-1}\mathbb{E}\|y^{k+1/2} - u^k\|^2$$

$$+ \frac{3\gamma^2\chi_{\text{coord}}}{K}\sum_{k=0}^{K-1}\mathbb{E}\|x^{k+1/2} - w^k\|^2.$$

If we choose $\tau \geq \frac{1}{2}$ and $\gamma$ as follows

$$\gamma \leq \frac{1}{4}\min\left\{1; \frac{1}{L_r}; \frac{1}{L_\ell}; \sqrt{\frac{1-\tau}{\chi_{\text{coord}}}}; \sqrt{\frac{1-\tau}{d\max_i\{\lambda_{\max}(A_i^T A_i)\}}};\right\},$$

then one can obtain

$$\mathbb{E}\text{gap}(\bar{x}^K, \bar{z}^K, \bar{y}^K) \leq \frac{1}{2\gamma K}\left(6\max_{x\in\mathcal{X}}\|x^0 - x\|^2 + \max_{x\in\mathcal{X}}\|w^0 - x\|^2 + \max_{z\in\mathcal{Z}}\|z^0 - z\|^2\right.$$

$$\left. + 6\max_{y\in\mathcal{Y}}\|y^0 - y\|^2 + \max_{y\in\mathcal{Y}}\|u^0 - y\|^2\right).$$

With $\gamma = \frac{1}{4}\min\left\{1; \frac{1}{L_r}; \frac{1}{L_\ell}; \sqrt{\frac{1-\tau}{\chi_{\text{coord}}}}; \sqrt{\frac{1-\tau}{d\max_i\{\lambda_{\max}(A_i^T A_i)\}}};\right\}$, we finish the proof. $\square$

### D.7 PROOF OF THEOREM 4.1

**Theorem D.8** (Theorem 4.1). *Let Assumption 2.1 holds. Let problem (6) be solved by Algorithm 6 (Appendix A). Then for*

$$\gamma = \tfrac{1}{2} \cdot \min\left\{ 1; \frac{1}{\sqrt{\max_i\{\lambda_{\max}(A_i^T A_i)\}}}; \frac{1}{L_r}; \frac{1}{nL_\ell} \right\},$$

*it holds that*

$$gap_1(\bar{x}^K, \bar{z}^K, \bar{y}^K) = \mathcal{O}\left( \frac{\left(1 + \sqrt{\max_{i=1,\ldots,n}\{\lambda_{\max}(A_i^T A_i)\}} + nL_\ell + L_r\right)D^2}{K} \right),$$

*where* $gap_1(x,y) := \max_{\tilde{y}_i \in \tilde{\mathcal{Y}}} \tilde{L}(x, z, \tilde{y}) - \min_{\tilde{x}, z \in \mathcal{X}, \tilde{\mathcal{Z}}} \tilde{L}(\tilde{x}, \tilde{z}, y)$ *and* $\bar{x}^K := \frac{1}{K}\sum_{k=0}^{K-1} x^{k+1/2}$, $\bar{z}^K := \frac{1}{K}\sum_{k=0}^{K-1} z^{k+1/2}$, $\bar{y}^K := \frac{1}{K}\sum_{k=0}^{K-1} y^{k+1/2}$ *and* $D^2 := \max_{x,z,y \in \mathcal{X}, \mathcal{Z}, \mathcal{Y}} \left[\|x^0 - x\|^2 + \|z^0 - z\|^2 + \|y^0 - y\|^2\right]$.

*Proof.* We start the proof from (11), since the updates for $x_i$ variables in Algorithms 1, 6 are the same (with a slight modification $y$ to $y_i$):

$$\|x_i^{k+1} - x_i\|^2 = \|x_i^k - x_i\|^2 - \|x_i^{k+1/2} - x_i^k\|^2 - \|x_i^{k+1/2} - x_i^{k+1}\|^2$$
$$- 2\gamma\langle A_i(x_i^{k+1/2} - x_i), y_i^{k+1/2}\rangle - 2\gamma\langle\nabla r_i(x_i^{k+1/2}), x_i^{k+1/2} - x_i\rangle$$
$$- 2\gamma\langle x_i^{k+1} - x_i^{k+1/2}, A_i^T(y_i^{k+1/2} - y_i^k)\rangle$$
$$- 2\gamma\langle\nabla r_i(x_i^{k+1/2}) - \nabla r_i(x_i^k), x_i^{k+1} - x_i^{k+1/2}\rangle.$$

By simple fact: $2\langle a, b\rangle \leq \eta\|a\|^2 + \frac{1}{\eta}\|b\|^2$ with $a = A_i^T(y_i^{k+1/2} - y_i^k)$, $b = x_i^{k+1/2} - x_i^{k+1}$, $\eta = 2\gamma$ and $a = \nabla r_i(x_i^{k+1/2}) - \nabla r_i(x_i^k)$, $b = x_i^{k+1/2} - x_i^{k+1}$, $\eta = 2\gamma$, we get

$$\|x_i^{k+1} - x_i\|^2 = \|x_i^k - x_i\|^2 - \|x_i^{k+1/2} - x_i^k\|^2$$
$$- 2\gamma\langle A_i(x_i^{k+1/2} - x_i), y_i^{k+1/2}\rangle - 2\gamma\langle\nabla r_i(x_i^{k+1/2}), x_i^{k+1/2} - x_i\rangle$$
$$+ 2\gamma^2\|A_i^T(y_i^{k+1/2} - y_i^k)\|^2 + 2\gamma^2\|\nabla r_i(x_i^{k+1/2}) - \nabla r_i(x_i^k)\|^2.$$

Summing over all $i$ from 1 to $n$ and using the notation of $A = [A_1, \ldots, A_i, \ldots, A_n]$, $x = [x_1^T, \ldots, x_i^T, \ldots, x_n^T]^T$, we deduce

$$\|x^{k+1} - x\|^2 = \|x^k - x\|^2 - \|x^{k+1/2} - x^k\|^2 - \|x^{k+1/2} - x^{k+1}\|^2$$
$$- 2\gamma\sum_{i=1}^n\langle A_i(x_i^{k+1/2} - x_i), y_i^{k+1/2}\rangle - 2\gamma\sum_{i=1}^n\langle\nabla r_i(x_i^{k+1/2}), x_i^{k+1/2} - x_i\rangle$$
$$- 2\gamma\sum_{i=1}^n\langle x_i^{k+1} - x_i^{k+1/2}, A_i^T(y_i^{k+1/2} - y_i^k)\rangle$$
$$- 2\gamma\sum_{i=1}^n\langle\nabla r_i(x_i^{k+1/2}) - \nabla r_i(x_i^k), x_i^{k+1} - x_i^{k+1/2}\rangle. \tag{47}$$

Using the same steps as for (47), one can obtain for the notation of $z = [z_1^T, \ldots, z_i^T, \ldots, z_n^T]^T$ and $y = [y_1^T, \ldots, y_i^T, \ldots, y_n^T]^T$,

$$\|z^{k+1} - z\|^2 \leq \|z^k - z\|^2 - \|z^{k+1/2} - z^k\|^2 - \|z^{k+1/2} - z^{k+1}\|^2$$
$$+ 2\gamma\sum_{i=1}^n\langle y_i^{k+1/2}, z_i^{k+1/2} - z_i\rangle - 2\gamma\langle\nabla\ell\left(\sum_{j=1}^n z_j^{k+1/2}, b\right), \sum_{j=1}^n z_j^{k+1/2} - \sum_{j=1}^n z_j\rangle$$
$$- 2\gamma^2 n\|\nabla\ell\left(\sum_{j=1}^n z_j^{k+1/2}, b\right) - \nabla\ell\left(\sum_{j=1}^n z_j^k, b\right)\|^2 + 2\gamma^2\|y^{k+1/2} - y^k\|^2,$$

$$\tag{48}$$

and,

$$\|y^{k+1} - y\|^2 \leq \|y^k - y\|^2 - \|y^{k+1/2} - y^k\|^2 - \|y^{k+1/2} - y^{k+1}\|^2$$
$$- 2\gamma \sum_{i=1}^n \langle z_i^{k+1/2}, y_i^{k+1/2} - y_i \rangle + 2\gamma \sum_{i=1}^n \langle A_i x_i^{k+1/2}, y_i^{k+1/2} - y_i \rangle$$
$$+ 2\gamma^2 \sum_{i=1}^n \|A_i(x_i^{k+1/2} - x_i^k)\|^2 + 2\gamma^2 \|z^{k+1/2} - z^k\|^2. \tag{49}$$

Summing up (47), (48) and (49), we obtain

$$\|x^{k+1} - x\|^2 + \|z^{k+1} - z\|^2 + \|y^{k+1} - y\|^2$$
$$\leq \|x^k - x\|^2 + \|z^k - z\|^2 + \|y^k - y\|^2$$
$$- \|x^{k+1/2} - x^k\|^2 - \|z^{k+1/2} - z^k\|^2 - \|y^{k+1/2} - y^k\|^2$$
$$- 2\gamma \sum_{i=1}^n \langle A_i(x_i^{k+1/2} - x_i), y_i^{k+1/2} \rangle + 2\gamma \sum_{i=1}^n \langle y_i^{k+1/2}, z_i^{k+1/2} - z_i \rangle$$
$$- 2\gamma \sum_{i=1}^n \langle z_i^{k+1/2}, y_i^{k+1/2} - y_i \rangle + 2\gamma \sum_{i=1}^n \langle A_i x_i^{k+1/2}, y_i^{k+1/2} - y_i \rangle$$
$$- 2\gamma \sum_{i=1}^n \langle \nabla r_i(x_i^{k+1/2}), x_i^{k+1/2} - x_i \rangle - 2\gamma \langle \nabla \ell \left( \sum_{j=1}^n z_j^{k+1/2}, b \right), \sum_{j=1}^n z_j^{k+1/2} - \sum_{j=1}^n z_j \rangle$$
$$+ 2\gamma^2 \sum_{i=1}^n \|A_i^T(y_i^{k+1/2} - y_i^k)\|^2 + 2\gamma^2 \sum_{i=1}^n \|\nabla r_i(x_i^{k+1/2}) - \nabla r_i(x_i^k)\|^2$$
$$+ 2\gamma^2 \|y^{k+1/2} - y^k\|^2 + 2\gamma^2 n \|\nabla \ell \left( \sum_{j=1}^n z_j^{k+1/2}, b \right) - \nabla \ell \left( \sum_{j=1}^n z_j^k, b \right) \|^2$$
$$+ 2\gamma^2 \|z^{k+1/2} - z^k\|^2 + 2\gamma^2 \sum_{i=1}^n \|A_i(x_i^{k+1/2} - x_i^k)\|^2.$$

Using the definition of $\lambda_{\max}(\cdot)$ as a maximum eigenvalue, we get

$$\|x^{k+1} - x\|^2 + \|z^{k+1} - z\|^2 + \|y^{k+1} - y\|^2$$
$$\leq \|x^k - x\|^2 + \|z^k - z\|^2 + \|y^k - y\|^2$$
$$- \|x^{k+1/2} - x^k\|^2 - \|z^{k+1/2} - z^k\|^2 - \|y^{k+1/2} - y^k\|^2$$
$$+ 2\gamma \sum_{i=1}^n \langle A_i x_i - z_i, y_i^{k+1/2} \rangle - 2\gamma \sum_{i=1}^n \langle A_i x_i^{k+1/2} - z_i^{k+1/2}, y_i \rangle$$
$$- 2\gamma \sum_{i=1}^n \langle \nabla r_i(x_i^{k+1/2}), x_i^{k+1/2} - x_i \rangle - 2\gamma \langle \nabla \ell \left( \sum_{j=1}^n z_j^{k+1/2}, b \right), \sum_{j=1}^n z_j^{k+1/2} - \sum_{j=1}^n z_j \rangle$$
$$+ 2\gamma^2 \sum_{i=1}^n \lambda_{\max}(A_i A_i^T) \|y_i^{k+1/2} - y_i^k\|^2 + 2\gamma^2 \sum_{i=1}^n \|\nabla r_i(x_i^{k+1/2}) - \nabla r_i(x_i^k)\|^2$$
$$+ 2\gamma^2 \|y^{k+1/2} - y^k\|^2 + 2\gamma^2 n \|\nabla \ell \left( \sum_{j=1}^n z_j^{k+1/2}, b \right) - \nabla \ell \left( \sum_{j=1}^n z_j^k, b \right) \|^2$$
$$+ 2\gamma^2 \|z^{k+1/2} - z^k\|^2 + 2\gamma^2 \sum_{i=1}^n \lambda_{\max}(A_i^T A_i) \|x_i^{k+1/2} - x_i^k\|^2$$
$$\leq \|x^k - x\|^2 + \|z^k - z\|^2 + \|y^k - y\|^2$$
$$- \|x^{k+1/2} - x^k\|^2 - \|z^{k+1/2} - z^k\|^2 - \|y^{k+1/2} - y^k\|^2$$

$$+ 2\gamma \sum_{i=1}^{n} \langle A_i x_i - z_i, y_i^{k+1/2} \rangle - 2\gamma \sum_{i=1}^{n} \langle A_i x_i^{k+1/2} - z_i^{k+1/2}, y_i \rangle$$

$$- 2\gamma \sum_{i=1}^{n} \langle \nabla r_i(x_i^{k+1/2}), x_i^{k+1/2} - x_i \rangle - 2\gamma \langle \nabla \ell \left( \sum_{j=1}^{n} z_j^{k+1/2}, b \right), \sum_{j=1}^{n} z_j^{k+1/2} - \sum_{j=1}^{n} z_j \rangle$$

$$+ 2\gamma^2 \max_i \{\lambda_{\max}(A_i A_i^T)\} \|y^{k+1/2} - y^k\|^2 + 2\gamma^2 \sum_{i=1}^{n} \|\nabla r_i(x_i^{k+1/2}) - \nabla r_i(x_i^k)\|^2$$

$$+ 2\gamma^2 \|y^{k+1/2} - y^k\|^2 + 2\gamma^2 n \|\nabla \ell \left( \sum_{j=1}^{n} z_j^{k+1/2}, b \right) - \nabla \ell \left( \sum_{j=1}^{n} z_j^k, b \right) \|^2$$

$$+ 2\gamma^2 \|z^{k+1/2} - z^k\|^2 + 2\gamma^2 \max_i \{\lambda_{\max}(A_i^T A_i)\} \|x^{k+1/2} - x^k\|^2.$$

Using convexity and $L_r$-smoothness of the function $r_i$ with convexity and $L_\ell$-smoothness of the function $\ell$, we have

$$\|x^{k+1} - x\|^2 + \|z^{k+1} - z\|^2 + \|y^{k+1} - y\|^2$$
$$\leq \|x^k - x\|^2 + \|z^k - z\|^2 + \|y^k - y\|^2$$
$$- \|x^{k+1/2} - x^k\|^2 - \|z^{k+1/2} - z^k\|^2 - \|y^{k+1/2} - y^k\|^2$$
$$+ 2\gamma \sum_{i=1}^{n} \langle A_i x_i - z_i, y_i^{k+1/2} \rangle - 2\gamma \sum_{i=1}^{n} \langle A_i x_i^{k+1/2} - z_i^{k+1/2}, y_i \rangle$$
$$- 2\gamma \sum_{i=1}^{n} (r_i(x_i^{k+1/2}) - r_i(x_i)) - 2\gamma (\ell \left( \sum_{j=1}^{n} z_j^{k+1/2}, b \right) - \ell \left( \sum_{j=1}^{n} z_j, b \right))$$
$$+ 2\gamma^2 \max_i \{\lambda_{\max}(A_i A_i^T)\} \|y^{k+1/2} - y^k\|^2 + 2\gamma^2 L_r^2 \|x^{k+1/2} - x^k\|^2$$
$$+ 2\gamma^2 \|y^{k+1/2} - y^k\|^2 + 2\gamma^2 n L_\ell^2 \| \sum_{j=1}^{n} z_j^{k+1/2} - \sum_{j=1}^{n} z_j^k \|^2$$
$$+ 2\gamma^2 \|z^{k+1/2} - z^k\|^2 + 2\gamma^2 \max_i \{\lambda_{\max}(A_i^T A_i)\} \|x^{k+1/2} - x^k\|^2.$$

Cauchy Schwartz inequality in the form: $\| \sum_{j=1}^{n} (z_j^{k+1/2} - z_j^k) \|^2 \leq n \sum_{j=1}^{n} \|z_j^{k+1/2} - z_j^k\|^2$, gives

$$\|x^{k+1} - x\|^2 + \|z^{k+1} - z\|^2 + \|y^{k+1} - y\|^2$$
$$\leq \|x^k - x\|^2 + \|z^k - z\|^2 + \|y^k - y\|^2$$
$$- \|x^{k+1/2} - x^k\|^2 - \|z^{k+1/2} - z^k\|^2 - \|y^{k+1/2} - y^k\|^2$$
$$+ 2\gamma \sum_{i=1}^{n} \langle A_i x_i - z_i, y_i^{k+1/2} \rangle - 2\gamma \sum_{i=1}^{n} \langle A_i x_i^{k+1/2} - z_i^{k+1/2}, y_i \rangle$$
$$- 2\gamma \sum_{i=1}^{n} (r_i(x_i^{k+1/2}) - r_i(x_i)) - 2\gamma (\ell \left( \sum_{j=1}^{n} z_j^{k+1/2}, b \right) - \ell \left( \sum_{j=1}^{n} z_j, b \right))$$
$$+ 2\gamma^2 \max_i \{\lambda_{\max}(A_i A_i^T)\} \|y^{k+1/2} - y^k\|^2 + 2\gamma^2 L_r^2 \|x^{k+1/2} - x^k\|^2$$
$$+ 2\gamma^2 \|y^{k+1/2} - y^k\|^2 + 2\gamma^2 n^2 L_\ell^2 \|z^{k+1/2} - z^k\|^2$$
$$+ 2\gamma^2 \|z^{k+1/2} - z^k\|^2 + 2\gamma^2 \max_i \{\lambda_{\max}(A_i^T A_i)\} \|x^{k+1/2} - x^k\|^2.$$

With the choice of $\gamma \leq \frac{1}{2} \cdot \min \left\{ 1; \frac{1}{\sqrt{\max_i \{\lambda_{\max}(A_i^T A_i)\}}}; \frac{1}{L_r}; \frac{1}{nL_\ell} \right\}$, we get

$$\|x^{k+1} - x\|^2 + \|z^{k+1} - z\|^2 + \|y^{k+1} - y\|^2$$

$$\leq \|x^k - x\|^2 + \|z^k - z\|^2 + \|y^k - y\|^2$$
$$+ 2\gamma \sum_{i=1}^{n} \langle A_i x_i - z_i, y_i^{k+1/2} \rangle - 2\gamma \sum_{i=1}^{n} \langle A_i x_i^{k+1/2} - z_i^{k+1/2}, y_i \rangle$$
$$- 2\gamma \sum_{i=1}^{n} (r_i(x_i^{k+1/2}) - r_i(x_i)) - 2\gamma(\ell\left(\sum_{j=1}^{n} z_j^{k+1/2}, b\right) - \ell\left(\sum_{j=1}^{n} z_j, b\right)).$$

After small rearrangements, we obtain

$$(\ell\left(\sum_{j=1}^{n} z_j^{k+1/2}, b\right) - \ell\left(\sum_{j=1}^{n} z_j, b\right)) + \sum_{i=1}^{n} \left(r_i(x_i^{k+1/2}) - r_i(x_i)\right)$$
$$+ \sum_{i=1}^{n} \langle A_i x_i^{k+1/2} - z_i^{k+1/2}, y_i \rangle - \sum_{i=1}^{n} \langle A_i x_i - z_i, y_i^{k+1/2} \rangle$$
$$\leq \frac{1}{2\gamma}\Big(\|x^k - x\|^2 + \|z^k - z\|^2 + \|y^k - y\|^2$$
$$- \|x^{k+1} - x\|^2 - \|z^{k+1} - z\|^2 - \|y^{k+1} - y\|^2\Big).$$

Then we sum all over $k$ from $0$ to $K-1$, divide by $K$, and have

$$\frac{1}{K} \sum_{k=0}^{K-1} (\ell\left(\sum_{j=1}^{n} z_j^{k+1/2}, b\right) - \ell\left(\sum_{j=1}^{n} z_j, b\right)) + \sum_{i=1}^{n} \frac{1}{K} \sum_{k=0}^{K-1} \left(r_i(x_i^{k+1/2}) - r_i(x_i)\right)$$
$$+ \sum_{i=1}^{n} \langle A_i x_i^{k+1/2} - z_i^{k+1/2}, y_i \rangle - \sum_{i=1}^{n} \langle A_i x_i - z_i, y_i^{k+1/2} \rangle$$
$$\leq \frac{1}{2\gamma K}\Big(\|x^0 - x\|^2 + \|z^0 - z\|^2 + \|y^0 - y\|^2$$
$$- \|x^K - x\|^2 - \|z^K - z\|^2 - \|y^K - y\|^2\Big)$$
$$\leq \frac{1}{2\gamma K}(\|x^0 - x\|^2 + \|z^0 - z\|^2 + \|y^0 - y\|^2).$$

With Jensen inequality for convex functions $\ell$ and $r_i$, one can note that

$$\ell\left(\frac{1}{K} \sum_{k=0}^{K-1} \sum_{j=1}^{n} z_j^{k+1/2}, b\right) \leq \frac{1}{K} \sum_{k=0}^{K-1} \ell\left(\sum_{j=1}^{n} z_j^{k+1/2}, b\right),$$
$$r_i\left(\frac{1}{K} \sum_{k=0}^{K-1} x_i^{k+1/2}\right) \leq \frac{1}{K} \sum_{k=0}^{K-1} r_i(x_i^{k+1/2}).$$

Then, with notation $\bar{x}_i^K = \frac{1}{K} \sum_{k=0}^{K-1} x_i^{k+1/2}$, $\bar{z}_i^K = \frac{1}{K} \sum_{k=0}^{K-1} z_i^{k+1/2}$, $\bar{y}_i^K = \frac{1}{K} \sum_{k=0}^{K-1} y_i^{k+1/2}$, we have

$$\ell\left(\sum_{i=1}^{n} \bar{z}_i^K, b\right) - \ell\left(\sum_{i=1}^{n} z_i, b\right) + \sum_{i=1}^{n} \left(r_i(\bar{x}_i^K) - r_i(x_i)\right)$$
$$+ \sum_{i=1}^{n} \langle A_i x_i^{k+1/2} - z_i^{k+1/2}, y_i \rangle - \sum_{i=1}^{n} \langle A_i x_i - z_i, y_i^{k+1/2} \rangle$$
$$\leq \frac{1}{2\gamma K}(\|x^0 - x\|^2 + \|z^0 - z\|^2 + \|y^0 - y\|^2).$$

Following the definition of $\mathrm{gap}_1$, we only need to take the maximum in the variable $y_i \in \mathcal{Y}$ and the minimum in $x \in \mathcal{X}$ and $z_i \in \mathcal{Z}$.

$$\mathrm{gap}_1(\bar{x}^K, \bar{z}^K, \bar{y}^K)$$

$$= \max_{\tilde{y} \in \tilde{\mathcal{Y}}} \tilde{L}(\bar{x}^K, \bar{z}^K, \tilde{y}) - \min_{\tilde{x}, z \in \mathcal{X}, \tilde{\mathcal{Z}}} \tilde{L}(\tilde{x}, \tilde{z}, \bar{y}^K)$$

$$= \max_{y \in \tilde{\mathcal{Y}}} \left[ \ell \left( \sum_{i=1}^{n} \bar{z}_i^K, b \right) + \sum_{i=1}^{n} r_i(\bar{x}_i^K) + \sum_{i=1}^{n} \langle A_i x_i^{k+1/2} - z_i^{k+1/2}, y_i \rangle \right]$$

$$- \min_{x, z \in \mathcal{X}, \tilde{\mathcal{Z}}} \left[ \ell \left( \sum_{i=1}^{n} z_i, b \right) + \sum_{i=1}^{n} r_i(x_i) + \sum_{i=1}^{n} \langle A_i x_i - z_i, y_i^{k+1/2} \rangle \right]$$

$$= \max_{y \in \tilde{\mathcal{Y}}} \max_{x, z \in \mathcal{X}, \tilde{\mathcal{Z}}} \left[ \ell \left( \sum_{i=1}^{n} \bar{z}_i^K, b \right) - \ell \left( \sum_{i=1}^{n} z_i, b \right) + \sum_{i=1}^{n} \left( r_i(\bar{x}_i^K) - r_i(x_i) \right) \right.$$

$$\left. + \sum_{i=1}^{n} \langle A_i x_i^{k+1/2} - z_i^{k+1/2}, y_i \rangle - \sum_{i=1}^{n} \langle A_i x_i - z_i, y_i^{k+1/2} \rangle \right]$$

$$\leq \frac{1}{2\gamma K} \left( \max_{x \in \mathcal{X}} \|x^0 - x\|^2 + \max_{z \in \tilde{\mathcal{Z}}} \|z^0 - z\|^2 + \max_{y \in \tilde{\mathcal{Y}}} \|y^0 - y\|^2 \right).$$

$$\square$$

### D.8    PROOF OF THEOREM C.1

**Theorem D.9** (Theorem C.1). *Let Assumption 2.1 holds. Let problem (8) be solved by Algorithm 9. Then for*

$$\gamma = \tfrac{1}{4} \cdot \min \left\{ 1; \tfrac{1}{\rho}; \tfrac{1}{\sqrt{\lambda_{\max}(A^T A)}}; \tfrac{1}{\sqrt{\rho \lambda_{\max}(A^T A)}}; \tfrac{1}{\rho \lambda_{\max}(A^T A)}; \tfrac{1}{L_r}; \tfrac{1}{L_\ell} \right\},$$

*it holds that*

$$gap_{aug}(\bar{x}^K, \bar{z}^K, \bar{y}^K) = \mathcal{O}\left( \frac{\left(1 + \rho + \sqrt{(1+\rho)\lambda_{\max}(A^T A)} + \rho \lambda_{\max}(A^T A) + L_\ell + L_r \right) D^2}{K} \right),$$

*where* $gap_{aug}(x, z, y) := \max_{\tilde{y} \in \mathcal{Y}} L_{aug}(x, z, \tilde{y}) - \min_{\tilde{x}, \tilde{z} \in \mathcal{X}, \mathcal{Z}} L_{aug}(\tilde{x}, \tilde{z}, y)$ *and* $\bar{x}^K := \frac{1}{K} \sum_{k=0}^{K-1} x^{k+1/2}$, $\bar{z}^K := \frac{1}{K} \sum_{k=0}^{K-1} z^{k+1/2}$, $\bar{y}^K := \frac{1}{K} \sum_{k=0}^{K-1} y^{k+1/2}$ *and* $D^2 := \max_{x, z, y \in \mathcal{X}, \mathcal{Z}, \mathcal{Y}} \left[ \|x^0 - x\|^2 + \|z^0 - z\|^2 + \|y^0 - y\|^2 \right]$.

To prove the convergence it is sufficient to show that the problem is convex–concave (Lemma D.12), to estimate the Lipschitz constant of gradients and use the general results from (Nemirovski, 2004). But for completeness, we give the proof of our special case here.

*Proof.* We start the proof with the following equations on the variables $x_i^{k+1}$, $x_i^{k+1/2}$, $x_i^k$ and any $x_i \in \mathbb{R}^{d_i}$:

$$\|x_i^{k+1} - x_i\|^2 = \|x_i^k - x_i\|^2 + 2\langle x_i^{k+1} - x_i^k, x_i^{k+1} - x_i \rangle - \|x_i^{k+1} - x_i^k\|^2,$$

$$\|x_i^{k+1/2} - x_i^{k+1}\|^2 = \|x_i^k - x_i^{k+1}\|^2 + 2\langle x_i^{k+1/2} - x_i^k, x_i^{k+1/2} - x_i^{k+1} \rangle - \|x_i^{k+1/2} - x_i^k\|^2.$$

Summing up two previous inequalities and making small rearrangements, we get

$$\|x_i^{k+1} - x_i\|^2 = \|x_i^k - x_i\|^2 - \|x_i^{k+1/2} - x_i^k\|^2 - \|x_i^{k+1/2} - x_i^{k+1}\|^2$$
$$+ 2\langle x_i^{k+1} - x_i^k, x_i^{k+1} - x_i \rangle + 2\langle x_i^{k+1/2} - x_i^k, x_i^{k+1/2} - x_i^{k+1} \rangle.$$

Using that $x_i^{k+1} - x_i^k = -\gamma(A_i^T y^{k+1/2} + \nabla r_i(x_i^{k+1/2}) + \rho A_i^T(\sum_{i=1}^{n} A_i x_i^{k+1/2} - z^{k+1/2}))$ and $x_i^{k+1/2} - x_i^k = -\gamma(A_i^T y^k + \nabla r_i(x_i^k) + \rho A_i^T(\sum_{i=1}^{n} A_i x_i^k - z^k))$ (see lines 5 and 10 of Algorithm 9), we obtain

$$\|x_i^{k+1} - x_i\|^2 = \|x_i^k - x_i\|^2 - \|x_i^{k+1/2} - x_i^k\|^2 - \|x_i^{k+1/2} - x_i^{k+1}\|^2$$
$$- 2\gamma \langle A_i^T y^{k+1/2} + \nabla r_i(x_i^{k+1/2}) + \rho A_i^T(\sum_{i=1}^{n} A_i x_i^{k+1/2} - z^{k+1/2}), x_i^{k+1} - x_i \rangle$$

$$- 2\gamma\langle A_i^T y^k + \nabla r_i(x_i^k) + \rho A_i^T(\sum_{i=1}^n A_i x_i^k - z^k), x_i^{k+1/2} - x_i^{k+1}\rangle$$

$$= \|x_i^k - x_i\|^2 - \|x_i^{k+1/2} - x_i^k\|^2 - \|x_i^{k+1/2} - x_i^{k+1}\|^2$$

$$- 2\gamma\langle A_i^T y^{k+1/2} + \nabla r_i(x_i^{k+1/2}) + \rho A_i^T(\sum_{i=1}^n A_i x_i^{k+1/2} - z^{k+1/2}), x_i^{k+1/2} - x_i\rangle$$

$$- 2\gamma\langle A_i^T(y^{k+1/2} - y^k) + \nabla r_i(x_i^{k+1/2}) - \nabla r_i(x_i^k), x_i^{k+1} - x_i^{k+1/2}\rangle$$

$$- 2\gamma\rho\langle A_i^T(\sum_{i=1}^n A_i(x_i^{k+1/2} - x_i^k) + z^k - z^{k+1/2}), x_i^{k+1} - x_i^{k+1/2}\rangle$$

$$= \|x_i^k - x_i\|^2 - \|x_i^{k+1/2} - x_i^k\|^2 - \|x_i^{k+1/2} - x_i^{k+1}\|^2$$

$$- 2\gamma\langle A_i(x_i^{k+1/2} - x_i), y^{k+1/2}\rangle - 2\gamma\langle\nabla r_i(x_i^{k+1/2}), x_i^{k+1/2} - x_i\rangle$$

$$- 2\rho\gamma\langle\sum_{i=1}^n A_i x_i^{k+1/2} - z^{k+1/2}, A_i(x_i^{k+1/2} - x_i)\rangle$$

$$- 2\gamma\langle A_i(x_i^{k+1} - x_i^{k+1/2}), y^{k+1/2} - y^k\rangle$$

$$- 2\gamma\langle\nabla r_i(x_i^{k+1/2}) - \nabla r_i(x_i^k), x_i^{k+1} - x_i^{k+1/2}\rangle$$

$$- 2\gamma\rho\langle A_i(x_i^{k+1} - x_i^{k+1/2}), \sum_{i=1}^n A_i(x_i^{k+1/2} - x_i^k)\rangle$$

$$- 2\gamma\rho\langle A_i(x_i^{k+1} - x_i^{k+1/2}), z^k - z^{k+1/2}\rangle. \tag{50}$$

Summing over all $i$ from 1 to $n$, we deduce

$$\sum_{i=1}^n \|x_i^{k+1} - x_i\|^2 = \sum_{i=1}^n \|x_i^k - x_i\|^2 - \sum_{i=1}^n \|x_i^{k+1/2} - x_i^k\|^2 - \sum_{i=1}^n \|x_i^{k+1/2} - x_i^{k+1}\|^2$$

$$- 2\gamma\langle\sum_{i=1}^n A_i(x_i^{k+1/2} - x_i), y^{k+1/2}\rangle - 2\gamma\sum_{i=1}^n\langle\nabla r_i(x_i^{k+1/2}), x_i^{k+1/2} - x_i\rangle$$

$$- 2\rho\gamma\sum_{i=1}^n\langle\sum_{i=1}^n A_i x_i^{k+1/2} - z^{k+1/2}, A_i(x_i^{k+1/2} - x_i)\rangle$$

$$- 2\gamma\sum_{i=1}^n\langle\nabla r_i(x_i^{k+1/2}) - \nabla r_i(x_i^k), x_i^{k+1} - x_i^{k+1/2}\rangle$$

$$- 2\gamma\langle\sum_{i=1}^n A_i(x_i^{k+1} - x_i^{k+1/2}), y^{k+1/2} - y^k\rangle$$

$$- 2\gamma\rho\sum_{i=1}^n\langle A_i(x_i^{k+1} - x_i^{k+1/2}), \sum_{i=1}^n A_i(x_i^{k+1/2} - x_i^k)\rangle$$

$$- 2\gamma\rho\sum_{i=1}^n\langle A_i(x_i^{k+1} - x_i^{k+1/2}), z^k - z^{k+1/2}\rangle.$$

With notation of $A = [A_1, \ldots, A_i, \ldots, A_n]$ and notation of $x = [x_1^T, \ldots, x_i^T, \ldots, x_n^T]^T$ from equation 1 and equation 2, one can obtain that $\sum_{i=1}^n A_i x_i = Ax$:

$$\|x^{k+1} - x\|^2 = \|x^k - x\|^2 - \|x^{k+1/2} - x^k\|^2 - \|x^{k+1/2} - x^{k+1}\|^2$$

$$- 2\gamma\langle A(x^{k+1/2} - x), y^{k+1/2}\rangle - 2\gamma\sum_{i=1}^n\langle\nabla r_i(x_i^{k+1/2}), x_i^{k+1/2} - x_i\rangle$$

$$- 2\rho\gamma\langle Ax^{k+1/2} - z^{k+1/2}, A(x^{k+1/2} - x)\rangle$$

$$- 2\gamma\langle A(x^{k+1} - x^{k+1/2}), y^{k+1/2} - y^k\rangle$$

$$- 2\gamma \sum_{i=1}^{n} \langle \nabla r_i(x_i^{k+1/2}) - \nabla r_i(x_i^k), x_i^{k+1} - x_i^{k+1/2} \rangle$$

$$- 2\gamma\rho \langle A(x^{k+1} - x^{k+1/2}), A(x^{k+1/2} - x^k) \rangle$$

$$- 2\gamma\rho \langle A(x^{k+1} - x^{k+1/2}), z^k - z^{k+1/2} \rangle$$

$$= \|x^k - x\|^2 - \|x^{k+1/2} - x^k\|^2 - \|x^{k+1/2} - x^{k+1}\|^2$$

$$- 2\gamma \langle A(x^{k+1/2} - x), y^{k+1/2} \rangle - 2\gamma \sum_{i=1}^{n} \langle \nabla r_i(x_i^{k+1/2}), x_i^{k+1/2} - x_i \rangle$$

$$- 2\rho\gamma \langle Ax^{k+1/2} - z^{k+1/2}, A(x^{k+1/2} - x) \rangle$$

$$- 2\gamma \langle A^T(y^{k+1/2} - y^k), x^{k+1} - x^{k+1/2} \rangle$$

$$- 2\gamma \sum_{i=1}^{n} \langle \nabla r_i(x_i^{k+1/2}) - \nabla r_i(x_i^k), x_i^{k+1} - x_i^{k+1/2} \rangle$$

$$- 2\gamma\rho \langle A(x^{k+1} - x^{k+1/2}), A(x^{k+1/2} - x^k) \rangle$$

$$- 2\gamma\rho \langle A(x^{k+1} - x^{k+1/2}), z^k - z^{k+1/2} \rangle.$$

By Cauchy Schwartz inequality, we get

$$\|x^{k+1} - x\|^2 \leq \|x^k - x\|^2 - \|x^{k+1/2} - x^k\|^2 - \|x^{k+1/2} - x^{k+1}\|^2$$

$$- 2\gamma \langle A(x^{k+1/2} - x), y^{k+1/2} \rangle - 2\gamma \sum_{i=1}^{n} \langle \nabla r_i(x_i^{k+1/2}), x_i^{k+1/2} - x_i \rangle$$

$$- 2\rho\gamma \langle Ax^{k+1/2} - z^{k+1/2}, A(x^{k+1/2} - x) \rangle$$

$$+ 4\gamma^2 \|A^T(y^{k+1/2} - y^k)\|^2 + \frac{1}{4} \|x^{k+1} - x^{k+1/2}\|^2$$

$$+ 4\gamma^2 \sum_{i=1}^{n} \|\nabla r_i(x_i^{k+1/2}) - \nabla r_i(x_i^k)\|^2 + \frac{1}{4} \sum_{i=1}^{n} \|x_i^{k+1} - x_i^{k+1/2}\|^2$$

$$+ 4\gamma^2\rho^2 \|A^T(z^{k+1/2} - z^k)\|^2 + \frac{1}{4} \|x^{k+1} - x^{k+1/2}\|^2$$

$$+ 4\gamma^2\rho^2 \|A^T A(x^{k+1/2} - x^k)\|^2 + \frac{1}{4} \|x^{k+1} - x^{k+1/2}\|^2$$

$$= \|x^k - x\|^2 - \|x^{k+1/2} - x^k\|^2$$

$$- 2\gamma \langle A(x^{k+1/2} - x), y^{k+1/2} \rangle - 2\gamma \sum_{i=1}^{n} \langle \nabla r_i(x_i^{k+1/2}), x_i^{k+1/2} - x_i \rangle$$

$$- 2\rho\gamma \langle Ax^{k+1/2} - z^{k+1/2}, A(x^{k+1/2} - x) \rangle$$

$$+ 4\gamma^2 \|A^T(y^{k+1/2} - y^k)\|^2 + 4\gamma^2 \sum_{i=1}^{n} \|\nabla r_i(x_i^{k+1/2}) - \nabla r_i(x_i^k)\|^2$$

$$+ 4\gamma^2\rho^2 \|A^T(z^{k+1/2} - z^k)\|^2 + 4\gamma^2\rho^2 \|A^T A(x^{k+1/2} - x^k)\|^2. \tag{51}$$

Using the same steps, one can obtain for $z \in \mathbb{R}^s$,

$$\|z^{k+1} - z\|^2 \leq \|z^k - z\|^2 - \|z^{k+1/2} - z^k\|^2$$

$$+ 2\gamma \langle y^{k+1/2}, z^{k+1/2} - z \rangle - 2\gamma \langle \nabla \ell(z^{k+1/2}, b), z^{k+1/2} - z \rangle$$

$$- 2\gamma\rho \langle z^{k+1/2} - Ax^{k+1/2}, z^{k+1/2} - z \rangle$$

$$+ 4\gamma^2 \|y^{k+1/2} - y^k\|^2 + 4\gamma^2 \|\nabla \ell(z^{k+1/2}, b) - \nabla \ell(z^k, b)\|^2$$

$$+ 4\gamma^2\rho^2 \|z^{k+1/2} - z^k\|^2 + 4\gamma^2\rho^2 \|A(x^{k+1/2} - x^k)\|^2. \tag{52}$$

and for all $y \in \mathbb{R}^s$,

$$\|y^{k+1} - y\|^2 \leq \|y^k - y\|^2 - \|y^{k+1/2} - y^k\|^2$$

$$- 2\gamma\langle z^{k+1/2}, y^{k+1/2} - y\rangle + 2\gamma\langle Ax^{k+1/2}, y^{k+1/2} - y\rangle$$
$$+ 2\gamma^2\|z^{k+1/2} - z^k\|^2 + 2\gamma^2\|A(x^{k+1/2} - x^k)\|^2. \tag{53}$$

Summing up equation 51, equation 52 and equation 53, we obtain

$$\|x^{k+1} - x\|^2 + \|z^{k+1} - z\|^2 + \|y^{k+1} - y\|^2$$
$$\leq \|x^k - x\|^2 + \|z^k - z\|^2 + \|y^k - y\|^2$$
$$- \|x^{k+1/2} - x^k\|^2 - \|z^{k+1/2} - z^k\|^2 - \|y^{k+1/2} - y^k\|^2$$
$$- 2\gamma\langle A(x^{k+1/2} - x), y^{k+1/2}\rangle + 2\gamma\langle y^{k+1/2}, z^{k+1/2} - z\rangle$$
$$- 2\gamma\langle z^{k+1/2}, y^{k+1/2} - y\rangle + 2\gamma\langle Ax^{k+1/2}, y^{k+1/2} - y\rangle$$
$$- 2\gamma\sum_{i=1}^{n}\langle\nabla r_i(x_i^{k+1/2}), x_i^{k+1/2} - x_i\rangle - 2\gamma\langle\nabla\ell(z^{k+1/2}, b), z^{k+1/2} - z\rangle$$
$$- 2\rho\gamma\langle Ax^{k+1/2} - z^{k+1/2}, A(x^{k+1/2} - x) - (z^{k+1/2} - z)\rangle$$
$$- 2\gamma\rho\langle z^{k+1/2} - Ax^{k+1/2}, z^{k+1/2} - z\rangle$$
$$+ 2\gamma^2\|A^T(y^{k+1/2} - y^k)\|^2 + 2\gamma^2\sum_{i=1}^{n}\|\nabla r_i(x_i^{k+1/2}) - \nabla r_i(x_i^k)\|^2$$
$$+ 4\gamma^2\|y^{k+1/2} - y^k\|^2 + 4\gamma^2\|\nabla\ell(z^{k+1/2}, b) - \nabla\ell(z^k, b)\|^2$$
$$+ 4\gamma^2\|z^{k+1/2} - z^k\|^2 + 4\gamma^2\|A(x^{k+1/2} - x^k)\|^2$$
$$+ 4\gamma^2\rho^2\|A^T(z^{k+1/2} - z^k)\|^2 + 4\gamma^2\rho^2\|A^T A(x^{k+1/2} - x^k)\|^2$$
$$+ 4\gamma^2\rho^2\|z^{k+1/2} - z^k\|^2 + 4\gamma^2\rho^2\|A(x^{k+1/2} - x^k)\|^2.$$

Using convexity and $L_r$-smoothness of the function $r_i$ with convexity and $L_\ell$-smoothness of the function $\ell$ (Assumption 2.1), we have

$$\|x^{k+1} - x\|^2 + \|z^{k+1} - z\|^2 + \|y^{k+1} - y\|^2$$
$$\leq \|x^k - x\|^2 + \|z^k - z\|^2 + \|y^k - y\|^2$$
$$- \|x^{k+1/2} - x^k\|^2 - \|z^{k+1/2} - z^k\|^2 - \|y^{k+1/2} - y^k\|^2$$
$$- 2\gamma\langle A(x^{k+1/2} - x), y^{k+1/2}\rangle + 2\gamma\langle y^{k+1/2}, z^{k+1/2} - z\rangle$$
$$- 2\gamma\langle z^{k+1/2}, y^{k+1/2} - y\rangle + 2\gamma\langle Ax^{k+1/2}, y^{k+1/2} - y\rangle$$
$$- 2\gamma\sum_{i=1}^{n}\left(r_i(x_i^{k+1/2}) - r_i(x_i)\right) - 2\gamma\left(l(z^{k+1/2}, b) - l(z, b)\right)$$
$$- 2\rho\gamma\langle Ax^{k+1/2} - z^{k+1/2}, A(x^{k+1/2} - x) - (z^{k+1/2} - z)\rangle$$
$$+ 4\gamma^2\|A^T(y^{k+1/2} - y^k)\| + 4\gamma^2 L_r^2\sum_{i=1}^{n}\|x_i^{k+1/2} - x_i^k\|^2$$
$$+ 4\gamma^2\|y^{k+1/2} - y^k\|^2 + 4\gamma^2 L_\ell^2\|z^{k+1/2} - z^k\|^2$$
$$+ 4\gamma^2\|z^{k+1/2} - z^k\|^2 + 4\gamma^2\|A(x^{k+1/2} - x^k)\|^2$$
$$+ 4\gamma^2\rho^2\|A^T(z^{k+1/2} - z^k)\|^2 + 4\gamma^2\rho^2\|A^T A(x^{k+1/2} - x^k)\|^2$$
$$+ 4\gamma^2\rho^2\|z^{k+1/2} - z^k\|^2 + 4\gamma^2\rho^2\|A(x^{k+1/2} - x^k)\|^2.$$

Using the definition of $\lambda_{\max}(\cdot)$ as a maximum eigenvalue, we get

$$\|x^{k+1} - x\|^2 + \|z^{k+1} - z\|^2 + \|y^{k+1} - y\|^2$$
$$\leq \|x^k - x\|^2 + \|z^k - z\|^2 + \|y^k - y\|^2$$
$$- \|x^{k+1/2} - x^k\|^2 - \|z^{k+1/2} - z^k\|^2 - \|y^{k+1/2} - y^k\|^2$$
$$- 2\gamma\langle A(x^{k+1/2} - x), y^{k+1/2}\rangle + 2\gamma\langle y^{k+1/2}, z^{k+1/2} - z\rangle$$

$$- 2\gamma\langle z^{k+1/2}, y^{k+1/2} - y\rangle + 2\gamma\langle Ax^{k+1/2}, y^{k+1/2} - y\rangle$$

$$- 2\gamma\sum_{i=1}^{n}\left(r_i(x_i^{k+1/2}) - r_i(x_i)\right) - 2\gamma\left(l(z^{k+1/2}, b) - l(z, b)\right)$$

$$- 2\rho\gamma\langle Ax^{k+1/2} - z^{k+1/2}, A(x^{k+1/2} - x) - (z^{k+1/2} - z)\rangle$$

$$+ 4\gamma^2\lambda_{\max}(AA^T)\|y^{k+1/2} - y^k\| + 4\gamma^2 L_r^2\|x^{k+1/2} - x^k\|^2$$

$$+ 4\gamma^2\|y^{k+1/2} - y^k\|^2 + 4\gamma^2 L_\ell^2\|z^{k+1/2} - z^k\|^2$$

$$+ 4\gamma^2\|z^{k+1/2} - z^k\|^2 + 4\gamma^2\lambda_{\max}(A^T A)\|x^{k+1/2} - x^k\|^2$$

$$+ 4\gamma^2\rho^2\lambda_{\max}(AA^T)\|z^{k+1/2} - z^k\|^2 + 4\gamma^2\rho^2\lambda_{\max}^2(AA^T)\|x^{k+1/2} - x^k\|^2$$

$$+ 4\gamma^2\rho^2\|z^{k+1/2} - z^k\|^2 + 4\gamma^2\rho^2\lambda_{\max}(AA^T)\|x^{k+1/2} - x^k\|^2.$$

With the choice of $\gamma \le \frac{1}{4} \cdot \min\left\{1; \frac{1}{\rho}; \frac{1}{\sqrt{\lambda_{\max}(A^T A)}}; \frac{1}{\sqrt{\rho\lambda_{\max}(A^T A)}}; \frac{1}{\rho\lambda_{\max}(A^T A)}; \frac{1}{L_r}; \frac{1}{L_\ell}\right\}$, we get

$$\|x^{k+1} - x\|^2 + \|z^{k+1} - z\|^2 + \|y^{k+1} - y\|^2$$

$$\le \|x^k - x\|^2 + \|z^k - z\|^2 + \|y^k - y\|^2$$

$$- 2\gamma\langle A(x^{k+1/2} - x), y^{k+1/2}\rangle + 2\gamma\langle y^{k+1/2}, z^{k+1/2} - z\rangle$$

$$- 2\gamma\langle z^{k+1/2}, y^{k+1/2} - y\rangle + 2\gamma\langle Ax^{k+1/2}, y^{k+1/2} - y\rangle$$

$$- 2\gamma\sum_{i=1}^{n}\left(r_i(x_i^{k+1/2}) - r_i(x_i)\right) - 2\gamma\left(l(z^{k+1/2}, b) - l(z, b)\right)$$

$$- 2\rho\gamma\langle Ax^{k+1/2} - z^{k+1/2}, A(x^{k+1/2} - x) - (z^{k+1/2} - z)\rangle$$

$$= \|x^k - x\|^2 + \|z^k - z\|^2 + \|y^k - y\|^2$$

$$+ 2\gamma\langle Ax - z, y^{k+1/2}\rangle - 2\gamma\langle Ax^{k+1/2} - z^{k+1/2}, y\rangle$$

$$- 2\gamma\sum_{i=1}^{n}\left(r_i(x_i^{k+1/2}) - r_i(x_i)\right) - 2\gamma\left(l(z^{k+1/2}, b) - l(z, b)\right)$$

$$- \rho\gamma\|Ax^{k+1/2} - z^{k+1/2}\|^2 + \rho\gamma\|Ax - z\|^2 - \rho\gamma\|A(x^{k+1/2} - x) - (z^{k+1/2} - z)\|^2.$$

After small rearrangements, we obtain

$$\left(\ell(z^{k+1/2}, b) - \ell(z, b)\right) + \sum_{i=1}^{n}\left(r_i(x_i^{k+1/2}) - r_i(x_i)\right)$$

$$+ \langle Ax^{k+1/2} - z^{k+1/2}, y\rangle - \langle Ax - z, y^{k+1/2}\rangle$$

$$+ \frac{\rho}{2}\|Ax^{k+1/2} - z^{k+1/2}\|^2 - \frac{\rho}{2}\|Ax - z\|^2$$

$$\le \frac{1}{2\gamma}\Big(\|x^k - x\|^2 + \|z^k - z\|^2 + \|y^k - y\|^2$$

$$- \|x^{k+1} - x\|^2 - \|z^{k+1} - z\|^2 - \|y^{k+1} - y\|^2\Big).$$

Then we sum all over $k$ from $0$ to $K - 1$, divide by $K$, and have

$$\frac{1}{K}\sum_{k=0}^{K-1}\left(\ell(z^{k+1/2}, b) - \ell(z, b)\right) + \sum_{i=1}^{n}\frac{1}{K}\sum_{k=0}^{K-1}\left(r_i(x_i^{k+1/2}) - r_i(x_i)\right)$$

$$+ \left\langle A\cdot\frac{1}{K}\sum_{k=0}^{K-1}x^{k+1/2} - \frac{1}{K}\sum_{k=0}^{K-1}z^{k+1/2}, y\right\rangle - \left\langle Ax - z, \frac{1}{K}\sum_{k=0}^{K-1}y^{k+1/2}\right\rangle$$

$$+ \frac{\rho}{2}\frac{1}{K}\sum_{k=0}^{K-1}\|Ax^{k+1/2} - z^{k+1/2}\|^2 - \frac{\rho}{2}\|Ax - z\|^2$$

$$\leq \frac{1}{2\gamma K} \Big( \|x^0 - x\|^2 + \|z^0 - z\|^2 + \|y^0 - y\|^2$$

$$- \|x^K - x\|^2 - \|z^K - z\|^2 - \|y^K - y\|^2 \Big)$$

$$\leq \frac{1}{2\gamma K} (\|x^0 - x\|^2 + \|z^0 - z\|^2 + \|y^0 - y\|^2).$$

With Jensen inequality for convex functions $\ell$, $r_i$ and $\|\cdot\|^2$, one can note that

$$\ell \left( \frac{1}{K} \sum_{k=0}^{K-1} z^{k+1/2}, b \right) \leq \frac{1}{K} \sum_{k=0}^{K-1} \ell(z^{k+1/2}, b),$$

$$r_i \left( \frac{1}{K} \sum_{k=0}^{K-1} x_i^{k+1/2} \right) \leq \frac{1}{K} \sum_{k=0}^{K-1} r_i(x_i^{k+1/2}),$$

$$\left\| A \frac{1}{K} \sum_{k=0}^{K-1} x^{k+1/2} - \frac{1}{K} \sum_{k=0}^{K-1} z^{k+1/2} \right\|^2 \leq \frac{1}{K} \sum_{k=0}^{K-1} \|Ax^{k+1/2} - z^{k+1/2}\|^2.$$

Then, with notation $\bar{x}_i^K = \frac{1}{K} \sum_{k=0}^{K-1} x_i^{k+1/2}$, $\bar{z}^K = \frac{1}{K} \sum_{k=0}^{K-1} z^{k+1/2}$, $\bar{y}^K = \frac{1}{K} \sum_{k=0}^{K-1} y^{k+1/2}$, we have

$$\ell(\bar{z}^K, b) - \ell(z, b) + \sum_{i=1}^{n} \left( r_i(\bar{x}_i^K) - r_i(x_i) \right) + \langle A\bar{x}^K - \bar{z}^K, y \rangle - \langle Ax - z, \bar{y}^K \rangle$$

$$+ \frac{\rho}{2} \|A\bar{x}^K - \bar{z}^K\|^2 - \frac{\rho}{2} \|Ax - z\|^2 \leq \frac{1}{2\gamma K} (\|x^0 - x\|^2 + \|z^0 - z\|^2 + \|y^0 - y\|^2).$$

Following the definition $\text{gap}_{\text{aug}}$, we only need to take the maximum in the variable $y \in \mathcal{Y}$ and the minimum in $x \in \mathcal{X}$ and $z \in \mathcal{Z}$.

$$\text{gap}_{\text{aug}}(\bar{x}^K, \bar{z}^K, \bar{y}^K)$$

$$= \max_{y \in \mathcal{Y}} L_{\text{aug}}(\bar{x}^K, \bar{z}^K, y) - \min_{x,z \in \mathcal{X}, \mathcal{Z}} L_{\text{aug}}(x, z, \bar{y}^K)$$

$$= \max_{y \in \mathcal{Y}} \left[ \ell(\bar{z}^K, b) + \sum_{i=1}^{n} r_i(\bar{x}_i^K) + \langle A\bar{x}^K - \bar{z}^K, y \rangle + \frac{\rho}{2} \|A\bar{x}^K - \bar{z}^K\|^2 \right]$$

$$- \min_{x,z \in \mathcal{X}, \mathcal{Z}} \left[ \ell(z, b) + \sum_{i=1}^{n} r_i(x_i) + \langle Ax - z, \bar{y}^K \rangle + \frac{\rho}{2} \|Ax - z\|^2 \right]$$

$$\leq \frac{1}{2\gamma K} (\max_{x \in \mathcal{X}} \|x^0 - x\|^2 + \max_{z \in \mathcal{Z}} \|z^0 - z\|^2 + \max_{y \in \mathcal{Y}} \|y^0 - y\|^2).$$

$$(54)$$

To complete the proof in the cases equation 54 , it remains to put $\gamma \leq \frac{1}{4} \cdot$ $\min \left\{ 1; \frac{1}{\rho}; \frac{1}{\sqrt{\lambda_{\max}(A^T A)}}; \frac{1}{\sqrt{\rho \lambda_{\max}(A^T A)}}; \frac{1}{\rho \lambda_{\max}(A^T A)}; \frac{1}{L_r}; \frac{1}{L_\ell} \right\}.$ $\qquad \square$

### D.9 PROOF OF THEOREM C.2

**Theorem D.10** (Theorem C.2). *Let $l^*$ be $L_{\ell^*}$-smooth and convex, $r$ be $L_r$-smooth and convex. Let problem (9) be solved by Algorithm 10. Then for*

$$\gamma = \frac{1}{2} \cdot \min \left\{ 1; \frac{1}{\sqrt{\lambda_{\max}(A^T A)}}; \frac{1}{L_r}; \frac{1}{L_{\ell^*}} \right\},$$

*it holds that*

$$gap_2(\bar{x}^K, \bar{y}^K) = \mathcal{O}\left( \frac{\left(1 + \sqrt{\lambda_{\max}(A^T A)} + L_{\ell^*} + L_r\right)\hat{D}^2}{K} \right),$$

*Proof.* We start the proof from (12), since the updates for $x$ variables in Algorithms 1, 10 are the same:

$$\|x^{k+1} - x\|^2 \leq \|x^k - x\|^2 - \|x^{k+1/2} - x^k\|^2$$
$$- 2\gamma \langle A(x^{k+1/2} - x), y^{k+1/2} \rangle - 2\gamma \sum_{i=1}^{n} \langle \nabla r_i(x_i^{k+1/2}), x_i^{k+1/2} - x_i \rangle$$
$$+ 2\gamma^2 \|A^T(y^{k+1/2} - y^k)\|^2 + 2\gamma^2 \sum_{i=1}^{n} \|\nabla r_i(x_i^{k+1/2}) - \nabla r_i(x_i^k)\|^2. \quad (55)$$

Using the same steps as for (12), one can obtain for $y \in \mathbb{R}^s$ from Algorithms 10,

$$\|y^{k+1} - y\|^2 \leq \|y^k - y\|^2 - \|y^{k+1/2} - y^k\|^2$$
$$- 2\gamma \langle \nabla \ell^*(y^{k+1/2}, b), y^{k+1/2} - y \rangle + 2\gamma \langle \sum_{i=1}^{n} A_i x_i^{k+1/2}, y^{k+1/2} - y \rangle$$
$$+ 2\gamma^2 \|\nabla \ell^*(y^{k+1/2}, b) - \nabla \ell^*(y^k, b)\|^2 + 2\gamma^2 \left\| \sum_{i=1}^{n} A_i(x_i^{k+1/2} - x_i^k) \right\|^2$$
$$= \|y^k - y\|^2 - \|y^{k+1/2} - y^k\|^2$$
$$- 2\gamma \langle \nabla \ell^*(y^{k+1/2}, b), y^{k+1/2} - y \rangle + 2\gamma \langle Ax^{k+1/2}, y^{k+1/2} - y \rangle$$
$$+ 2\gamma^2 \|\nabla \ell^*(y^{k+1/2}, b) - \nabla \ell^*(y^k, b)\|^2 + 2\gamma^2 \|A(x^{k+1/2} - x^k)\|^2. \quad (56)$$

Here we also use notation of $A$ and $x$. Summing up (55) and (56), we obtain

$$\|x^{k+1} - x\|^2 + \|y^{k+1} - y\|^2$$
$$\leq \|x^k - x\|^2 + \|y^k - y\|^2 - \|x^{k+1/2} - x^k\|^2 - \|y^{k+1/2} - y^k\|^2$$
$$- 2\gamma \langle A(x^{k+1/2} - x), y^{k+1/2} \rangle - 2\gamma \sum_{i=1}^{n} \langle \nabla r_i(x_i^{k+1/2}), x_i^{k+1/2} - x_i \rangle$$
$$- 2\gamma \langle \nabla \ell^*(y^{k+1/2}, b), y^{k+1/2} - y \rangle + 2\gamma \langle Ax^{k+1/2}, y^{k+1/2} - y \rangle$$
$$+ 2\gamma^2 \|A^T(y^{k+1/2} - y^k)\|^2 + 2\gamma^2 \sum_{i=1}^{n} \|\nabla r_i(x_i^{k+1/2}) - \nabla r_i(x_i^k)\|^2$$
$$+ 2\gamma^2 \|\nabla \ell^*(y^{k+1/2}, b) - \nabla \ell^*(y^k, b)\|^2 + 2\gamma^2 \|A(x^{k+1/2} - x^k)\|^2.$$

Using convexity and $L_r$-smoothness of the function $r_i$ with convexity and $L_{\ell^*}$-smoothness of the function $\ell$ and with the definition of $\lambda_{\max}(\cdot)$ as a maximum eigenvalue, we have

$$\|x^{k+1} - x\|^2 + \|y^{k+1} - y\|^2$$
$$\leq \|x^k - x\|^2 + \|y^k - y\|^2 - \|x^{k+1/2} - x^k\|^2 - \|y^{k+1/2} - y^k\|^2$$
$$+ 2\gamma \langle Ax, y^{k+1/2} \rangle - 2\gamma \sum_{i=1}^{n} [r_i(x_i^{k+1/2}) - r_i(x_i)]$$
$$- 2\gamma (l^*(y^{k+1/2}, b) - l^*(y, b)) - 2\gamma \langle Ax^{k+1/2}, y \rangle$$
$$+ 2\gamma^2 \lambda_{\max}(AA^T)\|y^{k+1/2} - y^k\|^2 + 2\gamma^2 L_r^2 \sum_{i=1}^{n} \|x_i^{k+1/2} - x_i^k\|^2$$
$$+ 2\gamma^2 L_{\ell^*}^2 \|y^{k+1/2} - y^k\|^2 + 2\gamma^2 \lambda_{\max}(A^T A)\|x^{k+1/2} - x^k\|^2.$$

With the choice of $\gamma \leq \frac{1}{2} \cdot \min \left\{ 1; \frac{1}{\sqrt{\lambda_{\max}(A^T A)}}; \frac{1}{L_r}; \frac{1}{L_{\ell^*}} \right\}$, we get

$$\|x^{k+1} - x\|^2 + \|y^{k+1} - y\|^2 \leq \|x^k - x\|^2 + \|y^k - y\|^2$$

$$+ 2\gamma\langle Ax, y^{k+1/2}\rangle - 2\gamma\sum_{i=1}^{n}[r_i(x_i^{k+1/2}) - r_i(x_i)]$$

$$- 2\gamma(l^*(y^{k+1/2}, b) - l^*(y, b)) - 2\gamma\langle Ax^{k+1/2}, y\rangle.$$

After small rearrangements, we obtain

$$l^*(y^{k+1/2}, b) - l^*(y, b) + \sum_{i=1}^{n}[r_i(x_i^{k+1/2}) - r_i(x_i)] + \langle Ax^{k+1/2}, y\rangle - \langle Ax, y^{k+1/2}\rangle$$

$$\leq \frac{1}{2\gamma}\left(\|x^k - x\|^2 + \|y^k - y\|^2 - \|x^{k+1} - x\|^2 - \|y^{k+1} - y\|^2\right).$$

Then we sum all over $k$ from $0$ to $K-1$, divide by $K$, and have

$$\frac{1}{K}\sum_{k=0}^{K-1}\left[l^*(y^{k+1/2}, b) - l^*(y, b) + \sum_{i=1}^{n}[r_i(x_i^{k+1/2}) - r_i(x_i)] + \langle Ax^{k+1/2}, y\rangle - \langle Ax, y^{k+1/2}\rangle\right]$$

$$\leq \frac{1}{2\gamma K}\left(\|x^0 - x\|^2 + \|y^0 - y\|^2\right).$$

With Jensen inequality for convex functions $\ell$ and $r_i$, one can note that

$$\ell^*\left(\frac{1}{K}\sum_{k=0}^{K-1}y^{k+1/2}, b\right) \leq \frac{1}{K}\sum_{k=0}^{K-1}\ell^*(y^{k+1/2}, b),$$

$$r_i\left(\frac{1}{K}\sum_{k=0}^{K-1}x_i^{k+1/2}\right) \leq \frac{1}{K}\sum_{k=0}^{K-1}r_i(x_i^{k+1/2}).$$

Then, with notation $\bar{x}_i^K = \frac{1}{K}\sum_{k=0}^{K-1}x_i^{k+1/2}$, $\bar{y}^K = \frac{1}{K}\sum_{k=0}^{K-1}y^{k+1/2}$, we have

$$\ell^*(\bar{y}^K, b) - \ell^*(y, b) + \sum_{i=1}^{n}[r_i(\bar{x}_i^K) - r_i(x_i)] + \langle A\bar{x}^K, y\rangle - \langle Ax, \bar{y}^K\rangle$$

$$\leq \frac{1}{2\gamma K}\left(\|x^0 - x\|^2 + \|y^0 - y\|^2\right).$$

Following the definition of $\text{gap}_2$, we only need to take the maximum in the variable $y \in \mathcal{Y}$ and the minimum in $x \in \mathcal{X}$.

$$\text{gap}_2(\bar{x}^K, \bar{y}^K)$$

$$= \max_{y\in\mathcal{Y}}\hat{L}(\bar{x}^K, y) - \min_{x\in\mathcal{X}}\hat{L}(x, \bar{y}^K)$$

$$= \max_{y\in\mathcal{Y}}\left[-\ell^*(y, b) + \sum_{i=1}^{n}r_i(\bar{x}_i^K) + y^T\left(\sum_{i=1}^{n}A_i\bar{x}_i^K\right)\right]$$

$$- \min_{x\in\mathcal{X}}\left[-\ell(\bar{y}^K, b) + \sum_{i=1}^{n}r_i(x_i) + (\bar{y}^K)^T\left(\sum_{i=1}^{n}A_ix_i\right)\right]$$

$$= \max_{y\in\mathcal{Y}}\max_{x\in\mathcal{X}}\left[\ell^*(\bar{y}^K, b) - \ell^*(y, b) + \sum_{i=1}^{n}[r_i(\bar{x}_i^K) - r_i(x_i)] + \langle A\bar{x}^K, y\rangle - \langle Ax, \bar{y}^K\rangle\right]$$

$$\leq \frac{1}{2\gamma K}\left(\max_{x\in\mathcal{X}}\|x^0 - x\|^2 + \max_{y\in\mathcal{Y}}\|y^0 - y\|^2\right).$$

To complete the proof, it remains to put $\gamma = \frac{1}{2}\cdot\min\left\{1; \frac{1}{\sqrt{\lambda_{\max}(A^T A)}}; \frac{1}{L_r}; \frac{1}{L_{\ell^*}}\right\}$. $\qquad\square$

### D.10 THREE LEMMAS

**Lemma D.11.** *If $\ell$ and $r_i$ are convex, then $L(x, z, y)$ from (4) is convex-concave.*

*Proof.* We start from checking of convexity.

$$\nabla_{(x,z)}L(x,z,y) = \begin{pmatrix} A_1^T y + \nabla r_1(x_1) \\ \dots \\ A_i^T y + \nabla r_i(x_i) \\ \dots \\ A_n^T y + \nabla r_n(x_n) \\ \nabla \ell(z,b) - y \end{pmatrix}.$$

Then, we need to check the condition of Theorem 2.1.3 from (Nesterov, 2003):

$$\langle \nabla_{(x,z)}L(x_1,z_1,y) - \nabla_{(x,z)}L(x_2,z_2,y), (x_1,z_1) - (x_2,z_2)\rangle$$

$$= \langle \begin{pmatrix} \nabla r_1(x_{1,1}) - \nabla r_1(x_{1,2}) \\ \dots \\ \nabla r_i(x_{i,1}) - \nabla r_i(x_{i,2}) \\ \dots \\ \nabla r_n(x_{n,1}) - \nabla r_n(x_{n,2}) \\ \nabla \ell(z_1,b) - \nabla \ell(z_2,b) \end{pmatrix}, \begin{pmatrix} x_{1,1} - x_{1,2} \\ \dots \\ x_{i,1} - x_{i,2} \\ \dots \\ x_{n,1} - x_{n,2} \\ z_1 - z_2 \end{pmatrix} \rangle \geq 0.$$

Here we also use that $\ell$ and $r_i$ are convex. It means that the problem (4) is convex on $(x,z)$. Next, we move to check concavity.

$$\nabla_y L(x,z,y) = \left(\sum_{i=1}^{n} A_i x_i - z\right).$$

Then, again with Theorem 2.1.3 from (Nesterov, 2003):

$$\langle \nabla_y L(x,z,y_1) - \nabla_y L(x,z,y_2), y_1 - y_2\rangle = 0 \leq 0,$$

we get that the problem (4) is concave on $y$. $\qquad\square$

**Lemma D.12.** *If $\ell$ and $r_i$ are convex, then $L_{aug}(x,z,y)$ from (8) is convex-concave.*

*Proof.* We start from checking of convexity.

$$\nabla_{(x,z)}L(x,z,y) = \begin{pmatrix} A_1^T y + \nabla r_1(x_1) + \rho A_1^T(Ax - z) \\ \dots \\ A_i^T y + \nabla r_i(x_i) + \rho A_i^T(Ax - z) \\ \dots \\ A_n^T y + \nabla r_n(x_n) + \rho A_n^T(Ax - z) \\ \nabla \ell(z,b) - y + \rho(z - Ax) \end{pmatrix}.$$

Then, we need to check the condition of Theorem 2.1.3 from (Nesterov, 2003):

$$\langle \nabla_{(x,z)}L(x_1,z_1,y) - \nabla_{(x,z)}L(x_2,z_2,y), (x_1,z_1) - (x_2,z_2)\rangle$$

$$= \langle \begin{pmatrix} \nabla r_1(x_{1,1}) - \nabla r_1(x_{1,2}) + \rho A_1^T[A(x_1 - x_2) - (z_1 - z_2)] \\ \dots \\ \nabla r_i(x_{i,1}) - \nabla r_i(x_{i,2}) + \rho A_i^T[A(x_1 - x_2) - (z_1 - z_2)] \\ \dots \\ \nabla r_n(x_{n,1}) - \nabla r_n(x_{n,2}) + \rho A_n^T[A(x_1 - x_2) - (z_1 - z_2)] \\ \nabla \ell(z_1,b) - \nabla \ell(z_2,b) + \rho[z_1 - z_2 - A(x_1 - x_2)] \end{pmatrix}, \begin{pmatrix} x_{1,1} - x_{1,2} \\ \dots \\ x_{i,1} - x_{i,2} \\ \dots \\ x_{n,1} - x_{n,2} \\ z_1 - z_2 \end{pmatrix} \rangle$$

$$= \langle \begin{pmatrix} \nabla r_1(x_{1,1}) - \nabla r_1(x_{1,2}) \\ \dots \\ \nabla r_i(x_{i,1}) - \nabla r_i(x_{i,2}) \\ \dots \\ \nabla r_n(x_{n,1}) - \nabla r_n(x_{n,2}) \\ \nabla \ell(z_1,b) - \nabla \ell(z_2,b) \end{pmatrix}, \begin{pmatrix} x_{1,1} - x_{1,2} \\ \dots \\ x_{i,1} - x_{i,2} \\ \dots \\ x_{n,1} - x_{n,2} \\ z_1 - z_2 \end{pmatrix} \rangle$$

$$+ \rho \left( \|z_1 - z_2\|^2 - 2(z_1 - z_2)^T A(x_1 - x_2) + \|A(x_1 - x_2)\|^2 \right)$$

$$= \left\langle \begin{pmatrix} \nabla r_1(x_{1,1}) - \nabla r_1(x_{1,2}) \\ \dots \\ \nabla r_i(x_{i,1}) - \nabla r_i(x_{i,2}) \\ \dots \\ \nabla r_n(x_{n,1}) - \nabla r_n(x_{n,2}) \\ \nabla \ell(z_1, b) - \nabla \ell(z_2, b) \end{pmatrix}, \begin{pmatrix} x_{1,1} - x_{1,2} \\ \dots \\ x_{i,1} - x_{i,2} \\ \dots \\ x_{n,1} - x_{n,2} \\ z_1 - z_2 \end{pmatrix} \right\rangle$$

$$+ \rho \|z_1 - z_2 - A(x_1 - x_2)\|^2$$

$$\geq 0.$$

Here we also use that $\ell$ and $r_i$ are convex. It means that the problem (8) is convex on $(x, z)$. Next, we move to check concavity.

$$\nabla_y L(x, z, y) = \left( \sum_{i=1}^n A_i x_i - z \right).$$

Then, again with Theorem 2.1.3 from (Nesterov, 2003):

$$\langle \nabla_y L(x, z, y_1) - \nabla_y L(x, z, y_2), y_1 - y_2 \rangle = 0 \leq 0,$$

we get that the problem (8) is concave on $y$. $\qquad \square$

**Lemma D.13.** *For any matrix $A = [A_1 \dots A_n]$ it holds that $\|A\| \leq \sqrt{\sum_{i=1}^n \|A_i\|^2}$.*

*Proof.* Let us consider $A = [A_1 A_2]$. Then, we have

$$\|A\| = \sup_{\|x\|^2 = 1} [\|Ax\|] = \sup_{\|x_1\|^2 + \|x_2\|^2 = 1} [\|A_1 x_1 + A_2 x_2\|] \leq \sup_{\|x_1\|^2 + \|x_2\|^2 = 1} [\|A_1 x_1\| + \|A_2 x_2\|]$$

$$= \sup_{\alpha \in [0;1]} \left[ \sup_{\|x_1\|^2 = \alpha} \|A_1 x_1\| + \sup_{\|x_1\|^2 = 1 - \alpha} \|A_2 x_2\| \right]$$

$$= \sup_{\alpha \in [0;1]} \left[ \sqrt{\alpha} \cdot \sup_{\|x_1\|^2 = 1} \|A_1 x_1\| + \sqrt{1 - \alpha} \cdot \sup_{\|x_1\|^2 = 1} \|A_2 x_2\| \right]$$

$$= \sup_{\alpha \in [0;1]} \left[ \sqrt{\alpha} \|A_1\| + \sqrt{1 - \alpha} \|A_2\| \right].$$

Optimizing $\alpha \in [0;1]$, we get that $\alpha^* = \frac{\|A_1\|^2}{\|A_1\|^2 + \|A_2\|^2}$ and

$$\|A\| \leq \sqrt{\|A_1\|^2 + \|A_2\|^2}.$$

This result can be extended to any $n$ by induction. In more details, if $A = [\tilde{A}_{n-1} A_n]$ with $\tilde{A}_{n-1} = [A_1 \dots A_{n-1}]$ and we assume that $\|\tilde{A}_{n-1}\| \leq \sqrt{\sum_{i=1}^{n-1} \|A_i\|^2}$, then we have

$$\|A\| \leq \sqrt{\|\tilde{A}_{n-1}\|^2 + \|A_n\|^2} \leq \sqrt{\sum_{i=1}^{n-1} \|A_i\|^2 + \|A_n\|^2} = \sqrt{\sum_{i=1}^n \|A_i\|^2}.$$

$\qquad \square$

### D.11 ON CONVERGENCE GAP

In our theoretical analysis, we use the criterion: $\text{gap}(x, z, y) := \max_{\tilde{y} \in \mathcal{Y}} L(x, z, \tilde{y}) - \min_{\tilde{x}, \tilde{z} \in \mathcal{X}, \mathcal{Z}} L(\tilde{x}, \tilde{z}, y)$, where $L(x, z, y) = \ell(z, b) + r(x) + y^T(Ax - z)$. Since $\min_{\tilde{x}, \tilde{z} \in \mathcal{X}, \mathcal{Z}} L(\tilde{x}, \tilde{z}, y) \leq L(x^*, z^*, y)$, we get

$$\max_{\tilde{y} \in \mathcal{Y}} L(x, z, \tilde{y}) - L(x^*, z^*, y) \leq \text{gap}(x, z, y).$$

We note that $Ax^* = z^*$, then

$$\max_{\tilde{y} \in \mathcal{Y}} L(x, z, \tilde{y}) - L(x^*, z^*, y) = [\ell(z, b) + r(x) + \max_{\tilde{y} \in \mathcal{Y}} \tilde{y}^T(Ax - z)] - [\ell(z^*, b) + r(x^*) + (y)^T(Ax^* - z^*)]$$

$$= [\ell(z, b) + r(x) + \max_{\tilde{y} \in \mathcal{Y}} \tilde{y}^T(Ax - z)] - [\ell(Ax^*, b) + r(x^*)].$$

When taking maximum for $y \in \mathcal{Y}$ we can define $\mathcal{Y}$ as we need. In particular, we can choose $\mathcal{Y} = [y \in R^s \mid \|y\|_\infty \leq C]$ for some $C > 0$. Then

$$\max_{\tilde{y} \in \mathcal{Y}} \tilde{y}^T(Ax - z) = C\|Ax - z^k\|_1 \geq C\|Ax - z\|.$$

Finally, we get

$$\text{gap}(x, z, y) \geq [\ell(z, b) + r(x) - \ell(Ax^*, b) + r(x^*)] + C\|Ax^k - z^k\| = \text{newgap}(x, z).$$

If it holds that $\text{gap}(x, z, y) \leq \varepsilon$, we guarantee that $\text{newgap}(x, z, y) \leq \varepsilon$. The question that arises is whether $\text{newgap}(x, z, y) \leq \varepsilon$ implies that $[\ell(z, b) + r(x) - \ell(Ax^*, b) + r(x^*)]$ as well as $\|Ax - z\|$ are also "small" in the sense that they are smaller than $\varepsilon$ (up to constants). In general, the answer is no: $[\ell(z, b) + r(x) - \ell(Ax^*, b) + r(x^*)]$ might be very small (and negative), and $\|Ax - z\|_2$ can be very large. But Theorem 3.60 from (Beck, 2017) shows that if $C$ is large enough such a conclusion can be drawn. In particular, if $\text{newgap}(x^k, z^k, y^k) \leq \varepsilon$ then $C\|Ax^k - z^k\|_2 \leq \varepsilon$ and we have $Ax^k \to z^k$.

## D.12   ON TUNING OF STEPSIZE

We can rewrite the original problem (1) in the following way:

$$\min_{x \in \mathbb{R}^d} \quad [\ell(Ax, b) + r(x)] = \left[\ell\left(\frac{1}{\beta} \cdot \beta Ax, b\right) + r(x)\right] = \left[\tilde{\ell}\left(\tilde{A}x, b\right) + r(x)\right],$$

where $\tilde{\ell}(y, b) = \ell\left(\frac{y}{\beta}, b\right)$ and $\tilde{A} = \beta A$. Next, we can estimate $L_{\tilde{\ell}}$ and $\lambda_{\max}(\tilde{A}^T\tilde{A})$:

$$\|\nabla\tilde{\ell}(y_1, b) - \nabla\tilde{\ell}(y_2, b)\| = \|\nabla_y\ell\left(\frac{y_1}{\beta}, b\right) - \nabla_y\ell\left(\frac{y_2}{\beta}, b\right)\|$$

$$= \frac{1}{\beta}\|\nabla\ell\left(\frac{y_1}{\beta}, b\right) - \nabla\ell\left(\frac{y_2}{\beta}, b\right)\| \leq \frac{L_\ell}{\beta^2}\|y_1 - y_2\|,$$

$$\lambda_{\max}(\tilde{A}^T\tilde{A}) = \lambda_{\max}(\beta^2 A^T A) = \beta^2\lambda_{\max}(A^T A).$$

We get that $L_{\tilde{\ell}} = \frac{L_\ell}{\beta^2}$ and $\lambda_{\max}(\tilde{A}^T\tilde{A}) = \beta^2\lambda_{\max}(A^T A)$.

Our goal is to equivalize $L_{\tilde{\ell}}$ and $\sqrt{\lambda_{\max}(\tilde{A}^T\tilde{A})}$ in Theorem 2.2 to make stepsize bigger for free. Then

$$\frac{L_\ell}{\beta^2} = L_{\tilde{\ell}} = \sqrt{\lambda_{\max}(\tilde{A}^T\tilde{A})} = \beta\sqrt{\lambda_{\max}(A^T A)}$$

$$\Rightarrow \quad \beta = \frac{L_\ell^{1/3}}{\lambda_{\max}^{1/6}(A^T A)} \quad \Rightarrow \quad L_{\tilde{\ell}} = L_\ell^{1/3}\lambda_{\max}^{1/3}(A^T A).$$

Hence, the bound on the stepsize in Theorem 2.2 become

$$\gamma = \tfrac{1}{2} \cdot \min\{1; \frac{1}{\sqrt[3]{L_\ell\lambda_{\max}(A^T A)}}; \frac{1}{L_r}\}.$$

This, in turn, modifies the convergence result of the theorem as follows:

$$\text{gap}(\bar{x}^K, \bar{z}^K, \bar{y}^K) = \mathcal{O}\left(\frac{(1 + \sqrt[3]{L_\ell\lambda_{\max}(A^T A)} + L_r)D^2}{K}\right).$$

## E  ADDITIONAL EXPERIMENTS

In the main part (Figure 1 of Section 6) we shown that the concept of the saddle point reformulation and Algorithm 1 for its solution is competitive in the deterministic case. Here we present additional experiments.

As in the main part, we conduct experiments on the linear regression problem: $\min_{x \in \mathbb{R}^d} f(x) = \frac{1}{2}\|Ax - b\|^2 + \lambda\|x\|_2^2$. We take `mushrooms`, `a9a`, `w8a` and `MNIST` datasets from LibSVM library (Chang & Lin, 2011). We vertically (by features) uniformly divide the whole dataset between 5 devices.

First we repeat the same experiments as in the main part, but now for each method we tune the parameters using a grid search. The results are shown in Figure 3. If we compare Figure 1 and Figure 3, the one method that accelerates the most is Algorithm 1.

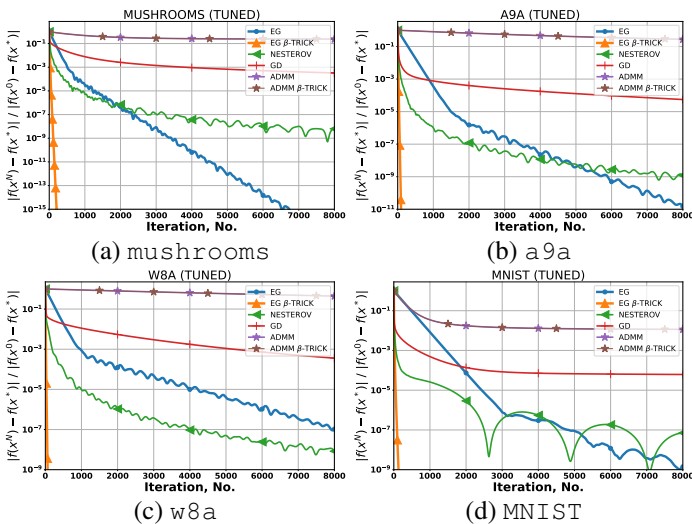

(a) `mushrooms`  (b) `a9a`

(c) `w8a`  (d) `MNIST`

Figure 3: Comparison of tuned methods for solving the VFL problem in different formulations: minimization (`GD`, `Nesterov`) and saddle point (`ADMM`, `ExtraGradient`/Algorithm 1). The comparison is made on LibSVM datasets `mushrooms`, `a9a`, `w8a` and `MNIST`.

Next, we want to consider modifications of Algorithm 1 and show that they can speed Algorithm 1 up from different points of view.

In the first group of experiments with modifications (Figure 4), we test the performance of Algorithm 2. We use the compression operator $Q = \text{RandK}\%$, which is a random selection coordinates: 100% (Algorithm 1), 50%, 25%, 10%. An important detail is that we set the same random generator and seed on each of the devices. Therefore, at each iteration we send random coordinates, but they are the same for all devices. The comparison is made in terms of the number of full vectors transmitted. In contrast to the main part, here we tune stepsizes, since with the theoretical step it is not possible to achieve the best acceleration compared to Algorithm 1. The comparison is done in two settings: the basic one and using the $\beta$-trick (see disscusion after Corollary 2.3). The results show that compression can indeed speed up the communication process.

In the second group of experiments with modifications (Figure 5), we test the performance of Algorithm 3 in comparison with Algorithm 2. We use the compression operators $C = \text{TopK}\%$ (for Algorithm 3), which is a greedy selection coordinates, and $Q = \text{RandK}\%$ (for Algorithm 2) with K = 25% and 10%. The comparison is made in terms of the number of full vectors transmitted. As in the previous experiment, we tune stepsizes. In experiments, we see that unbiased compression outperforms biased compression almost always. In the horizontal case, the opposite is usually true (Beznosikov et al., 2020). We attribute this effect to the fact that in the case of RandK% compression we set the same random generator and seed on different devices and therefore they send the same random coordinates at each iteration. In the case of using TopK% operator we cannot do this, therefore convergence is worse.

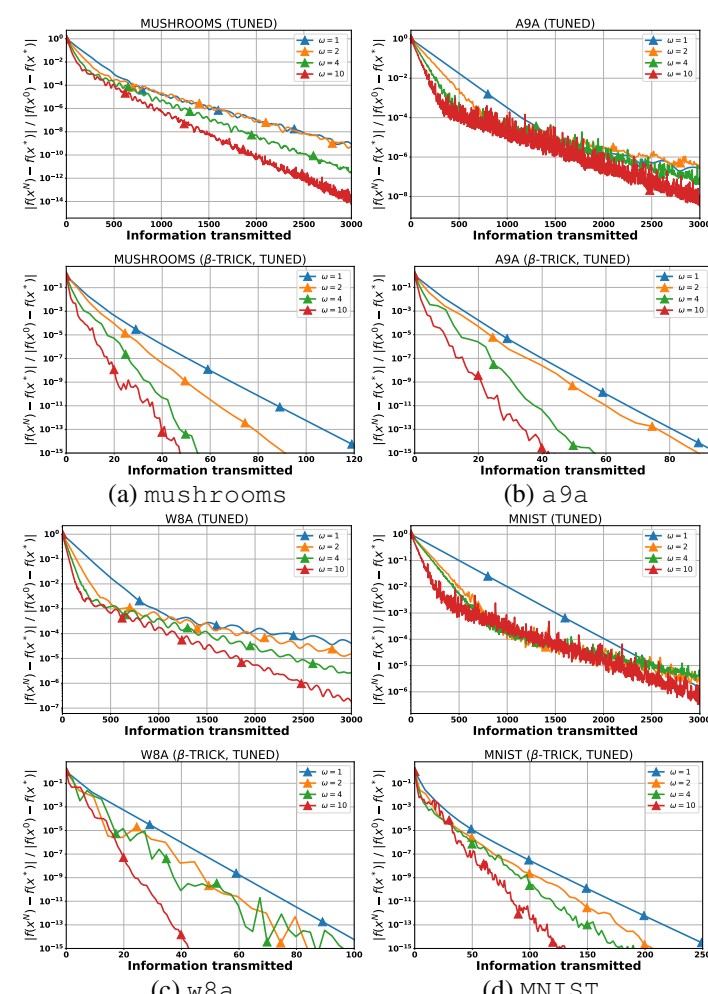

Figure 4: Comparison of Algorithm 2 for solving the VFL problem (4). The comparison is made on LibSVM datasets mushrooms, a9a, w8a and MNIST. The compression operator $Q = \text{RandK}\%$. The criterion for comparison is the number of full vectors transmitted. The top line reflects the work of methods on the basic problem, the bottom line solves the problem with the $\beta$-trick (see disscusion after Corollary 2.3).

In the third group of experiments with modifications (Figure 6), we test the performance of Algorithm 4. At each iteration we generate only 2 devices out of 5 that communicate. The comparison is made in terms of the number of full vectors transmitted from all devices. As in the previous experiments, we tune stepsizes. The results show that the partial participation technique can indeed speed up the communication process in terms of the number of devices communicated.

In the fouth group of experiments with modifications (Figure 7), we test the performance of Algorithm 5. We use a random selection coordinates: 100% (Algorithm 1), 50%, 25%, 10%. An important detail is that we set the same random generator and seed on each of the devices. The comparison is made in terms of the computational powers. Here we also tune stepsizes. The results show that the random coordinate selection can indeed speed up the computational process.

### E.1 TECHNICAL DETAILS

Our algorithms is written in Python 3.10, with the use of PyTorch optimization library. We implement a simulation of distributed optimization system on a single server. Our server is AMD Ryzen Threadripper 2950X 16-Core Processor @ 2.2 GHz CPU and x2 NVIDIA GeForce GTX 1080 Ti GPU.

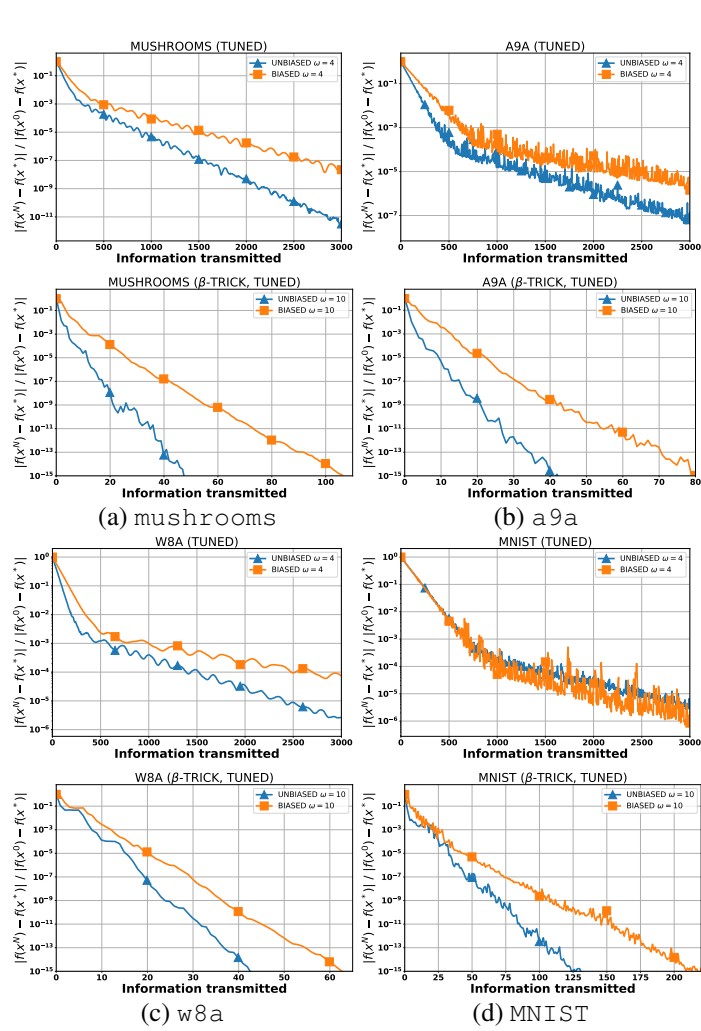

Figure 5: Comparison of Algorithm 3 and Algorithm 2 for solving the VFL problem (4). The comparison is made on LibSVM datasets mushrooms, a9a, w8a and MNIST. The compression operators $C = \text{TopK}\%$ and $Q = \text{RandK}\%$. The criterion for comparison is the number of full vectors transmitted. The top line reflects the work of methods on the basic problem, the bottom line solves the problem with the $\beta$-trick (see disscusion after Corollary 2.3).

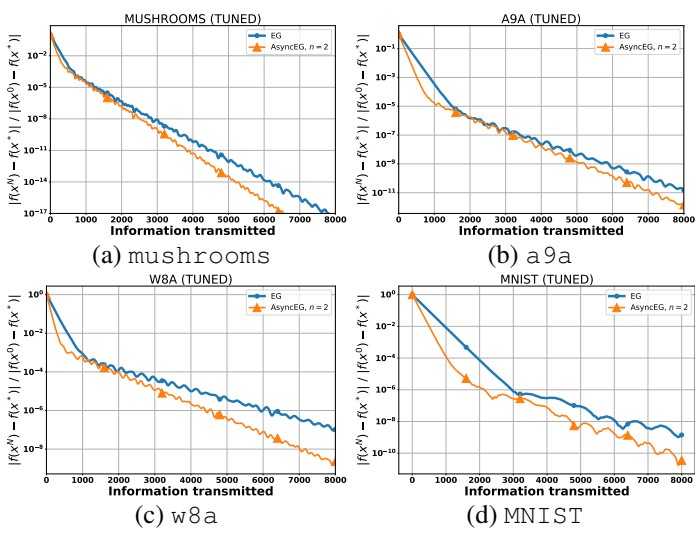

Figure 6: Comparison of Algorithm 4 for solving the VFL problem (4). The comparison is made on LibSVM datasets mushrooms, a9a, w8a and MNIST. The criterion for comparison is the number of full vectors transmitted.

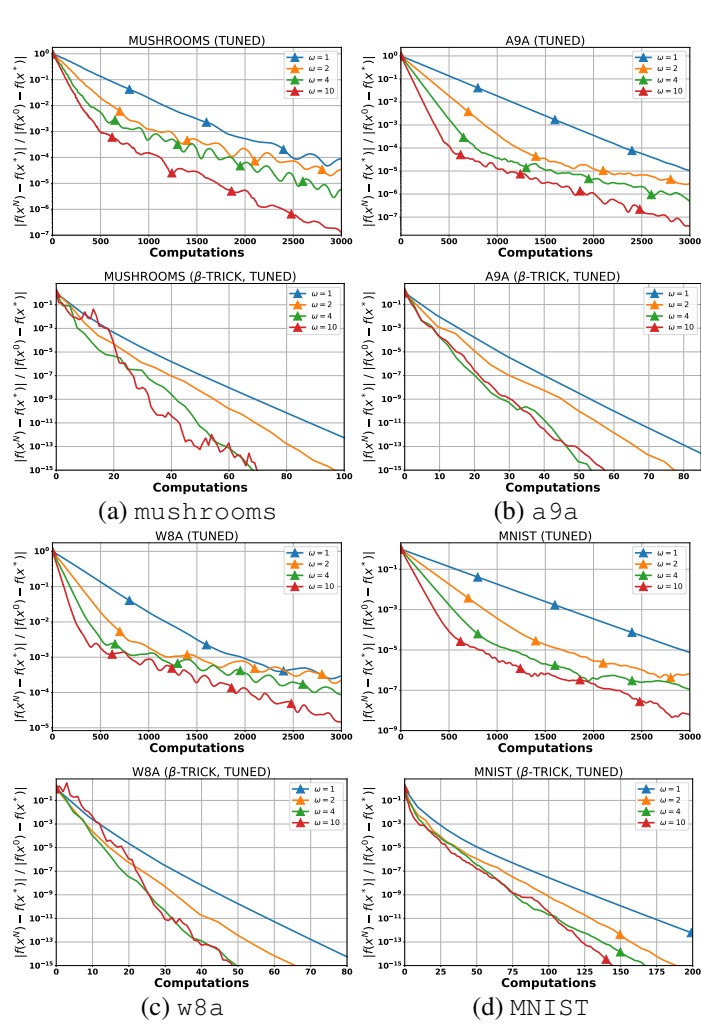

Figure 7: Comparison of Algorithm 5 for solving the VFL problem (4) on LibSVM datasets mushrooms, a9a, w8a and MNIST. The criterion for comparison is the computational powers. The top line reflects the work of methods on the basic problem, the bottom line solves the problem with the $\beta$-trick (see disscusion after Corollary 2.3).

