# OpenReview forum: "Exploring New Frontiers in Vertical Federated Learning: the Role of Saddle Point Reformulation"
_ICLR.cc/2025/Conference — ICLR 2025 Conference Withdrawn Submission_

### Official Review · Reviewer_T3kx · 2024-10-18

**Soundness:** 2
**Presentation:** 1
**Contribution:** 2
**Rating:** 3
**Confidence:** 5

**Summary:**

This paper explores new methods for Vertical Federated Learning (VFL) by reformulating the learning process using a saddle point framework instead of the traditional minimization approach. The proposed approach in the deterministic case enables solving VFL problems with enhanced convergence guarantees in terms of the eigenvalue of the data matrix. The authors also propose stochastic algorithms tailored to this reformulation and suggest modifications to address practical challenges such as communication efficiency and computational cost by implementing compression, partial participation, and coordinate selection, respectively. The paper validates the proposed methods through numerical experiments.

**Strengths:**

1. This paper proposes a new minimax framework for the VFL problem. The method has a better complexity constant compared to accelerated gradient descent.

2. The theoretical guarantee in the modification of quantization for the saddle point problem in VFL is novel.

**Weaknesses:**

1. Insufficient Preliminaries:
The paper lacks clear explanations of key concepts such as Vertical Federated Learning (VFL) modeling and biased/unbiased compression. It would be easier to understand the paper if a "Preliminaries" section defining VFL and compression techniques were added before diving into the technical details.  Especially:

1.1 in Section 3.1, the introduction of compression techniques is missing, and key notations, such as  $b^k$ appear without proper definition. This makes it difficult to understand how Algorithm 2 is formulated. The authors should include a notation table or glossary at the beginning of each major section or explicitly define each new symbol when it is first introduced.

1.2 The reference to noise affecting $Z$ (line 154) is unclear, as the source of the noise and the conditions under which it occurs are not specified. It would improve clarity to explain the origin of the noise and provide an example of when it might arise.


1.3 In the experiments section, SSP (line 462) is introduced without any prior definition. Please add explanation to SSP to improve the clarity.


2. Disorganized Presentation: The structure in lines 205–217 is difficult to follow as specified as follows:

2.1: it mainly focuses on previous work, but the purpose of mentioning it in this context is unclear. It would be clearer if the authors separated lines 242–245 into two parts: one for discussing the differences and merits compared to each mentioned previous approach, and the other for mathematically stating the relation between equation (5) and \(gap^*\).

2.2: there are multiple claims without supporting mathematical proofs. For example, the statement:
"Criterion (5) can also be used for unconstrained/unbounded problems. To do this, one can use the trick from (Nesterov, 2007) and introduce bounded sets" in line 207, and "one can show that in Theorem 2.2 ... we can use the criterion" in line 210. It would strengthen the work to provide rigorous proofs for these claims, especially if they support the contribution of this research.

2.3: the explanation in lines 242–245 regarding the convergence criterion using \(g(x,y) = xy\) (from lines 205–217) is confusing. Specifically, the claim that \(gap^*(x,y,z) = 0\) does not make sense because the example \(g(x,y) = xy\) does not contain a third variable \(z\).


4. Clarity, Missing Verbs and Poor Sentence Structure:

4.1 The use of "it" in line 172 is ambiguous, making the meaning unclear. To enhance readability, consider explicitly stating what "it" refers to in this context. Replacing vague pronouns with precise references will help avoid confusion.

4.2 In line 227-228, the sentence starting with "one cannot..." lacks a verb, which makes it incomplete. This is problematic because this sentence is critical for comparing the new results with prior work.

4.3 in line 520-521, the sentence is incomplete ("but only..."), further complicating readability.

I suggest the authors carefully proofread this work again for a better presentation.

5. Inadequate Experiments:
Since the tasks involve classification problems, I recommend that the experiments report test accuracy to effectively evaluate the proposed method.

On the other hand, this work proposes a new method for practical scenarios, such as partial attendance and compression. However, the impact of these proposed methods would be more convincing if accompanied by experiments demonstrating their effectiveness compared to previous approaches.
On the other hand, I did not find a discussion of related work in Vertical Federated Learning (VFL) that specifically addresses partial attendance and the use of quantization. Including a dedicated related work section summarizing prior research on these topics would provide better context and help position the contributions of this paper.

**Questions:**

Please refer to the weakness.

---

### Official Review · Reviewer_fhae · 2024-11-03

**Soundness:** 3
**Presentation:** 3
**Contribution:** 2
**Rating:** 6
**Confidence:** 3

**Summary:**

This paper proposes a saddle point reformulation for the vertical federated learning (VFL) and provides extragradient-based algorithms to solve the reformulation with the convergence rate $O(1/K)$ on the expected primal-dual gap of the reformulation. The authors also present some extension to non-convex models. The authors conduct numerical experiments in VFL by using linear regression with $l_2$-norm regularizations and using the ResNet18 neural network.

**Strengths:**

The authors start with the basic reformulation in Section 2, and then thoroughly consider several stochastic modifications such as quantization for effective communications, biased compression, partial participation for asynchronous communications and coordinate descent for reducing local computational cost. For each case, a modified algorithm is presented with the complete proof on the convergence rate $O(1/K)$.

**Weaknesses:**

There seems a gap between the considered linear models and non-convex models in the formulation (4) on page 3 and (7) on page 9. See Questions for the details.

**Questions:**

For solving convex-concave saddle point formulations with bilinear coupling terms (on $(x,z)$ and $y$ in the context of the paper), there are several well-studied methods such as the extragradient (or called mirror-prox) method [1] and the primal-dual hybrid gradient (PDHG) method [2,3]. [2,Section 4.1] mentions that the extragradient method has the same convergence rate $O(1/K)$.

Then, [4] extends the PDHG method to the case of convex-concave saddle point problems, where the coupling term $\Phi(x,z,y)$, not necessarily bilinear, is a continuous function with certain differentiability properties, convex in $(x,z)$ and concave $y$. [4] also proves the same convergence rate $O(1/K)$, which attains the lower bound proved in [5].

Question: Could the authors possibly show some examples in VFL where modifies (7) on page 9 by a nonlinear coupling term $y^\top(\sum_{i=1}^n g_i(A_i,w_i)x_i-z)$, convex in $(x,w,z)$ and concave in $y$? I suppose that these case can also be solved with the same convergence rate $O(1/K)$ based on the literature in optimization.


[1] Prox-Method with Rate of Convergence $O(1/t)$ for Variational Inequalities Arkadi Nemirovski SIAM J. Optim. 15(1), 229-251 (2004)

[2] A First-Order Primal-Dual Algorithm for Convex Problems with Applications to Imaging Antonin Chambolle, Thomas Pock J. Math Imaging Vis. 40(1), 120-145 (2011)

[3] On the ergodic convergence rates of a first-order primal-dual algorithm Antonin Chambolle, Thomas Pock Math. Program. 159(1-2), 253-287 (2016)

[4] A Primal-Dual Algorithm with Line Search for General Convex-Concave Saddle Point Problems Erfan Yazdandoost Hamedani, Necdet Serhat Aybat SIAM J. Optim. 31(2), 1299-1329 (2021)

[5] Lower complexity bounds of first-order methods for convex-concave bilinear saddle-point problems Yuyuan Ouyang, Yangyang Xu Math. Program. 185(1-2), 1-35 (2021)

---

### Official Review · Reviewer_jdC2 · 2024-11-03

**Soundness:** 3
**Presentation:** 2
**Contribution:** 2
**Rating:** 5
**Confidence:** 3

**Summary:**

This paper studies the vertical federated learning (VFL) problem with the linear model by its convex-concave saddle point reformulation, which separates the data matrix $A$ and the loss function $\ell$. Based on this reformulation, the authors propose an algorithm EGVFL for VFL based on the celebrated ExtraGradient method for convex-concave saddle point problems. They establish the convergence rate of EGVFL in terms of the duality gap, assuming $\ell$ and the regularizer $r$ are convex and smooth. The paper also provides convergence guarantees for EGVFL variants with biased and unbiased communication compression, partial participation, and local steps.

**Strengths:**

- The saddle point reformulation seems to be natural and well-motivated.
- When the model is linear, the authors established extensive convergence theory for the proposed algorithm and its extensions, accommodating key features such as communication compression, partial participation, and local steps. Besides, the convergence rate of EG improves upon GD in terms of $\lambda_{\max}(A^\top A)$.

**Weaknesses:**

- The proposed algorithms only have convergence guarantees for VFL with the linear model and the extension for nonconvex problems remains heuristic.
- In the experiments, only general-purpose optimizers are compared while existing algorithms specifically designed for VFL (e.g., [1] and its baselines) are completely missing.
- Figure 1 and Figure 2 only present the relative objective gap w.r.t. the number of iterations. This might be unfair since the per-iteration computational and communication costs of the proposed algorithms and baselines are different. For example, EG requires one extra communication per round than GD, and algorithms based on the saddle point reformulation also need to update auxiliary variables z and y.


[1] Xie, Chulin, Pin-Yu Chen, Qinbin Li, Arash Nourian, Ce Zhang, and Bo Li. "Improving privacy-preserving vertical federated learning by efficient communication with admm." In 2024 IEEE Conference on Secure and Trustworthy Machine Learning (SaTML), pp. 443-471. IEEE, 2024.

**Questions:**

- From the first page, the algorithm and its proof techniques seem to apply to general VFL problems, which could be misleading. I think the authors should clarify at the very beginning that the main theoretical results in this paper are only for linear models.
- In the experiments, could also show the relative objective gap w.r.t. communicated bits / total time?
- The sentence beginning with "One cannot..." in Line 227 appears to be broken.

---

### Official Review · Reviewer_7pWJ · 2024-11-09

**Soundness:** 3
**Presentation:** 3
**Contribution:** 3
**Rating:** 6
**Confidence:** 2

**Summary:**

The paper proposes a saddle point reformulation of the Vertical Federated Learning (VFL) problem, allowing for more efficient and privacy-preserving optimization compared to traditional minimization methods. The authors introduce a deterministic algorithm with several practical stochastic modifications that improve communication, handle asynchronous participation, and reduce computation costs.

**Strengths:**

1.Reformulating Vertical Federated Learning (VFL) as a saddle point problem is interesting and novel, it can offer an alternative to traditional minimization methods that could address VFL-specific challenges more effectively.
2.The paper presents comprehensive theoretic results.
3.The practical modifications for improving communication efficiency, asynchronous participation, and computational costs are well-aligned with real-world VFL challenges.

**Weaknesses:**

1. While the paper introduces several modifications to the basic deterministic algorithm, such as quantization, biased compression, and asynchronous participation, these are presented with high mathematical density and minimal illustrative examples. This makes it challenging for audience like me that are less familiar with saddle point methods and vertical federated learning to fully grasp each modification's practical implications and implementation nuances. I suggest the authors to enhance accessibility by maybe providing more intuitive explanations or visual illustrations (e.g., flow diagrams) of the modified algorithms.

2. The experiments mainly focus on benchmark datasets with linear regression and neural network fine-tuning tasks. The paper would benefit from exploring additional VFL scenarios that could showcase the flexibility of the proposed approach in handling diverse model architectures or real-world vertical partitioning cases. I am uncertain whether the datasets and settings used in the experiments are standard for the VFL field. If they are not, it would be beneficial to include a wider variety of commonly used VFL benchmarks to strengthen the empirical validation.

**Questions:**

Similar to the above section: can the authors clarify whether the datasets and experimental settings used (e.g., mushrooms, a9a, w8a, MNIST, CIFAR-10; uniformly dividing between 5 clients) are considered standard benchmarks in the VFL field? If not, would the authors consider incorporating more diverse and commonly-used VFL benchmarks that reflect real-world vertical partitioning scenarios? This would help assess the generalizability of the proposed methods across typical VFL applications.

---

### Note · Authors · 2024-12-02

I have read and agree with the venue's withdrawal policy on behalf of myself and my co-authors.